# BENCHMARKING EMPIRICAL PRIVACY PROTECTION FOR ADAPTATIONS OF LARGE LANGUAGE MODELS

**Bartłomiej Marek**[*], **Lorenzo Rossi**[*], **Vincent Hanke, Xun Wang,**
**Michael Backes, Franziska Boenisch, Adam Dziedzic**[†]
CISPA Helmholtz Center for Information Security

## ABSTRACT

Recent work has applied differential privacy (DP) to adapt large language models (LLMs) for sensitive applications, offering theoretical guarantees. However, its practical effectiveness remains unclear, partly due to LLM pretraining, where overlaps and interdependencies with adaptation data can undermine privacy despite DP efforts. To analyze this issue in practice, we investigate privacy risks under DP adaptations in LLMs using state-of-the-art attacks such as *robust membership inference* and *canary data extraction*. We benchmark these risks by systematically varying the adaptation data distribution, from exact overlaps with pretraining data, through in-distribution (IID) cases, to entirely out-of-distribution (OOD) examples. Additionally, we evaluate how different adaptation methods and different privacy regimes impact the vulnerability. Our results show that distribution shifts strongly influence privacy vulnerability: the closer the adaptation data is to the pretraining distribution, the higher the practical privacy risk at the same theoretical guarantee, even without direct data overlap. We find that parameter-efficient fine-tuning methods, such as LoRA, achieve the highest empirical privacy protection for OOD data. Our benchmark identifies key factors for achieving practical privacy in DP LLM adaptation, providing actionable insights for deploying customized models in sensitive settings. Looking forward, we propose a structured framework for *holistic* privacy assessment beyond adaptation privacy, to identify and evaluate risks across the full pretrain-adapt pipeline of LLMs.

## 1 INTRODUCTION

The use of *pretrained* large language models (LLMs) for sensitive downstream tasks, such as medical decision making, has grown rapidly (Labrak et al., 2024; Chen et al., 2023; Van Veen et al., 2024). To offer protection for the private data used to *adapt* the LLMs to these sensitive tasks, differential privacy (DP) (Dwork, 2006; Dwork et al., 2014) has emerged as a gold standard (Yu et al., 2021; 2022; Li et al., 2022; Duan et al., 2023a; Mehta et al., 2023). However, adapting a pretrained LLM with DP may not always provide the anticipated privacy protections (Tramèr et al., 2024). The challenge arises from potential overlap or complex interdependencies between data used to pretrain the LLMs and the adaptation dataset. The problem is exacerbated by the fact that for most LLMs, their pretraining datasets are not disclosed (OpenAI, 2023; Qwen et al., 2025; Touvron et al., 2023), rendering a structured reasoning of the interdependencies with the private adaptation data impossible.

While prior work has investigated privacy risks stemming from LLM pretraining (Carlini et al., 2023b;a), post-hoc leakage in non-private adaptations (Zhu et al., 2024), or auditing DP adaptations via synthetic canaries (Panda et al., 2024), we still lack a structured understanding of the *empirical privacy risks* of DP adaptations. This is a critical gap. Without a clear understanding of the practical risks, LLM practitioners are left with little guidance on how to privately apply LLMs in privacy-sensitive settings, including critical questions like: which adaptation method to use, what pretrained model is best given the private adaptation data, and what privacy levels will be protective enough.

---

[*]Equal contribution.

[†]For correspondence, please contact Franziska Boenisch (`boenisch@cispa.de`) and Adam Dziedzic (`dziedzic@cispa.de`).

To close this gap, we conduct a comprehensive benchmark evaluation that sheds light on the empirical leakage introduced by DP adaptations. We evaluate a wide range of private adaptation strategies, including full and last-layer DP fine-tuning (Li et al., 2022), parameter-efficient fine-tuning (PEFT) methods such as DP-LoRA (Hu et al., 2022; Yu et al., 2022), DP-Prefix Tuning (Liu et al., 2021), as well as DP prompting schemes (Duan et al., 2023a).

To assess leakage, we focus on the *Robust Membership Inference Attack* (RMIA) (Zarifzadeh et al., 2024), which represents the strongest state-of-the-art threat model for auditing LLM privacy, and complement this with *data extraction attacks* (Tramèr et al., 2022; Carlini et al., 2021; 2019) to evaluate more severe forms of information leakage. For the latter, we include *canary* data into the adaptation set and measure its exposure. A general overview of privacy auditing for adapted LLMs is provided in Figure 1.

Figure 1: **Setup for Privacy Auditing of private LLM Adaptations.**

We systematically analyze a spectrum of possible distributions for the adaptation data with respect to the pretraining data—ranging from data perfectly overlapping with the pretraining data, over IID scenarios, to entirely OOD examples— to understand the possible privacy implications for all setups. Our benchmark spans *six* datasets drawn from diverse domains, *four* adaptation methods, and *seven* pretrained LLMs trained on the Pile dataset (Gao et al., 2020), allowing for a complete analysis. These models, of various sizes and architectures, enable comprehensive comparisons across different setups. Furthermore, we include two recent fully open-source models, OLMo 1B (Groeneveld et al., 2024a) and OLMo2 1B (OLMo et al., 2024), though the lack of known validation datasets constrains the analysis of these models. We analyze a broad spectrum of privacy regimes from no privacy to high privacy, to understand the associated risks. Our study is guided by a central question: *What are the empirical privacy risks for the adaptation data that result from DP adaptations?*

Looking ahead, we emphasize the need to jointly audit privacy risks from pretraining and adaptation, as well as their interplay, since LLMs may leak information from either stage. To address this, we propose a new structured framework for holistic privacy assessment across the full pretrain-adapt pipeline. It defines four key audit stages: (1) pretraining, (2) adaptation, (3) their joint interaction, and (4) post-adaptation auditing of pretraining. To formally ground these audits and make them instantiable, we redefine each stage's membership inference game (Yeom et al., 2018; Jayaraman et al., 2020). We hope this formalization and our practical insights from the benchmark will guide researchers in developing future assessments and help practitioners deploy customized LLMs responsibly in sensitive domains.

## 2 BACKGROUND AND RELATED WORK

**Differential Privacy.** The mathematical framework of DP (Dwork, 2006) formalizes the intuition that privacy guarantees can be obtained when a randomized mechanism $\mathcal{M}$ executed on two neighboring datasets $D, D'$ that differ in only one data point, yields roughly the same result, *i.e.,*

$$\Pr[\mathcal{M}(D) \in S] \leq e^{\varepsilon} \cdot \Pr[\mathcal{M}(D') \in S] + \delta. \tag{1}$$

The privacy parameter $\varepsilon$ specifies how much the result can differ, $\delta$ is the probability of failure to meet that guarantee. There are two canonical algorithms to implement DP guarantees in machine learning (ML): DPSGD (*Differentially Private Stochastic Gradient Descent*) (Abadi et al., 2016), which extends standard stochastic gradient descent with clipping and noising gradients, and PATE (*Private Aggregation of Teacher Ensembles*) (Papernot et al., 2017; 2018), which is an inference time algorithm that privately transfers knowledge from an ensemble of teachers to a public student model.

**Private Adaptations of LLMs.** LLMs are pretrained on extensive amounts of public data, followed by adaptations to private downstream tasks. The existing methods for private LLM adaptations fall into two categories: (1) *private tuning methods*, such as PrivateLoRA (Yu et al., 2022) or PromptDPSGD (Duan et al., 2023a), that rely on access to the LLM gradients and are based on the

DPSGD algorithm, and (2) *private in-context learning (ICL) methods*, such as DP-ICL (Wu et al., 2024) or PromptPATE (Duan et al., 2023a), which require only API (black-box) access to the LLM and are based on PATE. See Section A.1 for details.

**Membership Inference Attacks.** A membership inference attack (MIA) (Shokri et al., 2017; Zarifzadeh et al., 2024; Shi et al., 2024b; Carlini et al., 2022) aims to determine whether a specific data point can be identified as part of a model's training set. This approach plays a crucial role in applications ranging from privacy assurance (Steinke et al., 2023; Mahloujifar et al., 2025; Rossi et al., 2025b) to identifying protected or copyrighted content embedded in pretraining data (Shafran et al., 2021). While most MIA research has focused on supervised learning settings (Carlini et al., 2022), new advancements reveal their broader relevance. Duan et al. (2023b) introduced a discrete-prompt-based MIA, disclosing vulnerabilities in proprietary LLMs like GPT-3, which risk leaking private information through prompt-based queries (Duan et al., 2023a). See Section A.2 for an in-depth discussion of the existing attacks.

**Canary Exposure and Data Extraction Attacks.** An alternative to membership inference attacks (MIAs) for evaluating privacy leakage in machine learning models is to measure the *exposure* of training data. Given a universe of candidates $\mathcal{U}$ and an attacker's ranking $\hat{Z}$ by likelihood of membership, the exposure of a target sample $z \in \mathcal{U}$ is defined as:

$$\textbf{exposure}(z, \hat{Z}) = \log_2 |\mathcal{U}| - \log_2\big(\text{rank}(z; \hat{Z})\big). \qquad (2)$$

This score is maximal when $z$ is ranked most likely and zero when ranked least likely. In a complementary vein, *extractability* quantifies how readily a model emits a secret string when prompted. A suffix $s$ is said to be *extractable with $k$ tokens of context* if there exists some prefix $p$ of length $k$ such that, under greedy decoding, the model outputs $s$ immediately following $p$. When $s$ is sufficiently long and random, its extractability serves as a practical metric of memorization in LLMs. Further discussion appears in Section A.3.

**Benchmarking Privacy Vulnerabilities.** Zhu et al. (2024) introduced *PrivAuditor*, which systematically and empirically evaluates the privacy leakage from LLM adaptations. In contrast to our work, they focus on *non-private* adaptations only. Li et al. (2024a) evaluated the privacy leakage of private LLMs adaptations through empirical privacy attacks, such as data extraction, MIAs, and embedding-level privacy attacks. This benchmark focuses mostly on tradeoffs between privacy and utility, highlighting the complexity of balancing them. Contrary to our work, this work does not explore the relationship between the pretraining data and the fine-tuning data. *LLM-PBE* (Li et al., 2024b) empirically evaluates privacy risks throughout the LLM lifecycle, including pretraining, fine-tuning, and querying. Zhou et al. (2025) investigated potential data leakage across widely used software engineering benchmarks.

## 3 EXPERIMENTAL SETUP

We begin by detailing the setup used for our benchmark. Further details are presented in Section B.

**Models and Pretraining Data.** Our work focuses primarily on the Pythia (Biderman et al., 2023) and GPT-Neo (Black et al., 2021) families, both of which were trained on the Pile dataset (Gao et al., 2020), in addition to the newer open-source models, OLMo (Groeneveld et al., 2024b) and OLMo2 (OLMo et al., 2024). To benchmark the effects over various model sizes, we use Pythia 1.4B, Pythia 1B, Pythia 410M, Pythia 160M, Pythia 70M, GPT Neo 1.3B, GPT Neo 125 M, OLMo 1B (Groeneveld et al., 2024b), and OLMo 2 1B (OLMo et al., 2024).[1] The Pile dataset (Gao et al., 2020) is an 800GB collection of diverse English-language datasets, including text from sources such as books, academic papers, or source code repositories. In all cases where a specific model is not explicitly mentioned, we use Pythia 1B as the default model.

**Adaptation Datasets.** We categorize the datasets used in our experiments into **in-distribution (IID)** and **out-of-distribution (OOD)**, depending on their relationship to the pretraining data. IID datasets

---

[1]Note that our analyses **cannot** be conducted on closed models, such as GPT-4 (Achiam et al., 2023) or Gemini (Comanici et al., 2025) since 1) their APIs do not offer the gradient-based adaptation capabilities with differential privacy; 2) it is not possible to perform expressive MIAs since the models do not output detailed token-probabilities/logits, and 3) we cannot access information about their training datasets, making it impossible to identify IID and OOD data for our analyses. The last point also discard all only open-weights models, such as Gemma2 (Team et al., 2024) or Llama (Grattafiori et al., 2024).

come from the same distribution as the pretraining data, and we identify two cases: one with a full overlap between pretraining and adaptation data, where we use data directly from the pretraining set for the adaptations, and one with no overlap, where the data is sourced from the corresponding validation set from the pretraining distribution. We focus on the following Pile subsets for the IID datasets: Bookcorpus2, GitHub, and Enron Emails (Klimt & Yang, 2004).

In contrast, OOD datasets are derived from a different distribution and do not overlap with pretraining data. Thus, we choose SAMSum (Gliwa et al., 2019), and GermanWiki (Cohere). To empirically quantify the distributional distances between our adaptation data and the respective adaptation datasets, we first compute sentence embeddings using Sentence-BERT (Reimers & Gurevych, 2019), then measure the Wasserstein distance (Villani, 2009; Arjovsky et al., 2017) between the pretraining data (all subsets of the Pile) and the respective adaptation dataset in Table 1. These results validate our dataset classifications: Pile-based datasets (Book-

Table 1: **Empirical quantification of dataset shift via the Wasserstein distance**.

| Dataset | Mean Distance | Min Distance |
|---|---|---|
| Bookcorpus2 Train | 0.0171 | 0.0019 |
| Bookcorpus2 Val | 0.0193 | 0.0057 |
| Github Val | 0.0180 | 0.0021 |
| Enron Val | 0.0202 | 0.0088 |
| SAMSum | 0.0250 | 0.0192 |
| GermanWiki | 0.0556 | 0.0520 |

corpus, GitHub, Enron) exhibit low distances, consistent with IID or overlapping status, whereas OOD datasets (SAMSum and GermanWiki) exhibit substantially higher distances. Notably, German-Wiki, which consists of German sentences rather than English ones like the other datasets, shows the greatest divergence, confirming its suitability as strongly OOD data for our evaluation. We elaborate more on the adaptation datasets in Section B.1.

**Adaptation Methods.** We evaluate different types of adaptations, including fine-tuning of all model parameters (Li et al., 2022), or the last layer (*i.e.,* the head), PEFT methods, such as LoRA (Hu et al., 2022; Yu et al., 2022) and Prefix Tuning (Liu et al., 2021; Duan et al., 2023a). Considering a Pythia 1B model, we train 1B parameters for Full Fine-Tuning, 1M for LoRA, 130M for Prefix Tuning, and 100M for last-layer (Head) Fine-Tuning. Since membership inference success is highly dependent on the train-test gap, for a fair comparison of the privacy leakage, we ensure similar evaluation perplexities, in particular, similar validation loss values at the end of the adaptation's training for specific datasets across adaptation methods, see Section E. More details on the adaptation setup are presented in Section B.2.

**Utility Analysis.** To assess the utility of our adapted models, in the main paper, we report perplexity and validation loss as proxies to ensure comparability across the wide range of datasets and adaptation settings studied. We additionally report the performance of generated content evaluated on the held-out set of each dataset with the Rouge-1 score and compare this with the perplexity values. The detailed results are displayed in Section E.2 and show that our proxies and the more fine-grained utility metric show the same privacy-utility trend across the different $\epsilon$ values. Finally, we also report the utility in loss on out-of-domain performance in Section E.3. These results show that, after DP fine-tuning, the adapted model improves on the adaptation task while showing no noticeable performance drop on the out-of-domain datasets.

**Membership Inference Attacks.** For MIA, we rely on the strongest state-of-the-art attack, RMIA (Robust Membership Inference Attack) (Zarifzadeh et al., 2024). We use the offline version because it is more efficient and does not need training customized reference models for each sample, unlike the online version. We leverage a single reference model for our experiments, as the authors show strong MIA performance even with a single reference model. Unless explicitly stated, we focus on using a "shadow" model (adaptation), in our case Pythia 1B, which is trained in the same way as the target model, but on a different split of the same fine-tuning data. We also evaluate the *Reference* method (Carlini et al., 2021), which calibrates the target model's loss using a reference model, and compare against Min-K% as a reference-less baseline attack. As with RMIA, we report the best AUC from a grid search over Min-K%'s parameter $K$. See Section B.4 for more details on the setup.

**Canary Exposure and Data Extraction Attacks.** To evaluate memorization, we insert adversarial canaries into a small portion of the adaptation data and estimate their exposure using two approximation methods: sampling and distribution modeling. Both approaches perform similarly when using 256 non-member canaries, and we adopt sampling for efficiency. Moreover, when considering $k$-extractable memorization, we set $k = 10$ tokens. See Section B.5 for extraction details.

Table 2: **Membership Inference for OOD Adaptations.** We audit only the adaptations and assume the same pretrained LLM is used for all adaptations. We present the AUC scores obtained with RMIA for the Pythia 1B model adapted on different datasets with $\varepsilon \in \{0.1, 8, \infty\}$.

| MIA | Dataset / Adapt. | SAMSum $\varepsilon = \infty$ | $\varepsilon = 8$ | $\varepsilon = 0.1$ | GermanWiki $\varepsilon = \infty$ | $\varepsilon = 8$ | $\varepsilon = 0.1$ | Average $\varepsilon = \infty$ | $\varepsilon = 8$ | $\varepsilon = 0.1$ |
|---|---|---|---|---|---|---|---|---|---|---|
| RMIA (shadow) | Prefix Tuning | 1.00 | 0.62 | 0.63 | 1.00 | 0.64 | 0.61 | 1.00 | 0.63 | 0.62 |
| | LoRA | 0.86 | 0.69 | 0.50 | 1.00 | 0.59 | 0.66 | 0.93 | 0.64 | 0.58 |
| | Full Fine-Tune | 1.00 | 0.82 | 0.62 | 1.00 | 0.71 | 0.55 | 1.00 | 0.77 | 0.59 |
| | Head Fine-Tune | 1.00 | 0.98 | 0.62 | 1.00 | 0.76 | 0.70 | 1.00 | 0.87 | 0.66 |
| | Average | 0.97 | 0.78 | 0.59 | 1.00 | 0.67 | 0.63 | 0.98 | 0.73 | 0.61 |
| Reference (Pythia 1B) | Prefix Tuning | 0.93 | 0.50 | 0.51 | 0.92 | 0.50 | 0.50 | 0.92 | 0.50 | 0.50 |
| | LoRA | 0.51 | 0.51 | 0.51 | 0.82 | 0.51 | 0.51 | 0.66 | 0.51 | 0.51 |
| | Full Fine-Tune | 0.94 | 0.51 | 0.51 | 0.99 | 0.51 | 0.50 | 0.96 | 0.51 | 0.51 |
| | Head Fine-Tune | 0.97 | 0.52 | 0.51 | 0.98 | 0.51 | 0.50 | 0.97 | 0.51 | 0.50 |
| | Average | 0.84 | 0.51 | 0.51 | 0.93 | 0.51 | 0.50 | 0.88 | 0.51 | 0.51 |

Table 3: **Membership Inference for in-distribution (IID) Adaptations** using the setup from Table 2.

| MIA | Dataset / Adapt. | Bookcorpus2 Val $\varepsilon=\infty$ | $\varepsilon=8$ | $\varepsilon=0.1$ | Bookcorpus2 Train $\varepsilon=\infty$ | $\varepsilon=8$ | $\varepsilon=0.1$ | Github Val $\varepsilon=\infty$ | $\varepsilon=8$ | $\varepsilon=0.1$ | Enron Val $\varepsilon=\infty$ | $\varepsilon=8$ | $\varepsilon=0.1$ | Average $\varepsilon=\infty$ | $\varepsilon=8$ | $\varepsilon=0.1$ |
|---|---|---|---|---|---|---|---|---|---|---|---|---|---|---|---|---|
| RMIA (shadow) | Prefix Tuning | 1.00 | 0.89 | 0.56 | 1.00 | 0.90 | 0.55 | 1.00 | 0.93 | 0.63 | 1.00 | 0.88 | 0.58 | 1.00 | 0.90 | 0.58 |
| | LoRA | 1.00 | 0.70 | 0.52 | 1.00 | 0.69 | 0.53 | 1.00 | 0.74 | 0.52 | 1.00 | 0.73 | 0.52 | 1.00 | 0.71 | 0.52 |
| | Full Fine-Tune | 1.00 | 0.75 | 0.77 | 1.00 | 0.75 | 0.76 | 1.00 | 0.78 | 0.80 | 1.00 | 0.91 | 0.66 | 1.00 | 0.80 | 0.75 |
| | Head Fine-Tune | 1.00 | 0.72 | 0.73 | 1.00 | 0.72 | 0.72 | 1.00 | 0.80 | 0.74 | 1.00 | 0.57 | 0.65 | 1.00 | 0.70 | 0.71 |
| | Average | 1.00 | 0.77 | 0.65 | 1.00 | 0.76 | 0.64 | 1.00 | 0.81 | 0.67 | 1.00 | 0.77 | 0.60 | 1.00 | 0.78 | 0.64 |
| Reference (Pythia 1B) | Prefix Tuning | 0.93 | 0.56 | 0.52 | 0.97 | 0.57 | 0.50 | 0.97 | 0.53 | 0.51 | 0.97 | 0.54 | 0.50 | 0.96 | 0.55 | 0.51 |
| | LoRA | 0.89 | 0.52 | 0.52 | 0.97 | 0.51 | 0.51 | 0.92 | 0.51 | 0.50 | 0.97 | 0.55 | 0.51 | 0.94 | 0.52 | 0.51 |
| | Full Fine-Tune | 1.00 | 0.54 | 0.52 | 1.00 | 0.54 | 0.52 | 0.99 | 0.54 | 0.52 | 0.98 | 0.59 | 0.50 | 0.99 | 0.55 | 0.51 |
| | Head Fine-Tune | 0.98 | 0.57 | 0.52 | 1.00 | 0.56 | 0.51 | 0.99 | 0.66 | 0.50 | 0.99 | 0.54 | 0.50 | 0.99 | 0.58 | 0.51 |
| | Average | 0.95 | 0.55 | 0.52 | 0.98 | 0.55 | 0.51 | 0.97 | 0.56 | 0.51 | 0.98 | 0.55 | 0.50 | 0.97 | 0.55 | 0.51 |

## 4 BENCHMARK DESIGN AND EXPERIMENTS

To address our benchmark's central question: *"What are the empirical privacy risks to adaptation data under DP adaptations?"*, we break it down into six concrete research questions.

### 4.1 RQ1: HOW DOES THE RELATIONSHIP (OVERLAPPING, IID, OOD) BETWEEN ADAPTATION AND PRETRAINING DATASETS IMPACT DATA PRIVACY?

**Motivation.** The pretrain-adapt paradigm uses LLMs pretrained on large public datasets, which are then adapted to smaller, often sensitive, private datasets using DP methods. While DP offers formal guarantees, its practical effectiveness under the pretrain-adapt paradigm remains unclear—particularly how the relationship and interplay between adaptation and pretraining data (*e.g.,* overlapping, IID, or OOD) influences actual privacy leakage.

**Summary of Findings.** Our results show that (1) privacy risks increase when the adaptation data distribution is closer to the pretraining data, even if there is no direct overlap. (2) Surprisingly, IID data from the pretraining validation set leaks as much as directly overlapping data, underscoring distributional closeness as the main driver of risk.

**Detailed Results.** We present our main results in Table 2, and Table 3. We focus our discussion on Pythia-1B, and further expand the discussion for the other models in Section C.1. They show that the average AUC is generally higher in IID settings than in OOD in all attacks and adaptations. For instance, looking at *RMIA (shadow)* using $\varepsilon = 8$, we observe that the average AUC is between 0.7 and 0.9 in the IID setting, while it is between 0.63 and 0.87 for the OOD setting. More detailed analyses for different attack setups and more privacy regimes are depicted in Section C.1. We also identify distributional closeness as a key risk factor, as overlapping data leaks similarly to IID. Moreover, our results indicate that under both a strong attack and in more practical scenarios, moderate privacy regimes (*e.g.,* $\varepsilon = 8$) still present a real threat of privacy leakage from IID. On the other hand, under this regime, privacy leakage in the OOD setting is mostly observed with a strong attack. Moreover, Section C.4, Figure 9 shows over the training epochs the Overlap (Train) and IID data (Val) privacy leakage, and further highlights a similar privacy leakage between Overlap and IID data across the whole training run. We also analyze the impact of subset characteristics on privacy leakage in Section C.3, and we discover that the pretraining dataset size and complexity influence the privacy leakage in the training datasets. We observe that privacy leakage increases with both the size and complexity of the subsets.

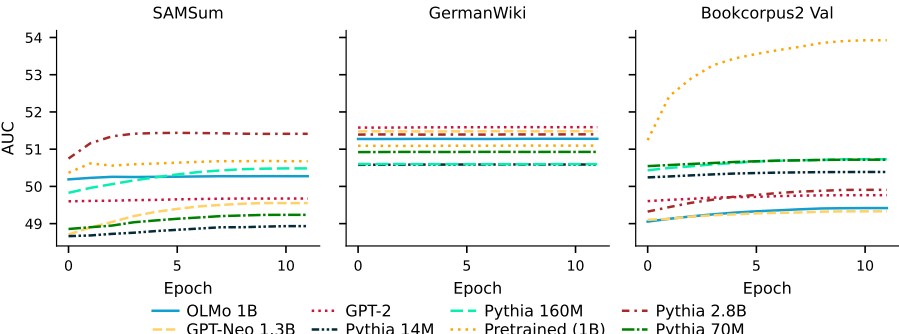

Figure 2: **IID data is more susceptible to leakage using the pretrained base model than OOD data.** We compare the effectiveness of performing RMIA on fully fine-tuned Pythia 1B with $\varepsilon = 8$ with different pretrained models as reference models.

## 4.2 RQ2: WHICH DP ADAPTATION METHOD IS THE MOST PROTECTIVE?

**Motivation.** It is known that the type of adaptation significantly impacts the utility of the final model (Zhu et al., 2024). However, different adaptations might also offer disparate empirical protection at the same formal privacy guarantee, motivating our empirical comparison.

**Summary of Findings.** While LoRA provides much better empirical privacy protection in non-private settings compared to other adaptations, the differences become more subtle under the DP regime. Despite this, LoRA consistently achieves a relatively low AUC, whereas the other adaptations show varying trends depending on the dataset or privacy budget.

**Detailed Results.** Specifically, as shown in Table 2 for OOD datasets with $\varepsilon = 8$, the most vulnerable adaptations are Full and Head Fine-Tune. On the other hand, for IID data, the strongest protection is provided by Head Fine-Tune, which is marginally better than LoRA. With stronger privacy guarantees, LoRA is the most private for OOD datasets with an AUC score of 0.58, thus slightly better than Full Fine-Tune. On the other hand, while adapting to the IID dataset, LoRA outperforms other adaptations. Notably, Full Fine-Tune and Head Fine-Tune show much lower privacy protection in these settings.

## 4.3 RQ3: ARE THE SAME ADAPTATIONS ROBUST AGAINST DATA EXTRACTION?

**Motivation.** Data extraction attacks are even more severe than MIAs. Therefore, it is crucial to evaluate the protectiveness of DP adaptations against this stronger threat.

**Summary of Findings.** We find that Prefix Tuning is the most vulnerable adaptation method in this setting. On the other hand, LoRA and Head Fine-Tune in both cases, with and without DP guarantees, exhibit resistance against data extraction.

**Detailed Results.** We report detailed results in Section C.2. In particular, Table 22 and Table 23 show that for $\varepsilon = 0.1$ the exposure is around 1.44, therefore, close to random guessing. We also noticed a limited influence on the choice of the canary prefix type. Moreover, the adversarial prefix is the main source of privacy leakage, with the interaction between the prefix and the individual sample playing a smaller role, see Figure 10 in Section C.5.

## 4.4 RQ4: HOW IMPORTANT IS THE ATTACKER'S KNOWLEDGE OF THE PRETRAINED MODEL?

**Motivation.** The attacker's knowledge of the pretrained model plays a crucial role in the success of MIAs, as it enables them to select more relevant reference models and non-member data for training, which is one of the main challenges of MIAs (Watson et al., 2022; Carlini et al., 2022). We investigate various setups, including an attacker who has access to a shadow model from the same pretraining distribution as the adapted LLM, another model sharing common traits (such as the same data but different architecture or size), and no access to external models. This helps us characterize the landscape of potential real-world risks and setups.

**Summary of Findings.** The performance of MIAs highly depends on the attacker's knowledge of the target model and pretraining data. In particular, RMIA performs best when a shadow model shares

architecture, initialization weights, and training data distribution. Meanwhile, RMIA's effectiveness rapidly deteriorates as shadow models are trained on different distributions or architectures. Particularly, we observe that when a shadow model trained on the same distribution of the target model is not available, using the pretrained model is the second-best choice, followed by models of the same family and similar size.

**Detailed Results.** To simulate attackers with various background knowledge, in this setting, we also consider other "shadow" models: Pythia 14M, Pythia 160M, Pythia 1B, Pythia 2.8B (Biderman et al., 2023), GPT-Neox (Black et al., 2021), OLMo 1B (Groeneveld et al., 2024a), and GPT-2 (Radford et al., 2019). The MIA performance is close to random for private adaptations with $\varepsilon = 8$. Furthermore, as shown in Figure 2, while the MIA's performance for Pythia 1B is higher on IID data, the choice of reference model has little effect when attacking models adapted on OOD data, even with architectural differences between the model and the reference model *i.e.,* GPT-Neo 1.3B and OLMo 1B. Moreover, as in the other case, Figure 12 (in Section D) shows that the privacy leakage is similar between IID and the corresponding overlapping data. We show further experiments in Section D.

### 4.5 RQ5: How does adaptation change the pretraining dataset vulnerability?

**Motivation.** DP adaptations only guarantee protection for the adaptation dataset. Yet, adapting the model to other data, while introducing noise, can also affect the pretraining leakage. This is an important aspect to study, as also pretraining data can be private (Tramèr et al., 2024), *e.g.,* conversations with ChatGPT used to improve the models, or emails used to pretrain Gemini. Therefore, we also empirically investigate how adapting pretrained LLMs affects the leakage of pretraining data.

**Summary of Findings.** Our findings show that the choice of adaptation method impacts the privacy of pretraining data. Our evaluation shows that Prefix Tuning reduces the leakage of memorized pretraining data from adapted language models, especially in high-privacy settings. For the other adaptations, this effect is negligible, and the adapted model retains most of the pretraining memorization.

**Detailed Results.** We evaluate the effect of OOD and IID adaptation data on the leakage of memorized pretraining data from the adapted LLM. Specifically, as we show in Figure 3, Prefix Tuning significantly reduces leakage, particularly in high-privacy regimes. For the other adaptation methods, the number of memorized samples often remains above 460 samples. For Prefix Tuning, the number of memorized samples is often lower than 460 and goes down to around 430 with $\varepsilon = 0.1$, thus suggesting that adaptation partially mitigates the pretraining memorization.

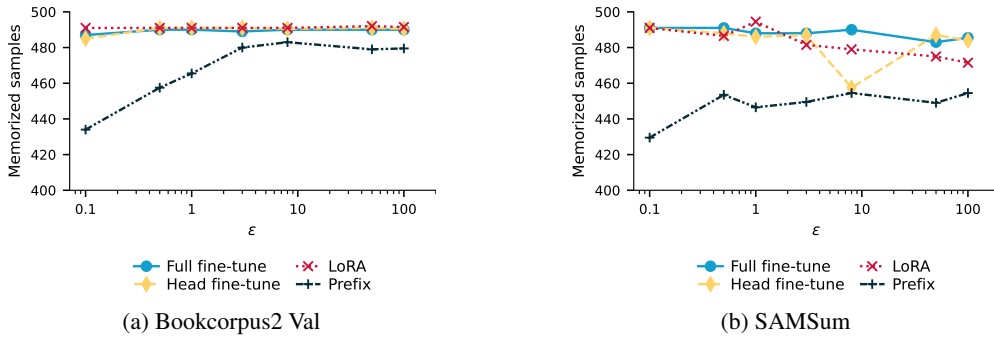

(a) Bookcorpus2 Val
(b) SAMSum

Figure 3: **Prefix tuning reduces the number of verbatim memorized samples, especially for small $\varepsilon$ values.** We show the result for Pythia 1B adapted on Bookcorpus2 val and SAMSum datasets with $\varepsilon = \{0.1, 1, 3, 8, 50, 100, \infty\}$. We present the x-axis using a log scale.

### 4.6 RQ6: How do Privacy-Utility Trade-Offs Behave?

**Motivation.** We also analyze the different empirical privacy-utility trade-offs that can be achieved by the various adaptation methods under the same theoretical privacy budget. This can guide practitioners in answering the question: *"For a given private dataset, model, and adaptation method, if I use it, what kind of privacy risks should I expect?"* Concretely, we run a hyperparameter grid search and report the resulting privacy-utility outcomes for Pythia-1B, showing the top four runs for each adaptation method, dataset, and privacy budget.

**Summary of Findings.** Our experiments show the hyperparameter choice can lead to varying levels of utility and privacy risks. Generally, LoRA consistently offers the best privacy-utility trade-offs. Specifically, at the same utility level, as measured by perplexity, it exhibits lower privacy risks, as indicated by the RMIA AUC.

**Detailed Results.** The analysis provided in Figure 4 illustrates these differences. For instance, on GermanWiki with $\varepsilon = 8$, at perplexity 14.27, LoRA exhibits an AUC of 0.77, whereas Full Fine-Tuning at perplexity 14.60 has 0.82. Similarly, for GitHub (val), at perplexity 4.8, LoRA exhibits an AUC of 0.6, whereas Full Fine-Tuning at the same perplexity has 0.83. These findings align with the trends observed in the other experiments. These concrete examples highlight the consistency of LoRA's advantages in reducing privacy risks, while maintaining performance. To further validate our utility results, we evaluated the adapted models on generative tasks based on the Rouge-1 score. Detailed comparison between perplexity and Rouge-1 score are shown in Section E.2.

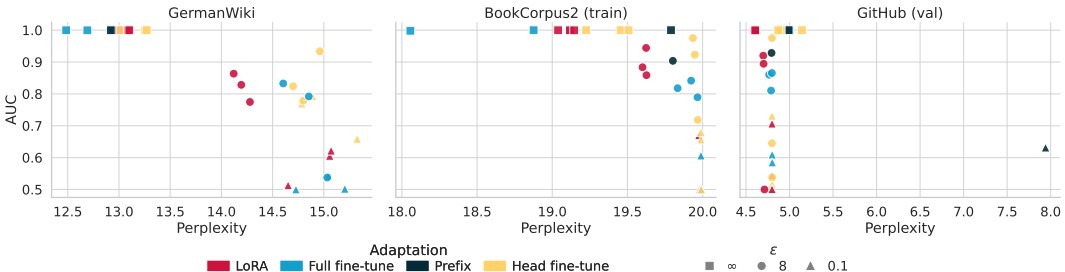

Figure 4: **Privacy-utility curves for the top perplexity-selected runs from the Pythia-1B hyperparameter search, shown for the chosen adaptation method, dataset, and privacy budget.**

## 5 DISCUSSION OF OUR RESULTS

Our findings reveal a complex interplay between pretraining and adaptation data. This significantly affects the privacy risks under DP adaptations. Below, we discuss the implications of these findings when adapting pretrained LLMs to sensitive domains using DP.

**Disparate Leakage Based on Distribution.** Our results demonstrate that the distributional closeness between pretraining and adaptation data is a key factor influencing empirical privacy leakage under DP. Adaptations using data from the same distribution but not seen during pretraining consistently show the highest vulnerability. This presents the fundamental trade-off: while adapting a model already pretrained on similar data is beneficial for utility, it simultaneously increases privacy risk.

**Disparate Leakage Based on Adaptation Method.** We also observe that not all DP adaptation methods offer equal protection, even when enforcing the same formal level guarantee, expressed in the same $\varepsilon$. This aligns with earlier findings in the non-private regime, where privacy-utility trade-offs differ across methods (Zhu et al., 2024). In our experiments, LoRA appeared most consistently robust against privacy attacks, while Prefix Tuning showed the least vulnerability to extraction attacks. These differences are highly relevant for practice: in addition to choosing methods that optimize downstream performance, practitioners should also consider empirical privacy leakage. The attacks we use in this paper offer a way to assess and understand such risks under realistic conditions.

**Choosing a Privacy Regime.** We find that in moderate privacy regimes, *e.g.,* $\varepsilon = 8$, sensitive adaptation data still experiences significant practical vulnerability against both MIAs and data extraction attacks. This highlights the necessity to perform private LLM adaptations in the high-privacy regime, *i.e.,* with low $\varepsilon$ to achieve practical protection.

**Reliance on Accurate Shadow Model.** We show that attackers gain a substantial advantage when they have access to the original pretrained LLM used during adaptation. Shadow models instantiated with the same pretrained model as the adapted LLM's base consistently achieved higher attack success. This is especially concerning given the rise of adapting publicly available LLMs, which makes strong shadow models easily accessible to adversaries. These findings further underscore the need for stringent privacy settings in DP adaptations.

**Towards a Holistic Privacy Auditing for LLMs.** Our results suggest that privacy assessments should not treat pretraining and adaptation in isolation. The strong interdependence between these

stages demands holistic analysis. Motivated by this insight, we introduce a structured framework in the next section that formalizes how privacy assessments and audits under the pretrain-adapt paradigm should be conducted. We hope this framework encourages the development of privacy assessment methods that match the complexity of modern private LLM pipelines.

# 6 HOLISTIC PRIVACY AUDITS UNDER THE PRETRAIN-ADAPT PARADIGM

## 6.1 AUDIT STAGES AND ADVERSARY GAMES FOR PRETRAIN-ADAPT PRIVACY AUDITING

While our understanding of empirical privacy risks has grown, more nuanced approaches are needed to address privacy risks in LLM adaptation. We therefore formalize a holistic privacy auditing framework for the pretrain-adapt paradigm with four stages (see Figure 5): (1) auditing pretraining, (2) auditing adaptation, (3) joint audit of pretraining and adaptations, and (4) post-adaptation auditing of the pretraining. Based on these stages, we instantiate the audits and contrast them with standard privacy auditing, modeled as an *adversarial game* $\mathcal{G}$ (Yeom et al., 2018; Jayaraman et al., 2020) where the task is to guess whether a data point $x$ was in a model's training set.

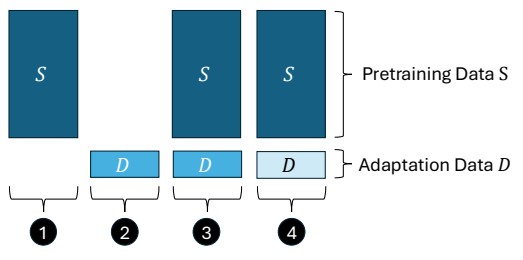

Figure 5: **Stages of Auditing.** We analyze four stages of auditing: ❶ Auditing Pretraining, ❷ Auditing Adaptation, ❸ Joint Auditing of Pretraining and Adaptations, ❹ Post-Adaptation Auditing of the Pretraining.

Figure 6: **Setup for Joint Adaptation Auditing (3).** We consider different datasets for pretraining and adaptation and the two separate training stages, distinguishing it from standard ML privacy auditing.

We define the adversarial game $\mathcal{G}$ analogous to the one for standard ML, yet take two datasets, $S$ the pretraining data, and $D$ the adaptation data into account. Additionally, we denote the pretraining procedure by $T$ and the adaptation by $T'$. Deviations to the original game are marked in blue.

1. The challenger samples $a \xleftarrow{\text{R}} \{0,1\}$ and $b \xleftarrow{\text{R}} \{0,1\}$ (where $a$ and $b$ are binary variables)

2. The challenger trains a model $\theta \xleftarrow{\text{T}} \tilde{S}, \theta_0$, where $\tilde{S} = S$ if $a = 0$, otherwise $\tilde{S} = S \cup \{x\}$

3. The challenger adapts $\theta$ such that $\theta' \xleftarrow{\text{T'}} \tilde{D}$, where $\tilde{D} = D$ if $b = 0$, otherwise $\tilde{D} = D \cup \{x\}$

4. The challenger sends $\theta'$ to the attacker

5. The attacker guesses $\hat{a}, \hat{b} \leftarrow \mathcal{A}(\theta, \theta', x)$

Whether the attacker has to guess both $\hat{a}, \hat{b}$ and what background knowledge they have, *i.e.,* whether they get access to both $\theta$ and $\theta'$, depends on the auditing stage. We detail the attacker's background knowledge and guesses, formulated as hypotheses with a null hypothesis $H_0$ and an alternative hypothesis $H_A$, for the four auditing stages from our taxonomy.

**(1) Auditing pretraining** resembles standard ML auditing, targeting privacy leakage from pretrained models. Differences arise from larger datasets and models, limiting both DP protection efficacy (Carlini et al., 2023b) and applicability of auditing techniques like MIA (Duan et al., 2024). In this setting, the challenger releases the pretrained model $\theta$ to the attacker. The attacker's goal is to correctly guess whether $x$ was in the pretraining data $S$. Their guesses $\hat{a}$, are over the random variable $a$.

$$H_0 : a = 0 \qquad H_A : a = 1$$

**(2) Auditing adaptation** detects leakage of the adaptation dataset from adapted LLMs in the pretrain-adapt paradigm. Unlike standard ML audits, adaptations start from a pretrained model rather than a random initialization; we assume the same pretrained model for all considered adaptations. The challenger releases only the adapted model $\theta'$ to the attacker. The attacker does not know whether $x \in S$ and therefore guesses $\hat{b}$ over the random variable $b$.

$$H_0 : b = 0 \qquad H_A : b = 1$$

**(3) Joint Auditing** evaluates combined leakage from both pretraining and adaptation datasets in the adapted LLM. Typical privacy-preserving approaches involve non-DP-trained LLMs with DP-trained adaptations. In this setting, the challenger releases both the pretrained model $\theta$ and the adapted $\theta'$ to the attacker. Depending on the attacker's background knowledge, we consider three possible cases.

| The attacker knows that $x \notin S$ and guesses $b$. | The attacker knows that $x \in S$ and guesses $b$. | The attacker knows that the target sample $x$ is either in both (pretraining and adaptation sets) or neither of them and guesses $(a, b)$. |
|---|---|---|
| $H_0 : (a, b) = (0, 0) \qquad H_A : (a, b) = (0, 1)$ | $H_0 : (a, b) = (1, 0) \qquad H_A : (a, b) = (1, 1)$ | $H_0 : (a, b) = (0, 0) \qquad H_A : (a, b) = (1, 1)$ |

**(4) Post-Adaptation Auditing** evaluates how (private) adaptations affect the protection of data points used for pretraining, which typically lacks formal guarantees. Changes in model behavior induced by the adaptation, including training noise, may alter the exposure of pretraining data in model predictions. In this setting, the challenger releases both the pretrained $\theta$ and the adapted $\theta'$. It is known that the target sample $x \notin D$ and the attacker guesses $a$.

$$H_0 : (a, b) = (0, 0) \qquad H_A : (a, b) = (1, 0)$$

In essence, auditing pretraining considers only the pretraining itself. Similarly, auditing the adaptations considers the adaptations themselves. On the other hand, the joint adaptation reasons about both pretraining and adaptation sets. Finally, the post-adaptation auditing is only for the pretraining set, but the applied adaptation influences the auditing.

## 6.2 PRACTICAL APPLICATION OF HOLISTIC AUDITS

Our new perspective on the pretrain-adapt paradigm gives both practitioners and researchers clearer insights into each threat model at its particular stage. Formalizing the auditing setup supports systematic reasoning about privacy risks, thereby clarifying the guarantees that different methods need to provide. Therefore, our formalization enables the creation of a unified interface for measuring privacy leakage, regardless of whether its source is pretraining or adaptation data. Moreover, our work demonstrates that looking at pretraining and adaptation components separately can lead to a misleading impression of privacy. The connection between these stages affects privacy leakage, which makes comprehensive auditing essential within pretrain-adapt paradigm. We believe that developing and sharing tools that support all stages of privacy assessment, from threat modeling and risk quantification to mitigation, will empower the research community to define risks and reduce privacy risks in practice more effectively.

## 7 CONCLUSIONS

In this work, we benchmark the practical privacy risks that arise under DP adaptations of LLMs within the pretrain-adapt paradigm. Our comprehensive empirical analysis confirms the theoretical concern that pretraining significantly amplifies the privacy risks associated with the *adaptation data*. We find that the closeness of adaptation and pretraining data distributions plays a critical role: even in the absence of overlap, higher distributional similarity results in increased privacy leakage. Additionally, we observe that the choice of adaptation method impacts privacy leakage, with PEFT methods, such as LoRA, offering significantly lower privacy risks while maintaining strong utility. Furthermore, we show Prefix Tuning can reduce the leakage of pretraining data, likely due to the added input noise during private adaptation. Our findings highlight the need for stringent DP constraints (*e.g.*, $\varepsilon < 0.1$) to mitigate privacy risks in LLM adaptations effectively. It also motivates the need for holistic privacy assessments under the pretrain-adapt paradigm and takes the first step towards it by formalizing such an assessment over the different stages. This work lays a foundational framework for future research efforts aimed at safeguarding privacy within the pretrain-adapt paradigm.

ACKNOWLEDGEMENTS

Franziska Boenisch received funding from the European Research Council (ERC) under the European Union's Horizon Europe research and innovation programme (grant agreement No 101220235). The project was also supported by the German Federal Ministry of Research, Technology and Space (BMFTR) under funding number 16KIS2114K. Additionally, we would like to acknowledge our sponsors, who support our research with financial and in-kind contributions: OpenAI and G-Research. We also thank members of the SprintML group for their feedback. Responsibility for the content of this publication lies with the authors.

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

# A BACKGROUND

## A.1 PRIVATE LLM ADAPTATIONS

Differentially Private Stochastic Gradient Descent (DP-SGD) (Abadi et al., 2016) is a widely used method for incorporating DP into deep learning. However, while applied to NLP tasks, DP-SGD can exhibit several limitations, particularly in model utility, increased memory usage, or slower convergence during training. These limitations motivate the exploration of alternative DP adaptation techniques.

**Full DP Fine-Tuning.** One approach to differentially private (DP) adaptation is to fine-tune the entire model using the DP-SGD algorithm (Abadi et al., 2016; Li et al., 2022; Yu et al., 2022). This method updates all model parameters while ensuring that each gradient step satisfies DP guarantees through gradient clipping and noise addition. Full-model DP fine-tuning provides high adaptability and task-specific performance. However, it is computationally expensive and memory-intensive, especially for large language models (LLMs), due to the need to compute, clip, and perturb gradients for all layers (Li et al., 2022).

**DP Head Fine-Tuning.** An alternative strategy is to fine-tune only the final layer (often called the classification or task-specific "head") of the model using DP-SGD. This significantly reduces the number of trainable parameters, leading to lower memory usage and faster training. Despite its simplicity, DP Head Fine-Tuning can still achieve competitive performance on certain tasks while providing formal privacy guarantees. However, its adaptability is limited, particularly when deeper model layers need task-specific adjustments.

**DP low-rank adaptation (LoRA).** LoRA (Hu et al., 2022) is an efficient technique for adapting LLMs that introduces low-rank matrices into each layer of a frozen pretrained model. Instead of updating the full weight matrix $W \in \mathbb{R}^{d \times k}$, LoRA learns a low-rank approximation $\Delta W = AB$, where $A \in \mathbb{R}^{d \times r}$, $B \in \mathbb{R}^{r \times k}$, and $r \ll \min(d, k)$. The adapted weights become $W' = W + AB$, with only $A$ and $B$ being trainable. DP LoRA (Yu et al., 2021) extends this approach by applying DP-SGD to the low-rank parameters. This ensures that the adaptation remains privacy-preserving, making LoRA suitable for sensitive-data applications with formal DP guarantees.

**DP Prompting.** Introducing a small set of additional parameters, typically under 1% of the LLM's total parameters, DP Prompting applies these only within the model's input space. These parameters may be added at the level of token embeddings (soft prompts (Liu et al., 2021; 2022)) or to all (attention) layers of the LLM (prefix-tuning (Lester et al., 2021; Li & Liang, 2021)). Duan et al. (2023a) proposed *PromptDPSGD*, which adapts the DP-SGD algorithm (Abadi et al., 2016) for use with soft prompts.

## A.2 MIAS

The following section provides a more detailed description of MIAs used in our benchmark.

**Min-K%.** Min-K% (Shi et al., 2024a) is a recently proposed black-box MIA for large language models. The intuition is that an unseen sample is likely to have low-probability tokens. The MIA score is defined as

$$\texttt{Min-K\%}(x) = \frac{1}{|S|} \sum_{x_i \in S} \log p(x_i | x_1, ..., x_{i-1}), \tag{3}$$

where S is the set of K% tokens with the smallest loss.

**Reference.** This approach (Carlini et al., 2021) uses a reference model to calibrate the MI score as follows

$$\texttt{Ref}(x) = \frac{\mathcal{L}(x|\theta)}{\mathcal{L}(x|\theta_{\text{ref}})}, \tag{4}$$

where $\mathcal{L}(x|\theta)$ indicates the loss of the target sample $x$ on the model $\theta$. $\theta_{\text{ref}}$ represents the reference model used.

**Robust Membership inference attack (RMIA).** RMIA outperforms previous methods by optimizing the null hypothesis and using a reference model along with population data, requiring only one reference (*shadow*) model at a time, unlike previous methods (Carlini et al., 2022; Rossi et al., 2025a; Pawelczyk et al., 2023) which required hundreds. RMIA has two hyperparameters, a threshold $\gamma$ and a scaling factor $\alpha$. The adapted RMIA score (Equation (5)) calculation for LLMs for text generation is based on comparing loss values rather than output probabilities. For this reason, we have to, instead of comparing prediction probabilities or logits, compare the loss of the target data point against the loss of reference models on population data (Equation (6)) and flip to a minority voting approach, where the decision is based on how much lower the loss of the target data is compared to the population data.

$$\text{Score}_{\text{MIA}}(x; \theta) = \Pr_{z \sim \pi} \left( \text{LR}_\theta(x, z) \geq \gamma \right) \tag{5}$$

$$\text{LR}_\theta(x, z) = \mathcal{L}(\theta|x) - \mathcal{L}(\theta|z) \tag{6}$$

### A.3 Canary Exposure and Data Extraction Attacks

Following Carlini et al. (2019); Tramèr et al. (2022), let $\mathcal{U}$ be the universe of candidate samples and let $\hat{Z}$ be the attacker's ranking of $\mathcal{U}$ by model-assigned likelihood. For a target $z \in \mathcal{U}$,

$$\text{exposure}(z, \hat{Z}) := \log_2 |\mathcal{U}| - \log_2 \left( \text{rank}(z; \hat{Z}) \right). \tag{7}$$

This metric ranges from 0 (least likely) to $\log_2 |\mathcal{U}|$ (most likely). To compute it efficiently when $|\mathcal{U}|$ is large, one can use: (1) **sampling**, which estimates exposure on a random subset of $\mathcal{U}$, or (2) **distribution modeling**, which approximates the distribution of model scores (e.g. via a skewed normal) to interpolate ranks. The expected exposure of an unmemorized canary is $\frac{1}{\ln 2} \approx 1.44$ (Jagielski, 2023). Complementing exposure-based metrics, Carlini et al. (2023a) introduce a contextual extraction framework to assess memorization and data extraction attacks. Let $f$ be a generative model and $s$ a secret suffix. We say $s$ is *extractable with $k$ tokens of context* if there exists a prefix $p$ of length $k$ such that, under greedy decoding,

$$f(p) = [p \,\|\, s].$$

When $s$ is long and random, its successful extraction indicates memorization. One can vary $k$ to characterize how much context the model needs before regurgitating $s$ verbatim.

## B Additional Details on the Setup

### B.1 Datasets

For the IID datasets, we focus on the following Pile subsets: Bookcorpus2, consisting of publicly available books, GitHub, a set of open-source code repositories, and Enron Emails (Klimt & Yang, 2004), a collection of various emails. The OOD datasets we choose for our experiments are: SAMSum (Gliwa et al., 2019), an English-language dialogue summarization dataset, and GermanWiki (Cohere), a large set of German Wikipedia entries. These OOD datasets were selected because of their different degrees of variation from the original distribution of the Pile dataset. Although SAMSum shares the same language (English), its general dialogue format, followed by the dialogue summary, is not present in the pretraining set. GermanWiki, on the other hand, presents wide syntactic and lexical variation from the pretraining dataset.

### B.2 Adaptations

We focus on four types of adaptations: Prefix Tuning, LoRA, Full Fine-Tune, and Head Fine-Tune. We train all the models using Adam with the privatization gradient method of DP-SGD (Abadi et al., 2016). For the Adam optimizer, we use the default HuggingFace hyperparameters except for the learning rate. For Prefix Tuning, we fix a prefix length of 64, while for LoRA, a rank $r = 8$ and $\alpha = 16$. For DP-SGD, following existing work (Li et al., 2022), we set the gradient clipping value to 0.1. Moreover, in all settings, we consider sentence-level DP, meaning that we concatenate all strings in the dataset and split them into 256 token chunks, corresponding to sentence-level privacy.

### B.3 HYPERPARAMETERS

For each task, model, and privacy budget, we performed a hyperparameter optimization using a random search strategy. Specifically, we explored the following ranges:

- Learning Rate: $1 \times 10^{-6}, 3 \times 10^{-6}, 1 \times 10^{-5}, 3 \times 10^{-5}, 5 \times 10^{-5}, 8 \times 10^{-5}, 1 \times 10^{-4}, 3 \times 10^{-4}, 1 \times 10^{-3}, 5 \times 10^{-3}$;
- Number of training epochs: $1, 2, 3, 5, 10, 15, 16, 20, 30, 32$;
- Batch size: $4, 8, 16, 32, 64$;

Our objective during hyperparameter search is to ensure comparable evaluation perplexities, specifically targeting similar validation loss values after adaptation training across different methods for specific datasets.

### B.4 MIA

The adopted offline mode (see Algorithm 1) shrinks from the need to retrain reference models per query, thus relying on pretrained LLMs, which are computationally expensive to train. For most experiments, we used just one reference model ($k = 1$), thus demonstrating the power of RMIA attack and highlighting data leakage, especially from pretrained data. For an ablation on the RMIA hyperparameters choice, see Figure 14 in Section H.

---

**Algorithm 1** MIA score calculation with offline RMIA (Zarifzadeh et al., 2024) adapted to LLMs.

**Input:** $k$ reference models $\Theta$, target sample $x$, threshold $\gamma$, scaling factor $\alpha$, population dataset $\pi$,
**Output:** $\text{Score}_{\text{MIA}}(x; \theta)$

1: Randomly choose a subset $Z$ from the population dataset
2: $C \leftarrow 0$
3: $\mathcal{L}(x)_{\text{OUT}} \leftarrow \frac{1}{k} \sum_{\theta' \in \Theta} \mathcal{L}(x|\theta')$
4: $\mathcal{L}(x) \leftarrow \frac{1}{2} \left( (1 + \alpha)\mathcal{L}(x)_{\text{OUT}} + (1 - \alpha) \right)$
5: $\text{Ratio}_x \leftarrow \frac{\mathcal{L}(x|\theta)}{\mathcal{L}(x)}$
6: **for** each sample $z$ in $Z$ **do**
7: $\quad \mathcal{L}(z) \leftarrow \frac{1}{k} \sum_{\theta' \in \Theta} \mathcal{L}(z|\theta')$
8: $\quad \text{Ratio}_z \leftarrow \frac{\mathcal{L}(z|\theta)}{\mathcal{L}(z)}$
9: $\quad$ **if** $\text{Ratio}_x / \text{Ratio}_z < \gamma$ **then**
10: $\quad \quad C \leftarrow C + 1$
11: $\quad$ **end if**
12: **end for**
13: **return** $\text{Score}_{\text{MIA}}(x; \theta) \leftarrow \frac{C}{|Z|}$

---

### B.5 CANARY EXPOSURE

We add an adversarial prefix to $p = 1\%$ of the adaptation data. If not specified otherwise, we set the number of canary tokens to $k = 10$ and the canary prefix length $l = 10$. To measure exposure, we generate 256 new canary prefixes from the same canary type and prepend them to the target sample $x$ whose exposure we want to measure. The resulting 256 samples can be considered as a form of non-members. On expectation, all canary prefixes are equally (un)likely. However, if the model is more confident about the one prefix it saw during adaptation than it is about the other 256 prefixes, it means that the model must have memorized this prefix and that it was part of the adaptation data. Given that there are two ways of approximating exposure (sampling and distribution modeling) as discussed in Section 2, we assess both of them to find whether one approach is more suitable. This ablation in Figure 13, Section F shows that the two approximations perform similarly when using 256 non-member canaries. In our experiments, we evaluated using *sampling* as an approximation since it is computationally cheaper.

**Canary Types.** The *random* canary prefix is the simplest type of canary prefix, and it is composed of completely random tokens sampled uniformly from the token universe $T$. The *common* and *rare*

prefixes comprise the most and least frequently occurring tokens, respectively, excluding special tokens *e.g.,* padding and end-of-string tokens. We count the total number of token occurrences in the adaptation dataset to measure the frequencies. Then, we choose the top $k$ tokens from a list sorted in descending order for *common* tokens and ascending order for *rare* tokens. Note that, for both *common* and *rare*, each adaptation dataset naturally has its own set of distinct prefix tokens. We also select the *random* tokens independently over each adaptation dataset for symmetry. The *invisible* canary prefix utilizes imperceptible Unicode symbols or space-like tokens, such as zero-width spaces or zero-width non-joiners, which are nearly undetectable by humans, thus incorporating the design approach known from other adversarial attacks (Boucher et al., 2021). Compared to the other canary types, the set of tokens is the same for each dataset. Again, we randomly sample $k$ imperceptible symbols to prepend as a canary prefix.

**Canary Adaptation Set Generation.** Algorithm 2 describes the procedure to construct the adaptation dataset with canary prefixes. Note that `concat(a,b)` concatenates two strings, and the tokens universe $T$ represents the set of all tokens accepted by the LLM. We prepend the canaries to a small fraction $p$ of the adaptation dataset prior to performing the adaptation. To each selected sample, we add $l$ tokens, randomly drawn with replacement from the respective $k$ canaries in the canary prefix sets. We do not combine tokens from our four different types of canary prefixes and consider each separately.

---

**Algorithm 2** Adding canary prefixes to the adaptation dataset.

---

**Input:** $D$ adaptation dataset, $t$ canary prefix type, $l$ canary prefix length, $k$ number of selected canaries, $p$ canary prefix probability, $T$ token universe.
**Output:** $\tilde{D}$ modified adaptation dataset
 1: **if** $t =$ "random" **then**
 2:     $C \leftarrow$ Randomly sample $k$ tokens from $T$
 3: **else if** $t =$ "rare" **then**
 4:     $C \leftarrow$ Select the $k$ least frequent tokens from $D$
 5: **else if** $t =$ "common" **then**
 6:     $C \leftarrow$ Select the $k$ most frequent tokens from $D$
 7: **else if** $t =$ "invisible" **then**
 8:     $C \leftarrow$ Randomly sample $k$ invisible tokens from $T$
 9: **end if**
10: $D_0, D_1 \leftarrow$ Randomly split $D$ in two datasets s.t. each
     sample is with probability $p$ in $D_1$
11: $\tilde{D}_1 \leftarrow \{\}$
12: **for** each sample $x \in D_1$ **do**
13:     $y \leftarrow$ Sample with replacement $l$ tokens from $C$
14:     $\tilde{D}_1 \leftarrow \tilde{D}_1 \cup \{\texttt{concat}(y, x)\}$
15: **end for**
16: **return** $D_0 \cup \tilde{D}_1$

---

### B.6 EXTRACTABLE MEMORIZATION

Another privacy concern shown in prior work (Carlini et al., 2023a) is the memorization of samples during pretraining of an LLM. We analyze how adaptations can reduce the effect of memorizing pretraining data. The definition of a memorized sample follows $k$-extractability (Carlini et al., 2023a). Here, we have a prompt $p$ of length $k$ and a suffix $s$. If a model given a prompt $p$ generates exactly $s$, the sequence consisting of $p$ and $s$ concatenated is memorized.

We report the number of identified memorized samples for each Pile subset and Pythia 1B in Table 40 (Section G). Furthermore, we also rely on samples from the Pile reported as memorized in Pythia 2.8B by prior work (Chang et al., 2024). This set of memorized samples consists of 505 sequences, and we refer to it as Mem Pile.

### B.7 COMPUTATIONAL SETUP

We conduct most of our experiments on a single 40GB NVIDIA A100 GPU. However, for larger models, we utilized a single NVIDIA A100 80GB Tensor Core GPU. The training time of the

adaptations varies depending on the applied adaptation method, the model size, the hyperparameters, and whether DP is applied.

## C  ADDITIONAL EXPERIMENTAL RESULTS

### C.1  MIAS

Table 4 and Table 5 present the MIA performance on OOD and IID datasets for the Pythia 1B model. We repeat these experiments with other models from the Pythia (Biderman et al., 2023) and GPT Neo (Black et al., 2021) families to broaden our study. Our findings include results for Pythia 1.4B (Table 6-Table 7), Pythia 410M (Table 8-Table 9), Pythia 160M (Table 10-Table 11), Pythia 70M (Table 12-Table 13), GPT-Neo 1.3B (Table 14-Table 15), and GPT-Neo 125M (Table 16-Table 17). Our results indicate a privacy risk while adapting LLMs, and an attacker has advantages such as architectural knowledge, direct data access, and an exact understanding of the data split, thus allowing for a powerful attack vector. LoRA and Prefix are consistently less vulnerable to MIA among most of the evaluated models and datasets than Full Fine-Tuning and Head-Fine-Tuning.

We also conduct additional experiments on OLMo 1B (Groeneveld et al., 2024a) and OLMo-2-0425 1B (OLMo et al., 2024). Since these models were trained on Dolma and Dolmino, respectively, and neither of them provides a validation set, our experiments are limited to overlap (using the known train data) and OOD scenarios using GermanWiki (Cohere) and SAMSum (Gliwa et al., 2019), which were not part of the training data. Our results presented in Table 18- Table 21 show consistent trends with the ones we reported for the other models, especially when the adaptation data distribution is close to the pretraining data. The privacy risks increase when the adaptation data is closer to the pretraining distribution.

Overall, we observe a similar pattern between Pythia 1B and the other evaluated models. For instance, for Pythia 410M (Table 8 - Table 9), looking at *RMIA (shadow)* using $\varepsilon = 8$, we observe that the average AUC is 0.83, while for IID it is 0.9. Similarly, for Pythia 160M (Table 10 - Table 11), the average AUC is 0.71 for OOD and 0.81 for IID data. These results follow our general trend that IID data taken from the pretraining validation set leaks just as much as data that directly overlaps, thus suggesting distributional closeness as the determining factor of privacy risk. Occasionally, we observe an anomaly, like the AUC for SAMSum in Table 6 being better under a privacy regime ($\varepsilon = 8$) than without privacy protection. This behavior is a consequence of the fact that the loss is higher for the $\varepsilon = \infty$ than for $\varepsilon = 8$. We prioritize having similar loss values across different adaptations for the given dataset and privacy budget. However, in some cases, the span of hyperparameters is too large to ensure that we have a similar loss across different $\varepsilon$ values.

Going further, we also evaluate protection under varying privacy budgets, specifically $\varepsilon \in \{0.1, 0.5, 1, 3, 8\}$. As illustrated in Figure 7, effective defense against privacy attacks, such as MIA, even for OOD data, requires a tight privacy bound of $\varepsilon \leq 0.1$ for all adaptation strategies evaluated.

### C.2  EXPOSURE

Table 22 and Table 23 show the exposure performance of the four types of canary prefixes. With canary exposure, we do not use any shadow or reference models. Therefore, the results are often close to random guessing when using DP for LLM adaptations. However, the results for canary exposure are still much higher than for Min-K%, the closest MIA method executed with the same assumptions.

### C.3  INFLUENCE OF SUBSET SIZE AND COMPLEXITY

We evaluate how subset characteristics, specifically size and complexity (as measured by the perplexity in Table 2 in the original publication on the Pile (Gao et al., 2020)), affect privacy leakage. Specifically, for this experiment, we use train subsets and adapt Pythia 1B privately with $\varepsilon = 8$. As shown in Figure 8, the analysis suggests that privacy leakage in datasets is influenced both by dataset size and the inherent complexity or diversity within the data. For instance, the largest subset with the CC dataset incurs the highest privacy leakage, likely due to its significant volume and potentially diverse content (with a perplexity of around 0.7). The other large and complex subsets, like ArXiv (a

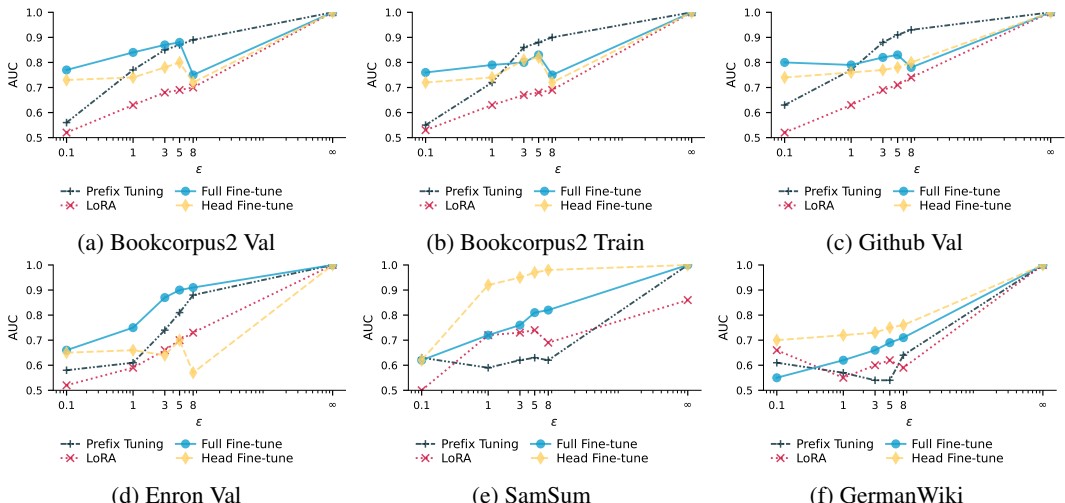

Figure 7: **Membership Inference for Adaptations over Various Privacy Regimes.** We audit the adaptations on the same pretrained LLM. We present the AUC scores obtained with RMIA for the Pythia 1B model adapted on different datasets with $\varepsilon \in \{0.1, 1, 3, 5, 8, \infty\}$.

Table 4: **Membership Inference for OOD Adaptations.** We audit only the adaptations and assume the same pretrained LLM is used for all adaptations. We present the AUC scores obtained with reference, and Min-K% MIAs for the Pythia 1B model adapted on different datasets with $\varepsilon \in \{0.1, 8, \infty\}$.

| | Dataset | SAMSum | | | GermanWiki | | | Average | | |
|---|---|---|---|---|---|---|---|---|---|---|
| MIA | Adaptation | $\varepsilon = \infty$ | $\varepsilon = 8$ | $\varepsilon = 0.1$ | $\varepsilon = \infty$ | $\varepsilon = 8$ | $\varepsilon = 0.1$ | $\varepsilon = \infty$ | $\varepsilon = 8$ | $\varepsilon = 0.1$ |
| | Prefix Tuning | 1.00 | 0.62 | 0.63 | 1.00 | 0.64 | 0.61 | 1.00 | 0.63 | 0.62 |
| | LoRA | 0.86 | 0.69 | 0.50 | 1.00 | 0.59 | 0.66 | 0.93 | 0.64 | 0.58 |
| RMIA (shadow) | Full Fine-Tune | 1.00 | 0.82 | 0.62 | 1.00 | 0.71 | 0.55 | 1.00 | 0.77 | 0.59 |
| | Head Fine-Tune | 1.00 | 0.98 | 0.62 | 1.00 | 0.76 | 0.70 | 1.00 | 0.87 | 0.66 |
| | Average | 0.97 | 0.78 | 0.59 | 1.00 | 0.67 | 0.63 | 0.98 | 0.73 | 0.61 |
| | Prefix Tuning | 0.94 | 0.51 | 0.51 | 0.91 | 0.50 | 0.50 | 0.92 | 0.50 | 0.51 |
| | LoRA | 0.51 | 0.51 | 0.51 | 0.81 | 0.51 | 0.51 | 0.66 | 0.51 | 0.51 |
| RMIA (Pythia 1B) | Full Fine-Tune | 0.94 | 0.51 | 0.51 | 0.98 | 0.51 | 0.51 | 0.96 | 0.51 | 0.51 |
| | Head Fine-Tune | 0.96 | 0.52 | 0.51 | 0.97 | 0.51 | 0.50 | 0.97 | 0.52 | 0.50 |
| | Average | 0.84 | 0.51 | 0.51 | 0.92 | 0.51 | 0.50 | 0.88 | 0.51 | 0.51 |
| | Prefix Tuning | 0.93 | 0.50 | 0.51 | 0.92 | 0.50 | 0.50 | 0.92 | 0.50 | 0.50 |
| | LoRA | 0.51 | 0.51 | 0.51 | 0.82 | 0.51 | 0.51 | 0.66 | 0.51 | 0.51 |
| Reference (Pythia 1B) | Full Fine-Tune | 0.94 | 0.51 | 0.51 | 0.99 | 0.51 | 0.50 | 0.96 | 0.51 | 0.51 |
| | Head Fine-Tune | 0.97 | 0.52 | 0.51 | 0.98 | 0.51 | 0.50 | 0.97 | 0.51 | 0.50 |
| | Average | 0.84 | 0.51 | 0.51 | 0.93 | 0.51 | 0.50 | 0.88 | 0.51 | 0.51 |
| | Prefix Tuning | 0.84 | 0.51 | 0.51 | 0.71 | 0.50 | 0.50 | 0.78 | 0.50 | 0.50 |
| | LoRA | 0.51 | 0.51 | 0.50 | 0.61 | 0.51 | 0.51 | 0.56 | 0.51 | 0.51 |
| Min-K% | Full Fine-Tune | 0.83 | 0.51 | 0.50 | 0.88 | 0.51 | 0.50 | 0.86 | 0.51 | 0.50 |
| | Head Fine-Tune | 0.92 | 0.51 | 0.50 | 0.87 | 0.51 | 0.51 | 0.89 | 0.51 | 0.50 |
| | Average | 0.77 | 0.51 | 0.50 | 0.77 | 0.50 | 0.51 | 0.77 | 0.51 | 0.50 |

perplexity of around 0.77), also have high leakage levels. For ArXiv, compared to Freelaw (which is similar in size but less diverse, with a perplexity of around 0.6), ArXiv's diversity increases leakage, as more unique samples may need to be memorized. Finally, much smaller and more structured subsets like Europarl (with a perplexity of 0.75) and Enron Emails (the smallest subset) exhibit the least leakage, likely due to limited diversity and lower complexity.

## C.4 PER-EPOCH LOSS

We compare the development of AUC scores during training on IID and overlap data, as shown in Figure 9. These results display the AUC score at each epoch during training. To better compare IID and overlap data, we adjust the x-axis to represent the loss difference at each training step, calculated as the initial pretraining loss minus the adapted loss at the current training step. This calibration of the x-axis allows us to compare the two dataset types more precisely. With this setup, we evaluate two subsets of the Pile pretraining set: GitHub and Bookcorpus2. First, the figures indicate that further adapting a model on IID data does not significantly improve its performance on that data, with the loss decreasing by only a maximum of 0.015 (GitHub with Full Fine-Tune). However, the observed increase in AUC score throughout training shows that the model does learn from the adaptation data.

Table 5: **Membership Inference for in-distribution (IID) Adaptations.** We use the same setup as in Table 4.

| MIA | Adaptation | Bookcorpus2 Val $\varepsilon=\infty$ | $\varepsilon=8$ | $\varepsilon=0.1$ | Bookcorpus2 Train $\varepsilon=\infty$ | $\varepsilon=8$ | $\varepsilon=0.1$ | Github val $\varepsilon=\infty$ | $\varepsilon=8$ | $\varepsilon=0.1$ | Enron Val $\varepsilon=\infty$ | $\varepsilon=8$ | $\varepsilon=0.1$ | Average $\varepsilon=\infty$ | $\varepsilon=8$ | $\varepsilon=0.1$ |
|---|---|---|---|---|---|---|---|---|---|---|---|---|---|---|---|---|
| RMIA (shadow) | Prefix Tuning | 1.00 | 0.89 | 0.56 | 1.00 | 0.90 | 0.55 | 1.00 | 0.93 | 0.63 | 1.00 | 0.88 | 0.58 | 1.00 | 0.90 | 0.58 |
| | LoRA | 1.00 | 0.70 | 0.52 | 1.00 | 0.69 | 0.53 | 1.00 | 0.74 | 0.52 | 1.00 | 0.73 | 0.52 | 1.00 | 0.71 | 0.52 |
| | Full Fine-Tune | 1.00 | 0.75 | 0.77 | 1.00 | 0.75 | 0.76 | 1.00 | 0.78 | 0.80 | 1.00 | 0.91 | 0.66 | 1.00 | 0.80 | 0.75 |
| | Head Fine-Tune | 1.00 | 0.72 | 0.73 | 1.00 | 0.72 | 0.72 | 1.00 | 0.80 | 0.74 | 1.00 | 0.57 | 0.65 | 1.00 | 0.70 | 0.71 |
| | Average | 1.00 | 0.77 | 0.65 | 1.00 | 0.76 | 0.64 | 1.00 | 0.81 | 0.67 | 1.00 | 0.77 | 0.60 | 1.00 | 0.78 | 0.64 |
| RMIA (Pythia 1B) | Prefix Tuning | 0.91 | 0.56 | 0.51 | 0.97 | 0.57 | 0.50 | 0.96 | 0.54 | 0.52 | 0.98 | 0.54 | 0.51 | 0.95 | 0.55 | 0.51 |
| | LoRA | 0.87 | 0.52 | 0.52 | 0.96 | 0.51 | 0.51 | 0.91 | 0.51 | 0.50 | 0.98 | 0.56 | 0.51 | 0.93 | 0.52 | 0.51 |
| | Full Fine-Tune | 0.99 | 0.54 | 0.52 | 1.00 | 0.54 | 0.52 | 0.99 | 0.53 | 0.52 | 0.99 | 0.59 | 0.50 | 1.00 | 0.55 | 0.51 |
| | Head Fine-Tune | 0.96 | 0.57 | 0.52 | 0.99 | 0.56 | 0.51 | 0.99 | 0.65 | 0.52 | 0.99 | 0.58 | 0.51 | 0.99 | 0.58 | 0.51 |
| | Average | 0.94 | 0.55 | 0.52 | 0.98 | 0.55 | 0.51 | 0.96 | 0.56 | 0.51 | 0.99 | 0.56 | 0.51 | 0.97 | 0.55 | 0.51 |
| Reference (Pythia 1B) | Prefix Tuning | 0.93 | 0.56 | 0.52 | 0.97 | 0.57 | 0.50 | 0.97 | 0.53 | 0.51 | 0.97 | 0.54 | 0.50 | 0.96 | 0.55 | 0.51 |
| | LoRA | 0.89 | 0.52 | 0.52 | 0.97 | 0.51 | 0.51 | 0.92 | 0.51 | 0.50 | 0.97 | 0.55 | 0.51 | 0.94 | 0.52 | 0.51 |
| | Full Fine-Tune | 1.00 | 0.54 | 0.52 | 1.00 | 0.54 | 0.52 | 0.99 | 0.54 | 0.52 | 0.98 | 0.59 | 0.50 | 0.99 | 0.55 | 0.51 |
| | Head Fine-Tune | 0.98 | 0.57 | 0.52 | 1.00 | 0.56 | 0.51 | 0.99 | 0.66 | 0.50 | 0.99 | 0.54 | 0.50 | 0.99 | 0.58 | 0.51 |
| | Average | 0.95 | 0.55 | 0.52 | 0.98 | 0.55 | 0.51 | 0.97 | 0.56 | 0.51 | 0.98 | 0.55 | 0.50 | 0.97 | 0.55 | 0.51 |
| Min-K% | Prefix Tuning | 0.78 | 0.51 | 0.50 | 0.70 | 0.51 | 0.50 | 0.65 | 0.52 | 0.52 | 0.66 | 0.51 | 0.52 | 0.70 | 0.51 | 0.51 |
| | LoRA | 0.67 | 0.51 | 0.51 | 0.63 | 0.50 | 0.50 | 0.61 | 0.52 | 0.52 | 0.65 | 0.51 | 0.51 | 0.64 | 0.51 | 0.51 |
| | Full Fine-Tune | 0.87 | 0.51 | 0.51 | 0.82 | 0.50 | 0.50 | 0.77 | 0.52 | 0.52 | 0.78 | 0.51 | 0.51 | 0.81 | 0.51 | 0.51 |
| | Head Fine-Tune | 0.75 | 0.51 | 0.51 | 0.72 | 0.50 | 0.51 | 0.64 | 0.52 | 0.52 | 0.70 | 0.51 | 0.51 | 0.70 | 0.51 | 0.51 |
| | Average | 0.77 | 0.51 | 0.51 | 0.72 | 0.50 | 0.50 | 0.67 | 0.52 | 0.52 | 0.70 | 0.51 | 0.51 | 0.71 | 0.51 | 0.51 |

Table 6: **Membership Inference for OOD Adaptations using Pythia 1.4B.** We present the AUC scores obtained with reference, and Min-K% MIAs for the Pythia 1.4B model adapted on different datasets with $\varepsilon \in \{0.1, 8, \infty\}$.

| MIA | Adaptation | Samsum $\varepsilon=\infty$ | $\varepsilon=8$ | $\varepsilon=0.1$ | German Wiki $\varepsilon=\infty$ | $\varepsilon=8$ | $\varepsilon=0.1$ | Average $\varepsilon=\infty$ | $\varepsilon=8$ | $\varepsilon=0.1$ |
|---|---|---|---|---|---|---|---|---|---|---|
| RMIA (shadow) | Prefix | 0.58 | 0.77 | 0.54 | 1.00 | 0.85 | 0.56 | 0.79 | 0.81 | 0.55 |
| | LoRA | 0.53 | 0.79 | 0.51 | 1.00 | 0.82 | 0.64 | 0.76 | 0.81 | 0.58 |
| | Full Fine-Tune | 1.00 | 0.99 | 0.62 | 1.00 | 1.00 | 0.90 | 1.00 | 1.00 | 0.76 |
| | Head Fine-Tune | 0.95 | 1.00 | 0.85 | 1.00 | 0.90 | 0.89 | 0.97 | 0.95 | 0.87 |
| | Average | 0.76 | 0.94 | 0.63 | 1.00 | 0.89 | 0.75 | 0.88 | 0.89 | 0.69 |
| RMIA (Pythia 1B) | Prefix | 0.52 | 0.52 | 0.51 | 0.92 | 0.53 | 0.50 | 0.72 | 0.53 | 0.51 |
| | LoRA | 0.50 | 0.54 | 0.50 | 0.97 | 0.51 | 0.50 | 0.74 | 0.52 | 0.50 |
| | Full Fine-Tune | 1.00 | 0.52 | 0.50 | 1.00 | 0.58 | 0.51 | 1.00 | 0.55 | 0.51 |
| | Head Fine-Tune | 0.51 | 0.56 | 0.51 | 0.92 | 0.61 | 0.52 | 0.71 | 0.59 | 0.51 |
| | Average | 0.63 | 0.54 | 0.50 | 0.95 | 0.56 | 0.51 | 0.79 | 0.55 | 0.51 |
| Reference (Pythia 1B) | Prefix | 0.52 | 0.52 | 0.51 | 0.93 | 0.54 | 0.49 | 0.72 | 0.53 | 0.50 |
| | LoRA | 0.50 | 0.53 | 0.50 | 0.98 | 0.51 | 0.49 | 0.74 | 0.52 | 0.49 |
| | Full Fine-Tune | 1.00 | 0.52 | 0.50 | 1.00 | 0.59 | 0.51 | 1.00 | 0.55 | 0.51 |
| | Head Fine-Tune | 0.51 | 0.56 | 0.51 | 0.93 | 0.61 | 0.51 | 0.72 | 0.59 | 0.51 |
| | Average | 0.63 | 0.53 | 0.50 | 0.96 | 0.56 | 0.50 | 0.80 | 0.55 | 0.50 |
| Min-K% | Prefix | 0.52 | 0.51 | 0.51 | 0.70 | 0.53 | 0.50 | 0.61 | 0.52 | 0.51 |
| | LoRA | 0.50 | 0.52 | 0.50 | 0.79 | 0.52 | 0.51 | 0.65 | 0.52 | 0.51 |
| | Full Fine-Tune | 1.00 | 0.51 | 0.51 | 0.98 | 0.54 | 0.52 | 0.99 | 0.53 | 0.51 |
| | Head Fine-Tune | 0.51 | 0.53 | 0.51 | 0.74 | 0.55 | 0.52 | 0.62 | 0.54 | 0.52 |
| | Average | 0.63 | 0.52 | 0.51 | 0.80 | 0.53 | 0.51 | 0.72 | 0.53 | 0.51 |

Table 7: **Membership Inference for IID Adaptations using Pythia 1.4B.** We present the AUC scores obtained with reference, and Min-K% MIAs for the Pythia 1.4B model adapted on different datasets with $\varepsilon \in \{0.1, 8, \infty\}$.

| MIA | Adaptation | Pile Bookcorpus2 Val $\varepsilon=\infty$ | $\varepsilon=8$ | $\varepsilon=0.1$ | Pile Bookcorpus2 Train $\varepsilon=\infty$ | $\varepsilon=8$ | $\varepsilon=0.1$ | Pile Github Val $\varepsilon=\infty$ | $\varepsilon=8$ | $\varepsilon=0.1$ | Pile Enron Val $\varepsilon=\infty$ | $\varepsilon=8$ | $\varepsilon=0.1$ | Average $\varepsilon=\infty$ | $\varepsilon=8$ | $\varepsilon=0.1$ |
|---|---|---|---|---|---|---|---|---|---|---|---|---|---|---|---|---|
| RMIA (shadow) | Prefix | 1.00 | 0.68 | 0.54 | 1.00 | 0.68 | 0.55 | 1.00 | 0.62 | 0.56 | 1.00 | 0.72 | 0.60 | 1.00 | 0.68 | 0.56 |
| | LoRA | 0.96 | 0.99 | 0.51 | 0.74 | 0.98 | 0.52 | 1.00 | 0.97 | 0.53 | 1.00 | 0.99 | 0.67 | 0.93 | 0.98 | 0.56 |
| | Full Fine-Tune | 0.98 | 1.00 | 0.71 | 0.99 | 0.99 | 0.70 | 1.00 | 0.99 | 0.71 | 1.00 | 1.00 | 0.62 | 0.99 | 0.99 | 0.69 |
| | Head Fine-Tune | 1.00 | 1.00 | 0.72 | 1.00 | 1.00 | 0.69 | 1.00 | 1.00 | 0.72 | 1.00 | 1.00 | 0.64 | 1.00 | 1.00 | 0.69 |
| | Average | 0.99 | 0.92 | 0.62 | 0.93 | 0.92 | 0.62 | 1.00 | 0.89 | 0.63 | 1.00 | 0.93 | 0.63 | 0.98 | 0.91 | 0.62 |
| RMIA (Pythia 1B) | Prefix | 0.79 | 0.52 | 0.51 | 0.85 | 0.52 | 0.51 | 0.76 | 0.51 | 0.51 | 0.78 | 0.51 | 0.51 | 0.79 | 0.52 | 0.51 |
| | LoRA | 0.56 | 0.58 | 0.51 | 0.50 | 0.59 | 0.51 | 0.90 | 0.57 | 0.52 | 0.97 | 0.59 | 0.51 | 0.73 | 0.58 | 0.51 |
| | Full Fine-Tune | 0.64 | 0.59 | 0.51 | 0.65 | 0.58 | 0.50 | 0.97 | 0.55 | 0.50 | 0.99 | 0.57 | 0.51 | 0.81 | 0.57 | 0.51 |
| | Head Fine-Tune | 0.79 | 0.64 | 0.50 | 0.54 | 0.63 | 0.50 | 0.91 | 0.64 | 0.51 | 0.99 | 0.64 | 0.51 | 0.81 | 0.64 | 0.51 |
| | Average | 0.69 | 0.58 | 0.51 | 0.64 | 0.58 | 0.50 | 0.88 | 0.57 | 0.51 | 0.93 | 0.58 | 0.51 | 0.79 | 0.58 | 0.51 |
| Reference (Pythia 1B) | Prefix | 0.80 | 0.52 | 0.51 | 0.86 | 0.52 | 0.51 | 0.76 | 0.50 | 0.50 | 0.77 | 0.49 | 0.50 | 0.80 | 0.51 | 0.50 |
| | LoRA | 0.57 | 0.58 | 0.51 | 0.49 | 0.59 | 0.51 | 0.92 | 0.55 | 0.50 | 0.96 | 0.60 | 0.50 | 0.73 | 0.58 | 0.51 |
| | Full Fine-Tune | 0.64 | 0.58 | 0.51 | 0.65 | 0.57 | 0.49 | 0.98 | 0.53 | 0.50 | 0.99 | 0.58 | 0.51 | 0.81 | 0.56 | 0.50 |
| | Head Fine-Tune | 0.80 | 0.64 | 0.51 | 0.54 | 0.64 | 0.50 | 0.91 | 0.67 | 0.51 | 0.99 | 0.65 | 0.50 | 0.81 | 0.65 | 0.50 |
| | Average | 0.70 | 0.58 | 0.51 | 0.64 | 0.58 | 0.50 | 0.89 | 0.56 | 0.50 | 0.93 | 0.58 | 0.50 | 0.79 | 0.58 | 0.50 |
| Min-K% | Prefix | 0.61 | 0.50 | 0.49 | 0.58 | 0.50 | 0.50 | 0.57 | 0.52 | 0.51 | 0.57 | 0.51 | 0.51 | 0.58 | 0.51 | 0.51 |
| | LoRA | 0.50 | 0.51 | 0.50 | 0.50 | 0.51 | 0.50 | 0.64 | 0.53 | 0.52 | 0.68 | 0.52 | 0.51 | 0.58 | 0.52 | 0.51 |
| | Full Fine-Tune | 0.52 | 0.53 | 0.49 | 0.52 | 0.54 | 0.50 | 0.73 | 0.56 | 0.51 | 0.83 | 0.52 | 0.51 | 0.65 | 0.54 | 0.51 |
| | Head Fine-Tune | 0.57 | 0.53 | 0.49 | 0.51 | 0.53 | 0.50 | 0.61 | 0.54 | 0.51 | 0.90 | 0.53 | 0.51 | 0.64 | 0.53 | 0.51 |
| | Average | 0.55 | 0.52 | 0.50 | 0.53 | 0.52 | 0.50 | 0.64 | 0.54 | 0.52 | 0.75 | 0.52 | 0.51 | 0.61 | 0.52 | 0.51 |

## C.5 PREFIX EXPOSURE

To investigate the source of privacy leakage, we present exposure observed with canary prefixes of varying lengths, after adapting Pythia 1B to the GitHub Val dataset with $\varepsilon = \infty$. Figure 10 show the

Table 8: **Membership Inference for OOD Adaptations using Pythia 410M.** We present the AUC scores obtained with reference, and Min-K% MIAs for the Pythia 410M model adapted on different datasets with $\varepsilon \in \{0.1, 8, \infty\}$.

| MIA | Adaptation | Samsum $\varepsilon=\infty$ | $\varepsilon=8$ | $\varepsilon=0.1$ | German Wiki $\varepsilon=\infty$ | $\varepsilon=8$ | $\varepsilon=0.1$ | Average $\varepsilon=\infty$ | $\varepsilon=8$ | $\varepsilon=0.1$ |
|---|---|---|---|---|---|---|---|---|---|---|
| RMIA (shadow) | Prefix | 0.87 | 0.67 | 0.51 | 0.90 | 0.66 | 0.50 | 0.88 | 0.67 | 0.51 |
| | LoRA | 0.93 | 0.62 | 0.52 | 0.71 | 0.97 | 0.54 | 0.82 | 0.79 | 0.53 |
| | Full Fine-Tune | 0.99 | 0.98 | 0.52 | 1.00 | 1.00 | 0.53 | 1.00 | 0.99 | 0.52 |
| | Head Fine-Tune | 1.00 | 0.76 | 0.76 | 0.94 | 1.00 | 0.82 | 0.97 | 0.88 | 0.79 |
| | Average | 0.95 | 0.76 | 0.58 | 0.89 | 0.91 | 0.60 | 0.92 | 0.83 | 0.59 |
| RMIA (Pythia 1B) | Prefix | 0.54 | 0.52 | 0.51 | 0.58 | 0.51 | 0.50 | 0.56 | 0.52 | 0.51 |
| | LoRA | 0.52 | 0.50 | 0.52 | 0.51 | 0.56 | 0.50 | 0.51 | 0.53 | 0.51 |
| | Full Fine-Tune | 0.80 | 0.55 | 0.50 | 0.93 | 0.58 | 0.51 | 0.86 | 0.56 | 0.50 |
| | Head Fine-Tune | 0.80 | 0.50 | 0.50 | 0.51 | 0.62 | 0.51 | 0.66 | 0.56 | 0.51 |
| | Average | 0.66 | 0.52 | 0.51 | 0.63 | 0.57 | 0.50 | 0.65 | 0.54 | 0.51 |
| Reference (Pythia 1B) | Prefix | 0.54 | 0.52 | 0.51 | 0.57 | 0.50 | 0.48 | 0.55 | 0.51 | 0.49 |
| | LoRA | 0.52 | 0.49 | 0.51 | 0.50 | 0.55 | 0.48 | 0.51 | 0.52 | 0.49 |
| | Full Fine-Tune | 0.79 | 0.55 | 0.50 | 0.92 | 0.56 | 0.49 | 0.85 | 0.55 | 0.49 |
| | Head Fine-Tune | 0.79 | 0.49 | 0.50 | 0.51 | 0.62 | 0.49 | 0.65 | 0.56 | 0.49 |
| | Average | 0.66 | 0.51 | 0.51 | 0.63 | 0.56 | 0.48 | 0.64 | 0.54 | 0.49 |
| Min-K% | Prefix | 0.52 | 0.51 | 0.51 | 0.54 | 0.52 | 0.51 | 0.53 | 0.52 | 0.51 |
| | LoRA | 0.51 | 0.50 | 0.51 | 0.52 | 0.54 | 0.51 | 0.51 | 0.52 | 0.51 |
| | Full Fine-Tune | 0.69 | 0.53 | 0.50 | 0.79 | 0.54 | 0.52 | 0.74 | 0.53 | 0.51 |
| | Head Fine-Tune | 0.69 | 0.50 | 0.50 | 0.52 | 0.56 | 0.52 | 0.60 | 0.53 | 0.51 |
| | Average | 0.60 | 0.51 | 0.51 | 0.59 | 0.54 | 0.51 | 0.60 | 0.53 | 0.51 |

Table 9: **Membership Inference for IID Adaptations using Pythia 410M.** We present the AUC scores obtained with reference, and Min-K% MIAs for the Pythia 410M model adapted on different datasets with $\varepsilon \in \{0.1, 8, \infty\}$.

| MIA | Adaptation | Pile Bookcorpus2 Val $\varepsilon=\infty$ | $\varepsilon=8$ | $\varepsilon=0.1$ | Pile Bookcorpus2 Train $\varepsilon=\infty$ | $\varepsilon=8$ | $\varepsilon=0.1$ | Pile Github Val $\varepsilon=\infty$ | $\varepsilon=8$ | $\varepsilon=0.1$ | Pile Enron Val $\varepsilon=\infty$ | $\varepsilon=8$ | $\varepsilon=0.1$ | Average $\varepsilon=\infty$ | $\varepsilon=8$ | $\varepsilon=0.1$ |
|---|---|---|---|---|---|---|---|---|---|---|---|---|---|---|---|---|
| RMIA (shadow) | Prefix | 0.83 | 0.65 | 0.55 | 0.86 | 0.67 | 0.52 | 0.65 | 0.69 | 0.51 | 0.89 | 0.65 | 0.53 | 0.81 | 0.67 | 0.53 |
| | LoRA | 0.72 | 0.91 | 0.58 | 0.73 | 0.89 | 0.57 | 0.74 | 0.92 | 0.51 | 0.74 | 0.98 | 0.57 | 0.73 | 0.92 | 0.56 |
| | Full Fine-Tune | 1.00 | 1.00 | 0.60 | 1.00 | 1.00 | 0.57 | 1.00 | 0.98 | 0.51 | 0.99 | 0.98 | 0.66 | 1.00 | 0.99 | 0.58 |
| | Head Fine-Tune | 0.96 | 1.00 | 0.74 | 0.96 | 1.00 | 0.69 | 1.00 | 1.00 | 0.62 | 1.00 | 1.00 | 0.72 | 0.98 | 1.00 | 0.69 |
| | Average | 0.87 | 0.89 | 0.62 | 0.89 | 0.89 | 0.59 | 0.85 | 0.90 | 0.54 | 0.91 | 0.90 | 0.62 | 0.88 | 0.90 | 0.59 |
| RMIA (Pythia 1B) | Prefix | 0.56 | 0.51 | 0.51 | 0.56 | 0.52 | 0.52 | 0.53 | 0.52 | 0.51 | 0.53 | 0.51 | 0.51 | 0.54 | 0.52 | 0.51 |
| | LoRA | 0.50 | 0.55 | 0.51 | 0.50 | 0.55 | 0.51 | 0.51 | 0.54 | 0.50 | 0.50 | 0.54 | 0.51 | 0.50 | 0.54 | 0.51 |
| | Full Fine-Tune | 0.91 | 0.58 | 0.50 | 0.93 | 0.59 | 0.51 | 0.91 | 0.55 | 0.52 | 0.83 | 0.54 | 0.50 | 0.90 | 0.57 | 0.51 |
| | Head Fine-Tune | 0.51 | 0.62 | 0.50 | 0.51 | 0.62 | 0.52 | 0.90 | 0.59 | 0.52 | 0.91 | 0.58 | 0.49 | 0.71 | 0.60 | 0.51 |
| | Average | 0.62 | 0.57 | 0.50 | 0.63 | 0.57 | 0.51 | 0.71 | 0.55 | 0.51 | 0.69 | 0.54 | 0.50 | 0.66 | 0.56 | 0.51 |
| Reference (Pythia 1B) | Prefix | 0.56 | 0.51 | 0.51 | 0.55 | 0.52 | 0.51 | 0.51 | 0.50 | 0.49 | 0.52 | 0.50 | 0.49 | 0.54 | 0.51 | 0.50 |
| | LoRA | 0.51 | 0.55 | 0.51 | 0.51 | 0.54 | 0.51 | 0.50 | 0.52 | 0.49 | 0.47 | 0.52 | 0.50 | 0.50 | 0.53 | 0.50 |
| | Full Fine-Tune | 0.91 | 0.57 | 0.50 | 0.93 | 0.58 | 0.51 | 0.88 | 0.53 | 0.49 | 0.80 | 0.52 | 0.49 | 0.88 | 0.55 | 0.50 |
| | Head Fine-Tune | 0.51 | 0.62 | 0.59 | 0.51 | 0.62 | 0.52 | 0.87 | 0.59 | 0.50 | 0.88 | 0.58 | 0.49 | 0.70 | 0.60 | 0.51 |
| | Average | 0.62 | 0.56 | 0.51 | 0.63 | 0.57 | 0.51 | 0.69 | 0.53 | 0.49 | 0.67 | 0.53 | 0.49 | 0.65 | 0.55 | 0.50 |
| Min-K% | Prefix | 0.51 | 0.50 | 0.50 | 0.52 | 0.51 | 0.51 | 0.53 | 0.52 | 0.50 | 0.52 | 0.51 | 0.52 | 0.52 | 0.51 | 0.51 |
| | LoRA | 0.50 | 0.51 | 0.50 | 0.50 | 0.52 | 0.50 | 0.51 | 0.54 | 0.51 | 0.50 | 0.52 | 0.51 | 0.50 | 0.52 | 0.51 |
| | Full Fine-Tune | 0.82 | 0.52 | 0.50 | 0.80 | 0.53 | 0.50 | 0.74 | 0.56 | 0.52 | 0.75 | 0.54 | 0.49 | 0.78 | 0.54 | 0.50 |
| | Head Fine-Tune | 0.50 | 0.53 | 0.49 | 0.50 | 0.53 | 0.51 | 0.62 | 0.53 | 0.51 | 0.68 | 0.52 | 0.50 | 0.57 | 0.53 | 0.50 |
| | Average | 0.58 | 0.52 | 0.50 | 0.58 | 0.52 | 0.51 | 0.60 | 0.54 | 0.51 | 0.61 | 0.52 | 0.51 | 0.59 | 0.52 | 0.51 |

Table 10: **Membership Inference for OOD Adaptations using Pythia 160M.** We present the AUC scores obtained with reference, and Min-K% MIAs for the Pythia 160M model adapted on different datasets with $\varepsilon \in \{0.1, 8, \infty\}$.

| MIA | Adaptation | Samsum $\varepsilon=\infty$ | $\varepsilon=8$ | $\varepsilon=0.1$ | German Wiki $\varepsilon=\infty$ | $\varepsilon=8$ | $\varepsilon=0.1$ | Average $\varepsilon=\infty$ | $\varepsilon=8$ | $\varepsilon=0.1$ |
|---|---|---|---|---|---|---|---|---|---|---|
| RMIA (shadow) | Prefix | 0.55 | 0.53 | 0.52 | 0.56 | 0.62 | 0.53 | 0.56 | 0.57 | 0.53 |
| | LoRA | 0.78 | 0.61 | 0.55 | 0.62 | 0.57 | 0.59 | 0.70 | 0.59 | 0.57 |
| | Full Fine-Tune | 1.00 | 0.74 | 0.61 | 0.90 | 0.99 | 0.65 | 0.95 | 0.86 | 0.63 |
| | Head Fine-Tune | 1.00 | 0.89 | 0.73 | 0.96 | 0.75 | 0.77 | 0.98 | 0.82 | 0.75 |
| | Average | 0.83 | 0.69 | 0.60 | 0.76 | 0.73 | 0.64 | 0.80 | 0.71 | 0.62 |
| RMIA (Pythia 1B) | Prefix | 0.51 | 0.51 | 0.50 | 0.51 | 0.51 | 0.51 | 0.51 | 0.51 | 0.50 |
| | LoRA | 0.51 | 0.50 | 0.50 | 0.51 | 0.51 | 0.50 | 0.51 | 0.50 | 0.50 |
| | Full Fine-Tune | 0.81 | 0.52 | 0.50 | 0.52 | 0.55 | 0.50 | 0.66 | 0.53 | 0.50 |
| | Head Fine-Tune | 0.69 | 0.51 | 0.50 | 0.52 | 0.51 | 0.52 | 0.60 | 0.51 | 0.51 |
| | Average | 0.63 | 0.51 | 0.50 | 0.51 | 0.52 | 0.51 | 0.57 | 0.51 | 0.50 |
| Reference (Pythia 1B) | Prefix | 0.51 | 0.51 | 0.51 | 0.50 | 0.49 | 0.49 | 0.50 | 0.50 | 0.50 |
| | LoRA | 0.51 | 0.50 | 0.51 | 0.49 | 0.49 | 0.49 | 0.50 | 0.50 | 0.50 |
| | Full Fine-Tune | 0.79 | 0.52 | 0.50 | 0.50 | 0.53 | 0.49 | 0.65 | 0.52 | 0.49 |
| | Head Fine-Tune | 0.69 | 0.51 | 0.50 | 0.50 | 0.49 | 0.50 | 0.60 | 0.50 | 0.50 |
| | Average | 0.63 | 0.51 | 0.50 | 0.50 | 0.50 | 0.49 | 0.56 | 0.51 | 0.50 |
| Min-K% | Prefix | 0.51 | 0.51 | 0.51 | 0.52 | 0.52 | 0.51 | 0.51 | 0.51 | 0.51 |
| | LoRA | 0.51 | 0.50 | 0.50 | 0.51 | 0.51 | 0.51 | 0.51 | 0.51 | 0.51 |
| | Full Fine-Tune | 0.71 | 0.52 | 0.50 | 0.52 | 0.53 | 0.51 | 0.61 | 0.52 | 0.51 |
| | Head Fine-Tune | 0.63 | 0.51 | 0.50 | 0.52 | 0.51 | 0.52 | 0.58 | 0.51 | 0.51 |
| | Average | 0.59 | 0.51 | 0.50 | 0.52 | 0.52 | 0.51 | 0.55 | 0.51 | 0.51 |

Table 11: **Membership Inference for IID Adaptations using Pythia 160M.** We present the AUC scores obtained with reference, and Min-K% MIAs for the Pythia 160M model adapted on different datasets with $\varepsilon \in \{0.1, 8, \infty\}$.

| MIA | Adaptation | Pile Bookcorpus2 Val $\varepsilon=\infty$ | $\varepsilon=8$ | $\varepsilon=0.1$ | Pile Bookcorpus2 Train $\varepsilon=\infty$ | $\varepsilon=8$ | $\varepsilon=0.1$ | Pile Github Val $\varepsilon=\infty$ | $\varepsilon=8$ | $\varepsilon=0.1$ | Pile Enron Val $\varepsilon=\infty$ | $\varepsilon=8$ | $\varepsilon=0.1$ | Average $\varepsilon=\infty$ | $\varepsilon=8$ | $\varepsilon=0.1$ |
|---|---|---|---|---|---|---|---|---|---|---|---|---|---|---|---|---|
| RMIA (shadow) | Prefix | 0.61 | 0.72 | 0.53 | 0.61 | 0.71 | 0.54 | 0.57 | 0.67 | 0.51 | 0.66 | 0.75 | 0.54 | 0.61 | 0.71 | 0.53 |
| | LoRA | 0.82 | 0.60 | 0.54 | 0.83 | 0.79 | 0.55 | 0.80 | 0.82 | 0.53 | 0.91 | 0.61 | 0.53 | 0.84 | 0.71 | 0.54 |
| | Full Fine-Tune | 1.00 | 0.89 | 0.58 | 0.89 | 0.93 | 0.56 | 1.00 | 0.95 | 0.56 | 1.00 | 0.97 | 0.52 | 0.97 | 0.94 | 0.55 |
| | Head Fine-Tune | 1.00 | 0.74 | 0.75 | 1.00 | 0.97 | 0.72 | 1.00 | 0.99 | 0.62 | 1.00 | 0.80 | 0.70 | 1.00 | 0.87 | 0.70 |
| | Average | 0.86 | 0.74 | 0.60 | 0.83 | 0.85 | 0.59 | 0.84 | 0.86 | 0.55 | 0.89 | 0.78 | 0.57 | 0.86 | 0.81 | 0.58 |
| RMIA (Pythia 1B) | Prefix | 0.50 | 0.50 | 0.50 | 0.52 | 0.53 | 0.52 | 0.51 | 0.51 | 0.51 | 0.50 | 0.50 | 0.50 | 0.51 | 0.51 | 0.51 |
| | LoRA | 0.50 | 0.50 | 0.50 | 0.51 | 0.52 | 0.51 | 0.51 | 0.51 | 0.51 | 0.50 | 0.50 | 0.50 | 0.51 | 0.51 | 0.51 |
| | Full Fine-Tune | 1.00 | 0.53 | 0.50 | 0.52 | 0.54 | 0.50 | 0.85 | 0.52 | 0.51 | 0.99 | 0.53 | 0.50 | 0.84 | 0.53 | 0.51 |
| | Head Fine-Tune | 0.71 | 0.50 | 0.50 | 0.74 | 0.56 | 0.51 | 0.67 | 0.55 | 0.51 | 0.78 | 0.50 | 0.50 | 0.73 | 0.53 | 0.51 |
| | Average | 0.68 | 0.51 | 0.50 | 0.57 | 0.53 | 0.51 | 0.64 | 0.52 | 0.51 | 0.69 | 0.51 | 0.50 | 0.65 | 0.52 | 0.51 |
| Reference (Pythia 1B) | Prefix | 0.50 | 0.51 | 0.50 | 0.52 | 0.52 | 0.52 | 0.50 | 0.50 | 0.50 | 0.50 | 0.50 | 0.50 | 0.51 | 0.51 | 0.50 |
| | LoRA | 0.51 | 0.50 | 0.50 | 0.52 | 0.52 | 0.51 | 0.50 | 0.51 | 0.50 | 0.49 | 0.49 | 0.49 | 0.50 | 0.50 | 0.50 |
| | Full Fine-Tune | 1.00 | 0.53 | 0.50 | 0.52 | 0.53 | 0.51 | 0.81 | 0.52 | 0.50 | 0.96 | 0.51 | 0.50 | 0.82 | 0.52 | 0.50 |
| | Head Fine-Tune | 0.69 | 0.50 | 0.50 | 0.71 | 0.55 | 0.51 | 0.64 | 0.54 | 0.50 | 0.72 | 0.49 | 0.49 | 0.69 | 0.52 | 0.50 |
| | Average | 0.68 | 0.51 | 0.50 | 0.57 | 0.53 | 0.51 | 0.61 | 0.52 | 0.50 | 0.67 | 0.50 | 0.49 | 0.63 | 0.51 | 0.50 |
| Min-K% | Prefix | 0.50 | 0.50 | 0.50 | 0.51 | 0.51 | 0.51 | 0.52 | 0.52 | 0.51 | 0.50 | 0.50 | 0.50 | 0.51 | 0.51 | 0.51 |
| | LoRA | 0.50 | 0.50 | 0.50 | 0.51 | 0.51 | 0.50 | 0.52 | 0.52 | 0.51 | 0.50 | 0.50 | 0.50 | 0.51 | 0.50 | 0.50 |
| | Full Fine-Tune | 0.96 | 0.51 | 0.50 | 0.51 | 0.51 | 0.50 | 0.67 | 0.52 | 0.52 | 0.93 | 0.52 | 0.51 | 0.77 | 0.52 | 0.50 |
| | Head Fine-Tune | 0.62 | 0.50 | 0.50 | 0.62 | 0.52 | 0.51 | 0.60 | 0.53 | 0.51 | 0.72 | 0.50 | 0.50 | 0.64 | 0.51 | 0.50 |
| | Average | 0.64 | 0.50 | 0.50 | 0.54 | 0.51 | 0.51 | 0.58 | 0.52 | 0.51 | 0.66 | 0.50 | 0.50 | 0.61 | 0.51 | 0.50 |

Table 12: **Membership Inference for OOD Adaptations using Pythia 70M.** We present the AUC scores obtained with reference, and Min-K% MIAs for the Pythia 70M model adapted on different datasets with $\varepsilon \in \{0.1, 8, \infty\}$.

| MIA | Adaptation | Samsum $\varepsilon=\infty$ | $\varepsilon=8$ | $\varepsilon=0.1$ | German Wiki $\varepsilon=\infty$ | $\varepsilon=8$ | $\varepsilon=0.1$ | Average $\varepsilon=\infty$ | $\varepsilon=8$ | $\varepsilon=0.1$ |
|---|---|---|---|---|---|---|---|---|---|---|
| RMIA (shadow) | Prefix | 0.53 | 0.62 | 0.51 | 0.60 | 0.63 | 0.56 | 0.57 | 0.63 | 0.53 |
| | LoRA | 0.68 | 0.58 | 0.55 | 0.59 | 0.61 | 0.57 | 0.63 | 0.59 | 0.56 |
| | Full Fine-Tune | 0.98 | 0.92 | 0.63 | 0.98 | 0.97 | 0.71 | 0.98 | 0.94 | 0.67 |
| | Head Fine-Tune | 1.00 | 0.93 | 0.73 | 0.95 | 0.93 | 0.77 | 0.97 | 0.93 | 0.75 |
| | Average | 0.80 | 0.76 | 0.61 | 0.78 | 0.78 | 0.65 | 0.79 | 0.77 | 0.63 |
| RMIA (Pythia 1B) | Prefix | 0.51 | 0.51 | 0.51 | 0.50 | 0.51 | 0.50 | 0.51 | 0.51 | 0.51 |
| | LoRA | 0.51 | 0.51 | 0.52 | 0.51 | 0.51 | 0.51 | 0.51 | 0.51 | 0.51 |
| | Full Fine-Tune | 0.52 | 0.53 | 0.52 | 0.53 | 0.55 | 0.50 | 0.53 | 0.54 | 0.51 |
| | Head Fine-Tune | 0.67 | 0.54 | 0.50 | 0.52 | 0.54 | 0.51 | 0.59 | 0.54 | 0.51 |
| | Average | 0.55 | 0.52 | 0.51 | 0.51 | 0.53 | 0.51 | 0.53 | 0.53 | 0.51 |
| Reference (Pythia 1B) | Prefix | 0.51 | 0.52 | 0.51 | 0.49 | 0.50 | 0.49 | 0.50 | 0.51 | 0.50 |
| | LoRA | 0.51 | 0.51 | 0.52 | 0.50 | 0.50 | 0.50 | 0.50 | 0.51 | 0.51 |
| | Full Fine-Tune | 0.52 | 0.53 | 0.52 | 0.51 | 0.53 | 0.49 | 0.52 | 0.53 | 0.51 |
| | Head Fine-Tune | 0.67 | 0.55 | 0.51 | 0.50 | 0.52 | 0.50 | 0.59 | 0.53 | 0.50 |
| | Average | 0.55 | 0.53 | 0.51 | 0.50 | 0.51 | 0.49 | 0.53 | 0.52 | 0.50 |
| Min-K% | Prefix | 0.51 | 0.51 | 0.50 | 0.51 | 0.52 | 0.51 | 0.51 | 0.51 | 0.51 |
| | LoRA | 0.51 | 0.50 | 0.51 | 0.52 | 0.52 | 0.52 | 0.51 | 0.51 | 0.51 |
| | Full Fine-Tune | 0.51 | 0.52 | 0.51 | 0.53 | 0.54 | 0.52 | 0.52 | 0.53 | 0.51 |
| | Head Fine-Tune | 0.64 | 0.53 | 0.50 | 0.53 | 0.54 | 0.52 | 0.58 | 0.54 | 0.51 |
| | Average | 0.54 | 0.52 | 0.50 | 0.52 | 0.53 | 0.52 | 0.53 | 0.52 | 0.51 |

Table 13: **Membership Inference for IID Adaptations using Pythia 70M.** We present the AUC scores obtained with reference, and Min-K% MIAs for the Pythia 70M model adapted on different datasets with $\varepsilon \in \{0.1, 8, \infty\}$.

| MIA | Adaptation | Pile Bookcorpus2 Val $\varepsilon=\infty$ | $\varepsilon=8$ | $\varepsilon=0.1$ | Pile Bookcorpus2 Train $\varepsilon=\infty$ | $\varepsilon=8$ | $\varepsilon=0.1$ | Pile Github Val $\varepsilon=\infty$ | $\varepsilon=8$ | $\varepsilon=0.1$ | Pile Enron Val $\varepsilon=\infty$ | $\varepsilon=8$ | $\varepsilon=0.1$ | Average $\varepsilon=\infty$ | $\varepsilon=8$ | $\varepsilon=0.1$ |
|---|---|---|---|---|---|---|---|---|---|---|---|---|---|---|---|---|
| RMIA (shadow) | Prefix | 0.63 | 0.66 | 0.50 | 0.60 | 0.68 | 0.52 | 0.57 | 0.73 | 0.56 | 0.65 | 0.72 | 0.53 | 0.61 | 0.70 | 0.53 |
| | LoRA | 0.57 | 0.80 | 0.54 | 0.81 | 0.80 | 0.55 | 0.84 | 0.83 | 0.55 | 0.85 | 0.63 | 0.50 | 0.77 | 0.77 | 0.54 |
| | Full Fine-Tune | 0.99 | 0.92 | 0.59 | 0.99 | 0.92 | 0.59 | 1.00 | 0.97 | 0.58 | 0.99 | 0.97 | 0.58 | 0.99 | 0.95 | 0.58 |
| | Head Fine-Tune | 0.97 | 0.94 | 0.72 | 1.00 | 0.95 | 0.70 | 1.00 | 0.98 | 0.76 | 1.00 | 0.97 | 0.76 | 0.99 | 0.96 | 0.73 |
| | Average | 0.79 | 0.83 | 0.59 | 0.85 | 0.84 | 0.59 | 0.85 | 0.88 | 0.61 | 0.87 | 0.82 | 0.59 | 0.84 | 0.84 | 0.60 |
| RMIA (Pythia 1B) | Prefix | 0.50 | 0.50 | 0.50 | 0.51 | 0.52 | 0.51 | 0.51 | 0.51 | 0.51 | 0.50 | 0.50 | 0.50 | 0.51 | 0.51 | 0.51 |
| | LoRA | 0.50 | 0.50 | 0.50 | 0.51 | 0.51 | 0.51 | 0.51 | 0.51 | 0.51 | 0.50 | 0.50 | 0.49 | 0.51 | 0.51 | 0.50 |
| | Full Fine-Tune | 0.52 | 0.52 | 0.50 | 0.53 | 0.53 | 0.51 | 0.87 | 0.52 | 0.51 | 0.51 | 0.52 | 0.50 | 0.61 | 0.52 | 0.51 |
| | Head Fine-Tune | 0.51 | 0.52 | 0.50 | 0.69 | 0.54 | 0.50 | 0.64 | 0.55 | 0.51 | 0.73 | 0.52 | 0.50 | 0.64 | 0.53 | 0.50 |
| | Average | 0.51 | 0.51 | 0.50 | 0.56 | 0.52 | 0.51 | 0.63 | 0.52 | 0.51 | 0.56 | 0.51 | 0.50 | 0.56 | 0.52 | 0.50 |
| Reference (Pythia 1B) | Prefix | 0.50 | 0.50 | 0.50 | 0.51 | 0.51 | 0.51 | 0.49 | 0.50 | 0.49 | 0.50 | 0.50 | 0.50 | 0.50 | 0.50 | 0.50 |
| | LoRA | 0.50 | 0.51 | 0.50 | 0.51 | 0.51 | 0.51 | 0.50 | 0.50 | 0.50 | 0.49 | 0.49 | 0.49 | 0.50 | 0.50 | 0.50 |
| | Full Fine-Tune | 0.52 | 0.52 | 0.50 | 0.52 | 0.53 | 0.51 | 0.81 | 0.51 | 0.50 | 0.50 | 0.51 | 0.49 | 0.59 | 0.52 | 0.50 |
| | Head Fine-Tune | 0.51 | 0.52 | 0.50 | 0.66 | 0.54 | 0.51 | 0.60 | 0.54 | 0.50 | 0.67 | 0.51 | 0.48 | 0.61 | 0.53 | 0.50 |
| | Average | 0.51 | 0.51 | 0.50 | 0.55 | 0.52 | 0.51 | 0.60 | 0.51 | 0.50 | 0.54 | 0.50 | 0.49 | 0.55 | 0.51 | 0.50 |
| Min-K% | Prefix | 0.50 | 0.50 | 0.49 | 0.51 | 0.51 | 0.51 | 0.52 | 0.52 | 0.52 | 0.52 | 0.52 | 0.52 | 0.51 | 0.51 | 0.51 |
| | LoRA | 0.49 | 0.50 | 0.50 | 0.51 | 0.51 | 0.51 | 0.52 | 0.52 | 0.52 | 0.51 | 0.51 | 0.51 | 0.51 | 0.51 | 0.51 |
| | Full Fine-Tune | 0.50 | 0.51 | 0.49 | 0.52 | 0.52 | 0.51 | 0.75 | 0.53 | 0.52 | 0.52 | 0.52 | 0.51 | 0.57 | 0.52 | 0.51 |
| | Head Fine-Tune | 0.50 | 0.51 | 0.49 | 0.64 | 0.53 | 0.51 | 0.62 | 0.55 | 0.51 | 0.76 | 0.53 | 0.50 | 0.63 | 0.53 | 0.50 |
| | Average | 0.50 | 0.50 | 0.49 | 0.54 | 0.52 | 0.51 | 0.60 | 0.53 | 0.52 | 0.58 | 0.52 | 0.51 | 0.55 | 0.52 | 0.51 |

exposure when only considering the first $N$ tokens. This highlights that the prefix itself is the main source of privacy leakage.

Table 14: **Membership Inference for OOD Adaptations using GPT-Neo 1.3B.** We present the AUC scores obtained with reference, and Min-K% MIAs for the GPT-Neo 1.3B model adapted on different datasets with $\varepsilon \in \{0.1, 8, \infty\}$.

| MIA | Adaptation | Samsum $\varepsilon=\infty$ | $\varepsilon=8$ | $\varepsilon=0.1$ | German Wiki $\varepsilon=\infty$ | $\varepsilon=8$ | $\varepsilon=0.1$ | Average $\varepsilon=\infty$ | $\varepsilon=8$ | $\varepsilon=0.1$ |
|---|---|---|---|---|---|---|---|---|---|---|
| RMIA (shadow) | Prefix | 0.50 | 0.51 | 0.50 | 0.98 | 0.50 | 0.51 | 0.74 | 0.50 | 0.51 |
| | LoRA | 0.53 | 0.85 | 0.51 | 0.55 | 0.89 | 0.50 | 0.54 | 0.87 | 0.51 |
| | Full Fine-Tune | 1.00 | 1.00 | 0.80 | 1.00 | 1.00 | 0.83 | 1.00 | 1.00 | 0.82 |
| | Head Fine-Tune | 0.93 | 1.00 | 0.81 | 0.93 | 1.00 | 0.85 | 0.93 | 1.00 | 0.83 |
| | Average | 0.74 | 0.84 | 0.66 | 0.86 | 0.85 | 0.68 | 0.80 | 0.84 | 0.67 |
| RMIA (Pythia 1B) | Prefix | 0.51 | 0.51 | 0.51 | 0.71 | 0.50 | 0.49 | 0.61 | 0.51 | 0.50 |
| | LoRA | 0.50 | 0.50 | 0.50 | 0.51 | 0.52 | 0.51 | 0.51 | 0.51 | 0.51 |
| | Full Fine-Tune | 0.58 | 0.56 | 0.51 | 0.74 | 0.63 | 0.51 | 0.66 | 0.60 | 0.51 |
| | Head Fine-Tune | 0.50 | 0.55 | 0.50 | 0.51 | 0.60 | 0.51 | 0.51 | 0.58 | 0.51 |
| | Average | 0.53 | 0.53 | 0.51 | 0.61 | 0.56 | 0.51 | 0.57 | 0.55 | 0.51 |
| Reference (Pythia 1B) | Prefix | 0.50 | 0.49 | 0.49 | 0.62 | 0.48 | 0.47 | 0.56 | 0.49 | 0.48 |
| | LoRA | 0.49 | 0.51 | 0.49 | 0.51 | 0.52 | 0.51 | 0.50 | 0.52 | 0.50 |
| | Full Fine-Tune | 0.59 | 0.57 | 0.49 | 0.74 | 0.61 | 0.49 | 0.66 | 0.59 | 0.49 |
| | Head Fine-Tune | 0.50 | 0.56 | 0.49 | 0.51 | 0.60 | 0.50 | 0.50 | 0.58 | 0.50 |
| | Average | 0.52 | 0.53 | 0.49 | 0.59 | 0.55 | 0.49 | 0.56 | 0.54 | 0.49 |
| Min-K% | Prefix | 0.52 | 0.50 | 0.50 | 0.65 | 0.50 | 0.51 | 0.58 | 0.50 | 0.51 |
| | LoRA | 0.51 | 0.51 | 0.51 | 0.52 | 0.53 | 0.52 | 0.52 | 0.52 | 0.52 |
| | Full Fine-Tune | 0.55 | 0.55 | 0.51 | 0.59 | 0.57 | 0.53 | 0.57 | 0.56 | 0.52 |
| | Head Fine-Tune | 0.51 | 0.54 | 0.51 | 0.53 | 0.56 | 0.52 | 0.52 | 0.55 | 0.52 |
| | Average | 0.52 | 0.52 | 0.51 | 0.57 | 0.54 | 0.52 | 0.55 | 0.53 | 0.51 |

Table 15: **Membership Inference for IID Adaptations using GPT-Neo 1.3B.** We present the AUC scores obtained with reference, and Min-K% MIAs for the GPT-Neo 1.3B model adapted on different datasets with $\varepsilon \in \{0.1, 8, \infty\}$.

| MIA | Adaptation | Pile Bookcorpus2 Val $\varepsilon=\infty$ | $\varepsilon=8$ | $\varepsilon=0.1$ | Pile Bookcorpus2 Train $\varepsilon=\infty$ | $\varepsilon=8$ | $\varepsilon=0.1$ | Pile Github Val $\varepsilon=\infty$ | $\varepsilon=8$ | $\varepsilon=0.1$ | Pile Enron Val $\varepsilon=\infty$ | $\varepsilon=8$ | $\varepsilon=0.1$ | Average $\varepsilon=\infty$ | $\varepsilon=8$ | $\varepsilon=0.1$ |
|---|---|---|---|---|---|---|---|---|---|---|---|---|---|---|---|---|
| RMIA (shadow) | Prefix | 0.62 | 0.51 | 0.52 | 0.78 | 0.51 | 0.50 | 0.95 | 0.50 | 0.51 | 0.79 | 0.57 | 0.55 | 0.78 | 0.52 | 0.52 |
| | LoRA | 0.53 | 0.81 | 0.52 | 0.54 | 0.81 | 0.51 | 0.55 | 0.89 | 0.50 | 0.57 | 0.64 | 0.62 | 0.55 | 0.79 | 0.54 |
| | Full Fine-Tune | 1.00 | 1.00 | 0.65 | 1.00 | 1.00 | 0.64 | 1.00 | 0.71 | 0.75 | 1.00 | 1.00 | 0.62 | 1.00 | 0.93 | 0.67 |
| | Head Fine-Tune | 0.96 | 1.00 | 0.70 | 1.00 | 1.00 | 0.70 | 1.00 | 1.00 | 0.87 | 1.00 | 1.00 | 0.65 | 0.99 | 1.00 | 0.73 |
| | Average | 0.78 | 0.83 | 0.60 | 0.83 | 0.83 | 0.59 | 0.87 | 0.78 | 0.66 | 0.84 | 0.80 | 0.61 | 0.83 | 0.81 | 0.61 |
| RMIA (Pythia 1B) | Prefix | 0.51 | 0.51 | 0.51 | 0.76 | 0.51 | 0.50 | 0.68 | 0.50 | 0.51 | 0.80 | 0.57 | 0.56 | 0.69 | 0.52 | 0.52 |
| | LoRA | 0.51 | 0.53 | 0.51 | 0.50 | 0.50 | 0.50 | 0.51 | 0.54 | 0.53 | 0.56 | 0.56 | 0.56 | 0.52 | 0.53 | 0.53 |
| | Full Fine-Tune | 0.72 | 0.62 | 0.54 | 0.71 | 0.62 | 0.51 | 0.91 | 0.54 | 0.53 | 0.70 | 0.67 | 0.57 | 0.76 | 0.61 | 0.53 |
| | Head Fine-Tune | 0.52 | 0.60 | 0.50 | 1.00 | 0.61 | 0.51 | 0.98 | 0.61 | 0.53 | 0.98 | 0.65 | 0.57 | 0.87 | 0.62 | 0.53 |
| | Average | 0.56 | 0.56 | 0.51 | 0.74 | 0.56 | 0.51 | 0.77 | 0.55 | 0.52 | 0.76 | 0.61 | 0.57 | 0.71 | 0.57 | 0.53 |
| Reference (Pythia 1B) | Prefix | 0.51 | 0.51 | 0.52 | 0.72 | 0.51 | 0.50 | 0.61 | 0.48 | 0.48 | 0.74 | 0.44 | 0.43 | 0.65 | 0.48 | 0.48 |
| | LoRA | 0.51 | 0.53 | 0.51 | 0.48 | 0.49 | 0.48 | 0.51 | 0.51 | 0.48 | 0.58 | 0.59 | 0.58 | 0.52 | 0.53 | 0.51 |
| | Full Fine-Tune | 0.72 | 0.62 | 0.52 | 0.71 | 0.62 | 0.51 | 0.89 | 0.50 | 0.50 | 0.74 | 0.65 | 0.56 | 0.77 | 0.60 | 0.52 |
| | Head Fine-Tune | 0.52 | 0.61 | 0.51 | 1.00 | 0.62 | 0.51 | 0.97 | 0.61 | 0.51 | 0.98 | 0.66 | 0.57 | 0.87 | 0.63 | 0.53 |
| | Average | 0.57 | 0.57 | 0.51 | 0.73 | 0.56 | 0.50 | 0.74 | 0.53 | 0.49 | 0.76 | 0.58 | 0.53 | 0.70 | 0.56 | 0.51 |
| Min-K% | Prefix | 0.50 | 0.51 | 0.51 | 0.65 | 0.52 | 0.50 | 0.67 | 0.52 | 0.53 | 0.77 | 0.55 | 0.55 | 0.65 | 0.53 | 0.52 |
| | LoRA | 0.50 | 0.50 | 0.50 | 0.50 | 0.50 | 0.50 | 0.52 | 0.54 | 0.53 | 0.57 | 0.57 | 0.57 | 0.52 | 0.53 | 0.52 |
| | Full Fine-Tune | 0.54 | 0.54 | 0.50 | 0.54 | 0.54 | 0.50 | 0.62 | 0.54 | 0.53 | 0.60 | 0.59 | 0.56 | 0.57 | 0.55 | 0.52 |
| | Head Fine-Tune | 0.50 | 0.52 | 0.49 | 0.91 | 0.53 | 0.50 | 0.74 | 0.54 | 0.53 | 0.87 | 0.59 | 0.57 | 0.76 | 0.55 | 0.52 |
| | Average | 0.51 | 0.52 | 0.50 | 0.65 | 0.52 | 0.50 | 0.64 | 0.54 | 0.53 | 0.70 | 0.58 | 0.56 | 0.63 | 0.54 | 0.52 |

Table 16: **Membership Inference for OOD Adaptations using GPT-Neo 125M.** We present the AUC scores obtained with reference, and Min-K% MIAs for the GPT-Neo 125M model adapted on different datasets with $\varepsilon \in \{0.1, 8, \infty\}$.

| MIA | Adaptation | Samsum $\varepsilon=\infty$ | $\varepsilon=8$ | $\varepsilon=0.1$ | German Wiki $\varepsilon=\infty$ | $\varepsilon=8$ | $\varepsilon=0.1$ | Average $\varepsilon=\infty$ | $\varepsilon=8$ | $\varepsilon=0.1$ |
|---|---|---|---|---|---|---|---|---|---|---|
| RMIA (shadow) | Prefix | 0.68 | 0.51 | 0.50 | 0.73 | 0.51 | 0.52 | 0.70 | 0.51 | 0.51 |
| | LoRA | 0.84 | 0.63 | 0.59 | 0.51 | 0.50 | 0.50 | 0.67 | 0.56 | 0.55 |
| | Full Fine-Tune | 1.00 | 0.99 | 0.72 | 1.00 | 0.52 | 0.79 | 1.00 | 0.75 | 0.75 |
| | Head Fine-Tune | 1.00 | 0.94 | 0.79 | 1.00 | 1.00 | 0.87 | 1.00 | 0.97 | 0.83 |
| | Average | 0.88 | 0.77 | 0.65 | 0.81 | 0.63 | 0.67 | 0.84 | 0.70 | 0.66 |
| RMIA (Pythia 1B) | Prefix | 0.52 | 0.51 | 0.51 | 0.55 | 0.50 | 0.50 | 0.54 | 0.50 | 0.50 |
| | LoRA | 0.51 | 0.51 | 0.51 | 0.50 | 0.50 | 0.50 | 0.50 | 0.51 | 0.51 |
| | Full Fine-Tune | 1.00 | 0.53 | 0.52 | 1.00 | 0.50 | 0.50 | 1.00 | 0.52 | 0.51 |
| | Head Fine-Tune | 1.00 | 0.51 | 0.50 | 0.54 | 0.57 | 0.51 | 0.77 | 0.54 | 0.51 |
| | Average | 0.76 | 0.52 | 0.51 | 0.65 | 0.52 | 0.50 | 0.70 | 0.52 | 0.51 |
| Reference (Pythia 1B) | Prefix | 0.52 | 0.49 | 0.49 | 0.51 | 0.48 | 0.48 | 0.51 | 0.49 | 0.49 |
| | LoRA | 0.51 | 0.51 | 0.51 | 0.49 | 0.49 | 0.49 | 0.50 | 0.50 | 0.50 |
| | Full Fine-Tune | 1.00 | 0.53 | 0.50 | 1.00 | 0.49 | 0.49 | 1.00 | 0.51 | 0.50 |
| | Head Fine-Tune | 1.00 | 0.51 | 0.50 | 0.53 | 0.55 | 0.49 | 0.76 | 0.53 | 0.50 |
| | Average | 0.76 | 0.51 | 0.50 | 0.63 | 0.50 | 0.49 | 0.69 | 0.51 | 0.49 |
| Min-K% | Prefix | 0.54 | 0.51 | 0.51 | 0.55 | 0.50 | 0.49 | 0.54 | 0.50 | 0.50 |
| | LoRA | 0.51 | 0.51 | 0.51 | 0.52 | 0.52 | 0.52 | 0.52 | 0.52 | 0.52 |
| | Full Fine-Tune | 1.00 | 0.53 | 0.52 | 1.00 | 0.52 | 0.52 | 1.00 | 0.53 | 0.52 |
| | Head Fine-Tune | 1.00 | 0.51 | 0.51 | 0.54 | 0.55 | 0.52 | 0.77 | 0.53 | 0.52 |
| | Average | 0.76 | 0.52 | 0.51 | 0.65 | 0.52 | 0.51 | 0.71 | 0.52 | 0.51 |

Table 17: **Membership Inference for IID Adaptations using GPT-Neo 125M.** We present the AUC scores obtained with reference, and Min-K% MIAs for the GPT-Neo 125M model adapted on different datasets with $\varepsilon \in \{0.1, 8, \infty\}$.

| MIA | Adaptation | Pile Bookcorpus2 Val | | | Pile Bookcorpus2 Train | | | Pile Github Val | | | Pile Enron Val | | | Average | | |
|---|---|---|---|---|---|---|---|---|---|---|---|---|---|---|---|---|
| | | $\varepsilon=\infty$ | $\varepsilon=8$ | $\varepsilon=0.1$ | $\varepsilon=\infty$ | $\varepsilon=8$ | $\varepsilon=0.1$ | $\varepsilon=\infty$ | $\varepsilon=8$ | $\varepsilon=0.1$ | $\varepsilon=\infty$ | $\varepsilon=8$ | $\varepsilon=0.1$ | $\varepsilon=\infty$ | $\varepsilon=8$ | $\varepsilon=0.1$ |
| RMIA (shadow) | Prefix | 0.68 | 0.52 | 0.51 | 0.52 | 0.53 | 0.51 | 0.77 | 0.50 | 0.50 | 0.76 | 0.57 | 0.53 | 0.68 | 0.53 | 0.51 |
| | LoRA | 0.52 | 0.51 | 0.50 | 1.00 | 0.51 | 0.51 | 1.00 | 0.50 | 0.51 | 1.00 | 0.50 | 0.50 | 0.88 | 0.51 | 0.50 |
| | Full Fine-Tune | 1.00 | 0.51 | 0.68 | 1.00 | 0.97 | 0.58 | 0.98 | 0.97 | 0.68 | 1.00 | 0.98 | 0.66 | 1.00 | 0.86 | 0.65 |
| | Head Fine-Tune | 1.00 | 1.00 | 0.70 | 1.00 | 1.00 | 0.66 | 0.96 | 1.00 | 0.87 | 1.00 | 1.00 | 0.70 | 0.99 | 1.00 | 0.74 |
| | Average | 0.80 | 0.64 | 0.60 | 0.88 | 0.75 | 0.57 | 0.93 | 0.74 | 0.64 | 0.94 | 0.76 | 0.60 | 0.89 | 0.72 | 0.60 |
| RMIA (Pythia 1B) | Prefix | 0.52 | 0.50 | 0.50 | 0.52 | 0.51 | 0.50 | 0.54 | 0.53 | 0.53 | 0.55 | 0.54 | 0.54 | 0.54 | 0.52 | 0.52 |
| | LoRA | 0.50 | 0.50 | 0.50 | 0.94 | 0.51 | 0.51 | 0.72 | 0.52 | 0.52 | 0.90 | 0.56 | 0.56 | 0.77 | 0.52 | 0.52 |
| | Full Fine-Tune | 1.00 | 0.50 | 0.50 | 1.00 | 0.54 | 0.52 | 0.67 | 0.54 | 0.53 | 1.00 | 0.58 | 0.56 | 0.92 | 0.54 | 0.53 |
| | Head Fine-Tune | 1.00 | 0.56 | 0.50 | 1.00 | 0.57 | 0.51 | 0.95 | 0.57 | 0.53 | 1.00 | 0.59 | 0.56 | 0.99 | 0.57 | 0.52 |
| | Average | 0.75 | 0.52 | 0.50 | 0.87 | 0.53 | 0.51 | 0.72 | 0.54 | 0.53 | 0.86 | 0.57 | 0.56 | 0.80 | 0.54 | 0.52 |
| Reference (Pythia 1B) | Prefix | 0.52 | 0.51 | 0.51 | 0.52 | 0.51 | 0.50 | 0.50 | 0.48 | 0.48 | 0.55 | 0.43 | 0.43 | 0.52 | 0.48 | 0.48 |
| | LoRA | 0.51 | 0.51 | 0.51 | 0.92 | 0.51 | 0.51 | 0.70 | 0.50 | 0.50 | 0.87 | 0.53 | 0.53 | 0.75 | 0.51 | 0.51 |
| | Full Fine-Tune | 1.00 | 0.51 | 0.51 | 1.00 | 0.54 | 0.52 | 0.58 | 0.52 | 0.49 | 1.00 | 0.56 | 0.55 | 0.89 | 0.53 | 0.52 |
| | Head Fine-Tune | 1.00 | 0.56 | 0.51 | 1.00 | 0.57 | 0.51 | 0.92 | 0.55 | 0.51 | 1.00 | 0.57 | 0.53 | 0.98 | 0.56 | 0.51 |
| | Average | 0.76 | 0.52 | 0.51 | 0.86 | 0.53 | 0.51 | 0.67 | 0.51 | 0.50 | 0.85 | 0.52 | 0.51 | 0.79 | 0.52 | 0.51 |
| Min-K% | Prefix | 0.53 | 0.50 | 0.50 | 0.52 | 0.51 | 0.50 | 0.55 | 0.54 | 0.53 | 0.56 | 0.54 | 0.54 | 0.54 | 0.52 | 0.52 |
| | LoRA | 0.50 | 0.50 | 0.50 | 0.70 | 0.50 | 0.50 | 0.60 | 0.52 | 0.52 | 0.74 | 0.56 | 0.56 | 0.63 | 0.52 | 0.52 |
| | Full Fine-Tune | 1.00 | 0.50 | 0.50 | 1.00 | 0.53 | 0.51 | 0.80 | 0.55 | 0.53 | 1.00 | 0.58 | 0.56 | 0.95 | 0.54 | 0.53 |
| | Head Fine-Tune | 1.00 | 0.52 | 0.50 | 1.00 | 0.53 | 0.50 | 0.94 | 0.55 | 0.53 | 1.00 | 0.58 | 0.56 | 0.98 | 0.55 | 0.52 |
| | Average | 0.75 | 0.50 | 0.50 | 0.80 | 0.52 | 0.51 | 0.72 | 0.54 | 0.53 | 0.83 | 0.56 | 0.55 | 0.78 | 0.53 | 0.52 |

Table 18: **Membership Inference for OOD Adaptations using Olmo 2 0425 1B.** We present the AUC scores obtained with reference, and Min-K% MIAs for the Olmo 2 0425 1B model adapted on different datasets with $\varepsilon \in \{0.1, 8, \infty\}$.

| MIA | Adaptation | Samsum | | | German Wiki | | | Average | | |
|---|---|---|---|---|---|---|---|---|---|---|
| | | $\varepsilon=\infty$ | $\varepsilon=8$ | $\varepsilon=0.1$ | $\varepsilon=\infty$ | $\varepsilon=8$ | $\varepsilon=0.1$ | $\varepsilon=\infty$ | $\varepsilon=8$ | $\varepsilon=0.1$ |
| RMIA (shadow) | Prefix | 0.95 | 0.83 | 0.58 | 0.96 | 0.87 | 0.54 | 0.96 | 0.85 | 0.56 |
| | LoRA | 0.87 | 0.93 | 0.68 | 0.97 | 0.95 | 0.76 | 0.92 | 0.94 | 0.72 |
| | Full fine-tune | 1.00 | 0.81 | 0.58 | 1.00 | 0.80 | 0.57 | 1.00 | 0.80 | 0.58 |
| | Head fine-tune | 0.99 | 0.79 | 0.63 | 0.99 | 0.77 | 0.60 | 0.99 | 0.57 | 0.60 |
| | Average | 0.94 | 0.85 | 0.62 | 0.98 | 0.85 | 0.70 | 0.96 | 0.85 | 0.65 |
| RMIA (Olmo 2 0425 1B) | Prefix | 0.54 | 0.52 | 0.50 | 0.60 | 0.54 | 0.50 | 0.57 | 0.53 | 0.50 |
| | LoRA | 0.51 | 0.52 | 0.50 | 0.56 | 0.56 | 0.51 | 0.54 | 0.54 | 0.50 |
| | Full fine-tune | 0.78 | 0.50 | 0.50 | 1.00 | 0.53 | 0.51 | 0.89 | 0.53 | 0.50 |
| | Head fine-tune | 0.58 | 0.50 | 0.50 | 0.60 | 0.52 | 0.51 | 0.59 | 0.51 | 0.50 |
| | Average | 0.61 | 0.51 | 0.50 | 0.67 | 0.54 | 0.51 | 0.66 | 0.53 | 0.50 |
| Reference (Olmo 2 0425 1B) | Prefix | 0.54 | 0.52 | 0.50 | 0.60 | 0.54 | 0.50 | 0.57 | 0.53 | 0.50 |
| | LoRA | 0.51 | 0.52 | 0.50 | 0.57 | 0.56 | 0.51 | 0.54 | 0.54 | 0.50 |
| | Full fine-tune | 0.79 | 0.51 | 0.50 | 1.00 | 0.53 | 0.50 | 0.89 | 0.53 | 0.50 |
| | Head fine-tune | 0.61 | 0.50 | 0.50 | 0.68 | 0.52 | 0.51 | 0.64 | 0.51 | 0.50 |
| | Average | 0.61 | 0.51 | 0.50 | 0.71 | 0.54 | 0.50 | 0.66 | 0.53 | 0.50 |
| Min-K% | Prefix | 0.52 | 0.51 | 0.50 | 0.53 | 0.52 | 0.51 | 0.52 | 0.51 | 0.51 |
| | LoRA | 0.50 | 0.51 | 0.50 | 0.52 | 0.52 | 0.51 | 0.51 | 0.51 | 0.51 |
| | Full fine-tune | 0.67 | 0.50 | 0.50 | 0.75 | 0.51 | 0.51 | 0.71 | 0.51 | 0.50 |
| | Head fine-tune | 0.56 | 0.50 | 0.50 | 0.60 | 0.51 | 0.51 | 0.58 | 0.51 | 0.51 |
| | Average | 0.56 | 0.50 | 0.50 | 0.60 | 0.52 | 0.51 | 0.58 | 0.51 | 0.50 |

Table 19: **Membership Inference for IID Adaptations using Olmo 2 0425 1B.** We present the AUC scores obtained with reference, and Min-K% MIAs for the Olmo 2 0425 1B model adapted on different datasets with $\varepsilon \in \{0.1, 8, \infty\}$.

| MIA | Adaptation | Dolmino Wiki | | |
|---|---|---|---|---|
| | | $\varepsilon=\infty$ | $\varepsilon=8$ | $\varepsilon=0.1$ |
| RMIA (shadow) | Prefix | 0.95 | 0.89 | 0.63 |
| | LoRA | 0.93 | 0.80 | 0.57 |
| | Full fine-tune | 1.00 | 0.91 | 0.63 |
| | Head fine-tune | 0.92 | 0.90 | 0.70 |
| | Average | 0.96 | 0.88 | 0.63 |
| RMIA (Olmo 2 0425 1B) | Prefix | 0.59 | 0.56 | 0.51 |
| | LoRA | 0.55 | 0.56 | 0.53 |
| | Full fine-tune | 0.87 | 0.56 | 0.51 |
| | Head fine-tune | 0.62 | 0.64 | 0.51 |
| | Average | 0.65 | 0.58 | 0.52 |
| Reference (Olmo 2 0425 1B) | Prefix | 0.59 | 0.56 | 0.52 |
| | LoRA | 0.55 | 0.56 | 0.53 |
| | Full fine-tune | 0.86 | 0.56 | 0.51 |
| | Head fine-tune | 0.65 | 0.63 | 0.51 |
| | Average | 0.67 | 0.58 | 0.52 |
| Min-K% | Prefix | 0.51 | 0.50 | 0.50 |
| | LoRA | 0.51 | 0.50 | 0.51 |
| | Full fine-tune | 0.58 | 0.50 | 0.50 |
| | Head fine-tune | 0.56 | 0.51 | 0.51 |
| | Average | 0.53 | 0.50 | 0.50 |

Table 20: **Membership Inference for OOD Adaptations using Olmo 1B.** We present the AUC scores obtained with reference, and Min-K% MIAs for the Olmo 1B model adapted on different datasets with $\varepsilon \in \{0.1, 8, \infty\}$.

| MIA | Adaptation | Samsum $\varepsilon = \infty$ | $\varepsilon = 8$ | $\varepsilon = 0.1$ | German Wiki $\varepsilon = \infty$ | $\varepsilon = 8$ | $\varepsilon = 0.1$ | Average $\varepsilon = \infty$ | $\varepsilon = 8$ | $\varepsilon = 0.1$ |
|---|---|---|---|---|---|---|---|---|---|---|
| RMIA (shadow) | Prefix | 1.00 | 0.95 | 0.72 | 1.00 | 0.95 | 0.67 | 1.00 | 0.95 | 0.70 |
| | LoRA | 0.98 | 0.97 | 0.69 | 1.00 | 0.98 | 0.73 | 0.99 | 0.98 | 0.71 |
| | Full fine-tune | 1.00 | 1.00 | 0.78 | 1.00 | 0.87 | 0.58 | 1.00 | 0.94 | 0.68 |
| | Head fine-tune | 0.98 | 0.78 | 0.50 | 0.99 | 0.73 | 0.76 | 0.98 | 0.76 | 0.63 |
| | Average | 0.99 | 0.92 | 0.67 | 1.00 | 0.89 | 0.69 | 1.00 | 0.90 | 0.68 |
| RMIA (Olmo 1B) | Prefix | 0.60 | 0.52 | 0.51 | 0.74 | 0.57 | 0.50 | 0.67 | 0.54 | 0.50 |
| | LoRA | 0.53 | 0.53 | 0.51 | 0.64 | 0.58 | 0.50 | 0.59 | 0.56 | 0.51 |
| | Full fine-tune | 0.98 | 0.54 | 0.51 | 1.00 | 0.54 | 0.52 | 0.99 | 0.54 | 0.51 |
| | Head fine-tune | 0.69 | 0.50 | 0.50 | 0.78 | 0.53 | 0.50 | 0.73 | 0.52 | 0.50 |
| | Average | 0.70 | 0.53 | 0.51 | 0.79 | 0.56 | 0.51 | 0.75 | 0.54 | 0.51 |
| Reference (Olmo 1B) | Prefix | 0.59 | 0.52 | 0.51 | 0.75 | 0.56 | 0.50 | 0.67 | 0.54 | 0.50 |
| | LoRA | 0.53 | 0.53 | 0.51 | 0.65 | 0.59 | 0.50 | 0.59 | 0.56 | 0.50 |
| | Full fine-tune | 0.99 | 0.54 | 0.51 | 1.00 | 0.54 | 0.52 | 0.99 | 0.54 | 0.51 |
| | Head fine-tune | 0.66 | 0.50 | 0.50 | 0.68 | 0.53 | 0.50 | 0.67 | 0.52 | 0.50 |
| | Average | 0.69 | 0.52 | 0.51 | 0.77 | 0.56 | 0.50 | 0.73 | 0.54 | 0.51 |
| Min-K% | Prefix | 0.55 | 0.52 | 0.50 | 0.57 | 0.53 | 0.51 | 0.56 | 0.52 | 0.51 |
| | LoRA | 0.51 | 0.52 | 0.50 | 0.54 | 0.52 | 0.51 | 0.53 | 0.52 | 0.51 |
| | Full fine-tune | 0.54 | 0.52 | 0.50 | 0.58 | 0.51 | 0.51 | 0.56 | 0.52 | 0.51 |
| | Head fine-tune | 0.65 | 0.50 | 0.50 | 0.56 | 0.51 | 0.51 | 0.53 | 0.51 | 0.51 |
| | Average | 0.56 | 0.51 | 0.50 | 0.56 | 0.52 | 0.51 | 0.54 | 0.52 | 0.51 |

Table 21: **Membership Inference for IID Adaptations using Olmo 1B.** We present the AUC scores obtained with reference, and Min-K% MIAs for the Olmo 1B model adapted on different datasets with $\varepsilon \in \{0.1, 8, \infty\}$.

| MIA | Adaptation | Dolmino Wiki $\varepsilon = \infty$ | $\varepsilon = 8$ | $\varepsilon = 0.1$ |
|---|---|---|---|---|
| RMIA (shadow) | Prefix | 0.97 | 0.89 | 0.54 |
| | LoRA | 0.93 | 0.85 | 0.52 |
| | Full fine-tune | 0.98 | 0.93 | 0.61 |
| | Head fine-tune | 0.90 | 0.90 | 0.60 |
| | Average | 0.96 | 0.89 | 0.63 |
| RMIA (Olmo 1B) | Prefix | 0.59 | 0.56 | 0.51 |
| | LoRA | 0.55 | 0.56 | 0.50 |
| | Full fine-tune | 0.87 | 0.56 | 0.51 |
| | Head fine-tune | 0.62 | 0.64 | 0.51 |
| | Average | 0.65 | 0.58 | 0.51 |
| Reference (Olmo 1B) | Prefix | 0.59 | 0.56 | 0.52 |
| | LoRA | 0.55 | 0.56 | 0.51 |
| | Full fine-tune | 0.86 | 0.56 | 0.51 |
| | Head fine-tune | 0.65 | 0.63 | 0.51 |
| | Average | 0.67 | 0.58 | 0.51 |
| Min-K% | Prefix | 0.51 | 0.50 | 0.50 |
| | LoRA | 0.50 | 0.50 | 0.50 |
| | Full fine-tune | 0.60 | 0.50 | 0.51 |
| | Head fine-tune | 0.54 | 0.51 | 0.50 |
| | Average | 0.53 | 0.50 | 0.50 |

Table 22: **Canary Exposure for OOD Datasets.** Prefix Tuning and Full Fine-Tuning adaptation methods have a higher exposure on OOD datasets than the other adaptation approaches like LoRA and Head Fine-Tuning. We audit only the adaptations and assume the same pretrained LLM is used for all adaptations. We present the exposure scores obtained using the model loss for the Pythia 1B model adapted to different OOD datasets with $\varepsilon \in \{0.1, 8, \infty\}$. The exposure differs between the adaptations only for $\varepsilon = \infty$ and approaches random guessing (values close to 1.44) for $\varepsilon \in \{0.1, 8\}$.

| Canary Prefix Type | Adaptation | SAMSum $\varepsilon = \infty$ | $\varepsilon = 8$ | $\varepsilon = 0.1$ | German Wiki $\varepsilon = \infty$ | $\varepsilon = 8$ | $\varepsilon = 0.1$ | Average $\varepsilon = \infty$ | $\varepsilon = 8$ | $\varepsilon = 0.1$ |
|---|---|---|---|---|---|---|---|---|---|---|
| Random | Prefix Tuning | 7.35 | 1.72 | 1.82 | 6.07 | 1.81 | 1.40 | 6.71 | 1.76 | 1.61 |
| | LoRA | 1.85 | 1.76 | 1.76 | 3.34 | 1.43 | 1.41 | 2.59 | 1.60 | 1.58 |
| | Full Fine-Tune | 6.91 | 1.77 | 1.75 | 5.76 | 1.43 | 1.43 | 6.33 | 1.60 | 1.59 |
| | Head Fine-Tune | 1.88 | 1.75 | 1.77 | 4.44 | 1.43 | 1.42 | 3.16 | 1.59 | 1.59 |
| | Average | 4.50 | 1.75 | 1.77 | 4.90 | 1.53 | 1.42 | 4.70 | 1.64 | 1.59 |
| Rare | Prefix Tuning | 6.44 | 1.41 | 1.55 | 5.22 | 1.82 | 2.11 | 5.83 | 1.61 | 1.83 |
| | LoRA | 1.54 | 1.49 | 1.52 | 2.47 | 1.81 | 1.79 | 2.01 | 1.65 | 1.66 |
| | Full Fine-Tune | 4.28 | 1.51 | 1.53 | 4.13 | 1.81 | 1.81 | 4.21 | 1.66 | 1.67 |
| | Head Fine-Tune | 1.54 | 1.56 | 1.52 | 3.65 | 1.81 | 1.80 | 2.60 | 1.69 | 1.66 |
| | Average | 3.45 | 1.49 | 1.53 | 3.87 | 1.81 | 1.88 | 3.66 | 1.65 | 1.70 |
| Common | Prefix Tuning | 7.54 | 1.97 | 1.81 | 5.02 | 2.17 | 2.54 | 6.28 | 2.07 | 2.17 |
| | LoRA | 1.90 | 1.92 | 2.00 | 2.84 | 1.75 | 1.82 | 2.37 | 1.83 | 1.91 |
| | Full Fine-Tune | 6.34 | 1.93 | 1.99 | 4.63 | 1.74 | 1.75 | 5.49 | 1.84 | 1.87 |
| | Head Fine-Tune | 3.05 | 1.93 | 1.98 | 3.30 | 1.74 | 1.76 | 3.18 | 1.83 | 1.87 |
| | Average | 4.71 | 1.94 | 1.94 | 3.95 | 1.85 | 1.97 | 4.33 | 1.89 | 1.96 |
| Invisible | Prefix Tuning | 5.16 | 2.14 | 2.19 | 7.17 | 1.96 | 1.25 | 6.16 | 2.05 | 1.72 |
| | LoRA | 3.82 | 1.74 | 1.61 | 2.54 | 1.44 | 1.40 | 3.18 | 1.59 | 1.50 |
| | Full Fine-Tune | 8.00 | 1.91 | 1.74 | 5.62 | 1.44 | 1.45 | 6.81 | 1.67 | 1.59 |
| | Head Fine-Tune | 5.91 | 1.67 | 1.59 | 3.66 | 1.44 | 1.45 | 4.78 | 1.55 | 1.52 |
| | Average | 5.72 | 1.87 | 1.78 | 4.75 | 1.57 | 1.39 | 5.23 | 1.72 | 1.58 |

Table 23: **Canary Exposure for IID Datasets.** We use the same setup as in Table 22 and observe the same trends, with higher privacy leakage for Prefix tuning and Full Fine-Tuning than for LoRA and Head Fine-Tuning.

| Canary Prefix Type | Adaptation | Bookcorpus2 Val | | | Bookcorpus2 Train | | | Github Val | | | Enron Val | | | Average | | |
|---|---|---|---|---|---|---|---|---|---|---|---|---|---|---|---|---|
| | | $\varepsilon=\infty$ | $\varepsilon=8$ | $\varepsilon=0.1$ | $\varepsilon=\infty$ | $\varepsilon=8$ | $\varepsilon=0.1$ | $\varepsilon=\infty$ | $\varepsilon=8$ | $\varepsilon=0.1$ | $\varepsilon=\infty$ | $\varepsilon=8$ | $\varepsilon=0.1$ | $\varepsilon=\infty$ | $\varepsilon=8$ | $\varepsilon=0.1$ |
| Random | Prefix Tuning | 8.00 | 2.02 | 1.24 | 8.00 | 1.69 | 1.59 | 7.86 | 1.88 | 1.22 | 5.80 | 0.91 | 1.58 | 7.41 | 1.63 | 1.41 |
| | LoRA | 3.65 | 2.06 | 2.05 | 3.19 | 1.55 | 1.55 | 3.22 | 1.89 | 1.88 | 2.04 | 0.67 | 0.67 | 3.03 | 1.54 | 1.54 |
| | Full Fine-Tune | 6.59 | 2.04 | 4.00 | 6.45 | 1.60 | 3.88 | 6.52 | 1.91 | 3.07 | 4.38 | 0.70 | 4.00 | 5.98 | 1.56 | 3.74 |
| | Head Fine-Tune | 2.81 | 2.03 | 1.84 | 2.34 | 1.58 | 1.59 | 2.70 | 1.89 | 1.85 | 1.20 | 0.69 | 0.75 | 2.26 | 1.55 | 1.51 |
| | Average | 5.26 | 2.04 | 2.28 | 5.00 | 1.61 | 2.15 | 5.08 | 1.89 | 2.01 | 3.35 | 0.74 | 1.75 | 4.67 | 1.57 | 2.05 |
| Rare | Prefix Tuning | 8.00 | 1.39 | 0.93 | 7.94 | 1.39 | 2.06 | 7.79 | 1.60 | 1.17 | 6.13 | 1.15 | 1.93 | 7.47 | 1.38 | 1.52 |
| | LoRA | 3.24 | 1.54 | 1.54 | 2.48 | 1.30 | 1.30 | 2.31 | 1.67 | 1.67 | 2.15 | 1.24 | 1.23 | 2.55 | 1.44 | 1.44 |
| | Full Fine-Tune | 5.40 | 1.54 | 3.23 | 4.87 | 1.31 | 2.82 | 4.73 | 1.68 | 4.52 | 4.05 | 1.27 | 1.79 | 4.76 | 1.45 | 3.09 |
| | Head Fine-Tune | 2.64 | 1.53 | 1.46 | 1.97 | 1.30 | 1.45 | 2.18 | 1.67 | 1.54 | 1.73 | 1.22 | 1.10 | 2.13 | 1.43 | 1.39 |
| | Average | 4.82 | 1.50 | 1.79 | 4.32 | 1.32 | 1.91 | 4.25 | 1.65 | 2.23 | 3.52 | 1.22 | 1.51 | 4.23 | 1.42 | 1.86 |
| Common | Prefix Tuning | 6.61 | 1.44 | 2.29 | 7.05 | 1.71 | 2.09 | 6.79 | 1.60 | 2.50 | 5.08 | 0.86 | 2.36 | 6.38 | 1.40 | 2.31 |
| | LoRA | 3.83 | 1.58 | 1.59 | 3.56 | 1.72 | 1.72 | 3.81 | 1.75 | 1.75 | 2.15 | 0.89 | 0.89 | 3.33 | 1.49 | 1.49 |
| | Full Fine-Tune | 5.27 | 1.60 | 2.91 | 4.66 | 1.75 | 2.80 | 6.24 | 1.74 | 3.08 | 3.60 | 0.90 | 1.98 | 4.94 | 1.50 | 2.69 |
| | Head Fine-Tune | 1.68 | 1.57 | 1.40 | 1.85 | 1.74 | 1.60 | 2.28 | 1.74 | 1.64 | 1.15 | 0.92 | 0.87 | 1.74 | 1.49 | 1.37 |
| | Average | 4.35 | 1.55 | 2.04 | 4.28 | 1.73 | 2.05 | 4.78 | 1.71 | 2.24 | 2.99 | 0.89 | 1.52 | 4.10 | 1.47 | 1.97 |
| Invisible | Prefix | 2.45 | 1.10 | 1.54 | 2.22 | 1.45 | 1.63 | 6.41 | 1.47 | 1.55 | 0.88 | 1.76 | 2.07 | 2.99 | 1.45 | 1.70 |
| | LoRA | 3.93 | 1.30 | 1.30 | 4.02 | 1.41 | 1.40 | 3.68 | 1.27 | 1.26 | 0.77 | 0.80 | 0.80 | 3.10 | 1.19 | 1.19 |
| | Full Fine-Tune | 8.00 | 1.34 | 1.32 | 8.00 | 1.45 | 1.52 | 6.30 | 1.30 | 1.33 | 5.21 | 0.78 | 0.82 | 6.88 | 1.22 | 1.25 |
| | Head Fine-Tune | 1.96 | 1.29 | 1.29 | 2.01 | 1.40 | 1.41 | 2.01 | 1.24 | 1.27 | 1.48 | 0.80 | 0.80 | 1.87 | 1.18 | 1.19 |
| | Average | 4.08 | 1.26 | 1.36 | 4.06 | 1.43 | 1.49 | 4.60 | 1.32 | 1.35 | 2.09 | 1.03 | 1.12 | 3.71 | 1.26 | 1.33 |

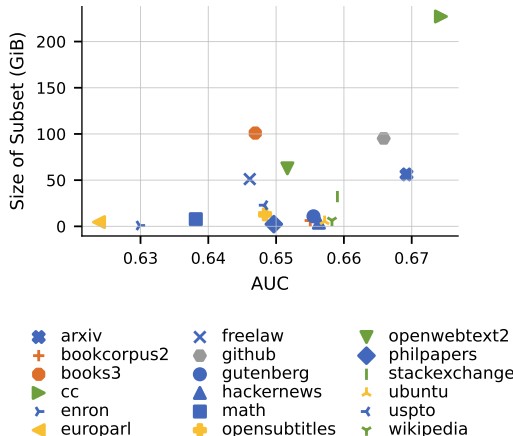

Figure 8: **Subset Size and Complexity.** The effect of the pretraining data subsets' size and complexity on the incurred privacy leakage from the corresponding LLM adaptations. We evaluate the leakage using AUC and the Pythia 1B adapted with $\varepsilon = 8$.

## C.6    COMPUTATIONAL COST ANALYSIS

To provide practical guidance on the computational efficiency of different adaptation methods, similarly to (Hanke et al., 2024), we measured the training time for each method on Pythia 1B under DP across all datasets. Since we use consistent data sizes and training epochs across all epsilon values, the reported training times are representative of the private training cost for each method. The results are listed in Table 24.

Table 24: **Training Time (minutes) for Different Adaptation Methods on Pythia 1B.** All methods are trained under differential privacy.

| Adaptation | SAMSum | German Wiki | Bookcorpus2 Val | Bookcorpus2 Train | Github Val | Enron Val |
|---|---|---|---|---|---|---|
| Prefix Tuning | 17.2 | 18 | 16 | 16 | 16 | 16 |
| LoRA | 15.5 | 6.5 | 5 | 5.5 | 5.2 | 5.2 |
| Full Fine-tune | 19.25 | 20 | 18.4 | 18.5 | 18.2 | 18.2 |
| Head Fine-tune | 3.5 | 4.7 | 3.2 | 3.25 | 3.5 | 3.5 |

The results show clear efficiency differences: Head Fine-tune is the fastest (3.2-4.7 minutes), followed by LoRA (5.0-15.5 minutes), Prefix Tuning (16-18 minutes), and Full Fine-tune (18.2-20 minutes). Combining these cost measurements with our privacy and utility analysis (RQ6), LoRA provides the best overall tradeoff: it achieves strong empirical privacy protection, maintains competitive utility, and requires moderate computational cost, significantly more efficient than Full Fine-tune while offering better privacy than Head Fine-tune.

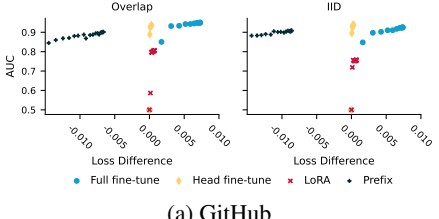 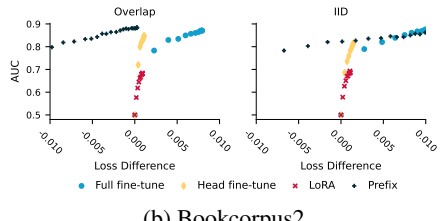

(a) GitHub                                      (b) Bookcorpus2

Figure 9: **Overlap and IID data show the same amount of privacy leakage across training.** The x-axis shows the difference between the initial pretrained loss and the evaluation loss. The y-axis represents the AUC score. We adapt Pythia 1B with $\varepsilon = 8$.

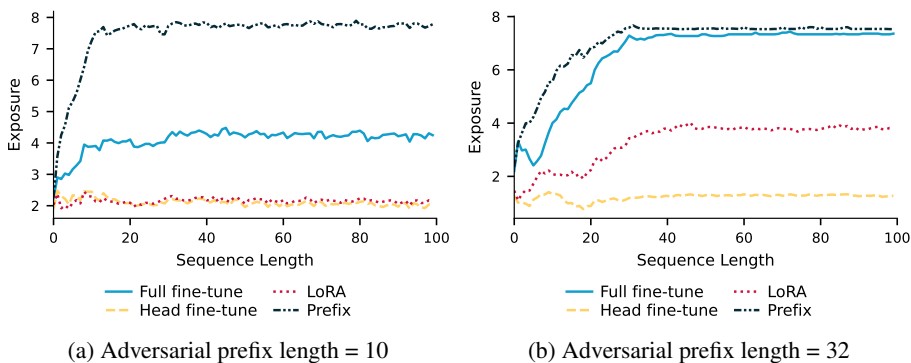

(a) Adversarial prefix length = 10           (b) Adversarial prefix length = 32

Figure 10: **The privacy leakage comes mostly from the adversarial prefix and much less from the interaction between the prefix and the sample.** We present the exposure when considering different lengths of canary prefixes after adapting Pythia 1B on Github Val. The evaluation was done for $\varepsilon = \infty$.

## D  INFLUENCE OF THE ATTACKER'S KNOWLEDGE

We can observe how impactful an attacker's knowledge about the target model and its pertaining data is. Specifically, under moderate privacy regimes (*i.e.*, $\varepsilon = 8$), *RMIA (shadow)* consistently achieves the best performance among models and datasets, as indicated in Table 6 - Table 17. However, the effectiveness of MIAs quickly drops off when we move to more realistic scenarios, such as using a pretrained model as a shadow model or having no shadow models available at all.

To model attackers with varying levels of background knowledge, we use a range of *shadow* models, including Pythia 14M, Pythia 160M, Pythia 1B, Pythia 2.8B (Biderman et al., 2023), GPT-neox (Black et al., 2021), OLMo 1B (Groeneveld et al., 2024a), and GPT-2 (Radford et al., 2019). Therefore, we can simulate various attacker capabilities and assess their impact on RMIA's effectiveness. As we can see in Figure 11, the choice of reference model has a small impact when attacking models fine-tuned on OOD data, even when architectural differences exist, such as between GPT-Neo 1.3B and OLMo 1B. On the other hand, the MIA achieves higher success rates on IID data when targeting the Pythia 1B model.

Additionally, Figure 12 illustrates the performance of various potential reference models over time. We consistently observe the significant impact of knowing the target model's architecture, especially when the target and *shadow* models share the same architecture. The only exception to this pattern appears in one of the OOD datasets, SAMSum.

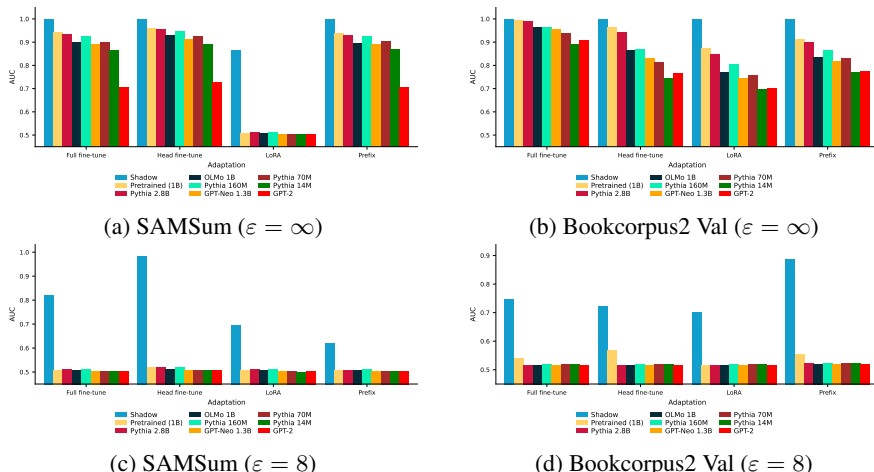

(a) SAMSum ($\varepsilon = \infty$)  (b) Bookcorpus2 Val ($\varepsilon = \infty$)

(c) SAMSum ($\varepsilon = 8$)  (d) Bookcorpus2 Val ($\varepsilon = 8$)

Figure 11: **Using at least one shadow model is crucial for RMIA, particularly for differentially private adaptations.** We present the AUC using RMIA with different types of shadow models after adapting Pythia 1B on Bookcorpus2 Val and SAMSum. The evaluation was done for $\varepsilon = \{8, \infty\}$.

# E  LOSS VALUES

## E.1  INITIAL LOSS OF THE LLM

Table 25 shows the loss at initialization for each dataset for the pretrained model and for a model adapted with an untrained Prefix Tuning.

Table 25: **Initial Losses for the Pythia 1B model on different datasets.** Standard refers to the model with default initialization, whereas Prefix refers to prepending an untrained Prefix Tuning to the hidden states.

| Adaptation \ Dataset | SAMSum | GermanWiki | Bookcorpus2 Val | Bookcorpus2 Train | GitHub Val | Enron Val |
|---|---|---|---|---|---|---|
| | $\varepsilon = 0$ | $\varepsilon = 0$ | $\varepsilon = 0$ | $\varepsilon = \infty$ | $\varepsilon = 0$ | $\varepsilon = 0$ |
| Standard | 2.747 | 2.732 | 3.011 | 2.997 | 1.539 | 2.388 |
| Prefix Tuning | 3.161 | 5.348 | 3.529 | 3.534 | 2.141 | 3.062 |

Table 26: **Validation loss values for the Pythia 1B model on different adaptation datasets.**

| Adaptation \ Dataset | SAMsum | | | German Wiki | | | Bookcorpus2 Val | | | Bookcorpus2 Train | | | Github Val | | | Enron Val | | |
|---|---|---|---|---|---|---|---|---|---|---|---|---|---|---|---|---|---|---|
| | $\varepsilon = \infty$ | $\varepsilon = 8$ | $\varepsilon = 0.1$ | $\varepsilon = \infty$ | $\varepsilon = 8$ | $\varepsilon = 0.1$ | $\varepsilon = \infty$ | $\varepsilon = 8$ | $\varepsilon = 0.1$ | $\varepsilon = \infty$ | $\varepsilon = 8$ | $\varepsilon = 0.1$ | $\varepsilon = \infty$ | $\varepsilon = 8$ | $\varepsilon = 0.1$ | $\varepsilon = \infty$ | $\varepsilon = 8$ | $\varepsilon = 0.1$ |
| Prefix Tuning | 2.311 | 2.451 | 2.778 | 2.573 | 2.738 | 2.838 | 2.968 | 2.993 | 3.387 | 2.997 | 2.994 | 3.390 | 1.599 | 1.557 | 2.054 | 2.412 | 2.426 | 3.002 |
| LoRA | 2.313 | 2.462 | 2.761 | 2.578 | 2.737 | 2.801 | 2.951 | 3.007 | 3.013 | 2.979 | 3.002 | 3.003 | 1.558 | 1.572 | 1.558 | 2.394 | 2.402 | 2.403 |
| Full Fine-Tune | 2.251 | 2.457 | 2.759 | 2.511 | 2.726 | 2.747 | 2.934 | 2.999 | 3.028 | 2.960 | 2.995 | 3.020 | 1.598 | 1.566 | 1.577 | 2.375 | 2.397 | 2.413 |
| Head Fine-Tune | 2.354 | 2.454 | 2.761 | 2.574 | 2.731 | 2.756 | 2.949 | 3.007 | 3.339 | 2.966 | 3.002 | 3.332 | 1.577 | 1.573 | 1.750 | 2.409 | 2.403 | 2.536 |
| Average | 2.307 | 2.456 | 2.764 | 2.559 | 2.733 | 2.785 | 2.950 | 3.002 | 3.192 | 2.976 | 2.998 | 3.186 | 1.583 | 1.567 | 1.734 | 2.397 | 2.407 | 2.589 |

## E.2  FINAL LOSS AND UTILITY OF THE LLM

Table 26 shows the final loss on the validation set. The hyperparameters are chosen to have similar loss between different adaptations using the same dataset and $\varepsilon$. We also present the loss for all other models in Tables 27 to 31. In Tables 33 to 38, we show that loss is an effective universal proxy for LLM utility. As evidence, we present Rouge-1 (R1) and perplexity for Pythia 1B adapted on each dataset. Since lower loss directly indicates higher utility, the reported loss reliably reflects model performance.

## E.3  OUT-OF-DOMAIN UTILITY

We also investigate the out-of-domain utility of adapted models. The evaluation should show how much of information not part of the adaptation set is influenced if a model is fine-tuned with DP. Our evaluations are done for $\varepsilon = 8$ and Pythia-1B on all datasets. The experiment was done by

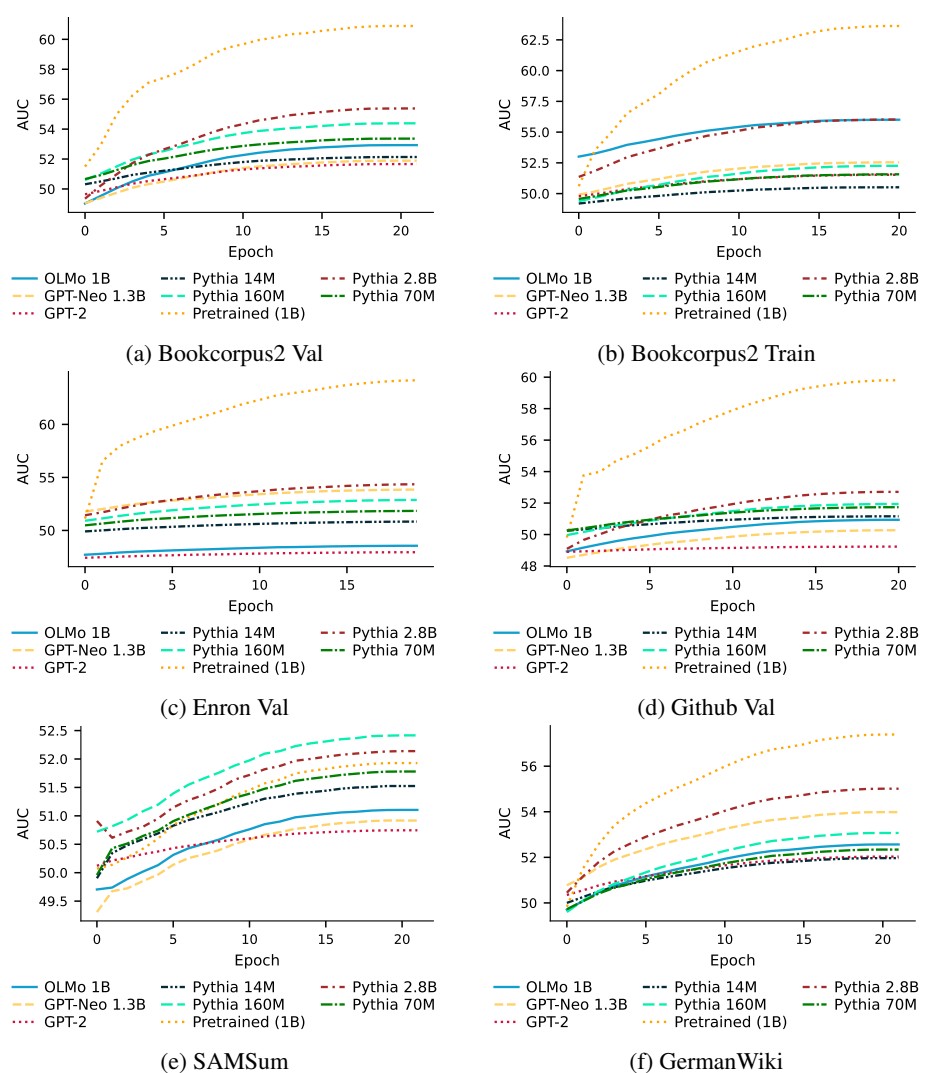

Figure 12: **Further analysis of the effectiveness of RMIA with pretrained models as a reference model.** As an extension of Figure 2, we fully fine-tuned Pythia 1B with $\varepsilon = 8$ using three additional IID datasets: Bookcorpus2 Train, GitHub Train, and GitHub Val.

Table 27: **Validation loss values for the Pythia 1.4B model on different adaptation datasets.**

| Adaptation \ Dataset | Samsum | | | German Wiki | | | Bookcorpus2 Val | | | Bookcorpus2 Train | | | Github Val | | | Enron Val | | |
|---|---|---|---|---|---|---|---|---|---|---|---|---|---|---|---|---|---|---|
| | $\varepsilon=\infty$ | $\varepsilon=8$ | $\varepsilon=0.1$ | $\varepsilon=\infty$ | $\varepsilon=8$ | $\varepsilon=0.1$ | $\varepsilon=\infty$ | $\varepsilon=8$ | $\varepsilon=0.1$ | $\varepsilon=\infty$ | $\varepsilon=8$ | $\varepsilon=0.1$ | $\varepsilon=\infty$ | $\varepsilon=8$ | $\varepsilon=0.1$ | $\varepsilon=\infty$ | $\varepsilon=8$ | $\varepsilon=0.1$ |
| Prefix | 2.712 | 2.456 | 3.451 | 2.465 | 2.655 | 5.246 | 2.901 | 3.538 | 3.857 | 2.929 | 3.657 | 3.918 | 1.542 | 2.564 | 2.909 | 2.411 | 2.973 | 3.791 |
| LoRA | 2.677 | 2.362 | 2.682 | 2.456 | 2.498 | 4.112 | 2.895 | 3.046 | 3.887 | 2.923 | 3.055 | 3.945 | 1.492 | 1.751 | 2.401 | 2.296 | 2.347 | 2.779 |
| Full fine-tune | 2.779 | 2.262 | 2.639 | 2.458 | 2.493 | 2.595 | 2.885 | 3.815 | 2.975 | 2.889 | 3.872 | 2.965 | 1.492 | 2.739 | 1.534 | 2.299 | 2.283 | 2.319 |
| Head fine-tune | 2.665 | 2.454 | 3.038 | 2.465 | 2.625 | 3.151 | 2.889 | 3.273 | 3.594 | 2.920 | 3.292 | 3.584 | 1.502 | 1.743 | 1.877 | 2.389 | 2.621 | 2.543 |
| Average | 2.708 | 2.384 | 2.952 | 2.461 | 2.568 | 3.776 | 2.892 | 3.418 | 3.578 | 2.915 | 3.469 | 3.603 | 1.507 | 2.199 | 2.180 | 2.349 | 2.556 | 2.858 |

Table 28: **Validation loss values for the Pythia 410M model on different adaptation datasets.**

| Adaptation \ Dataset | Samsum | | | German Wiki | | | Bookcorpus2 Val | | | Bookcorpus2 Train | | | Github Val | | | Enron Val | | |
|---|---|---|---|---|---|---|---|---|---|---|---|---|---|---|---|---|---|---|
| | $\varepsilon=\infty$ | $\varepsilon=8$ | $\varepsilon=0.1$ | $\varepsilon=\infty$ | $\varepsilon=8$ | $\varepsilon=0.1$ | $\varepsilon=\infty$ | $\varepsilon=8$ | $\varepsilon=0.1$ | $\varepsilon=\infty$ | $\varepsilon=8$ | $\varepsilon=0.1$ | $\varepsilon=\infty$ | $\varepsilon=8$ | $\varepsilon=0.1$ | $\varepsilon=\infty$ | $\varepsilon=8$ | $\varepsilon=0.1$ |
| Prefix | 2.486 | 2.966 | 7.227 | 2.957 | 3.345 | 9.669 | 3.249 | 3.583 | 4.702 | 3.284 | 3.665 | 4.792 | 2.139 | 2.760 | 8.701 | 2.990 | 3.835 | 4.869 |
| LoRA | 2.403 | 2.830 | 7.176 | 2.880 | 3.276 | 8.365 | 3.125 | 3.454 | 3.219 | 3.133 | 3.490 | 3.333 | 1.698 | 2.288 | 7.484 | 2.588 | 2.798 | 4.170 |
| Full fine-tune | 2.415 | 2.690 | 7.867 | 2.892 | 3.084 | 10.101 | 3.104 | 3.577 | 3.506 | 3.133 | 3.616 | 3.153 | 1.851 | 2.768 | 8.616 | 2.845 | 3.715 | 5.681 |
| Head fine-tune | 2.481 | 2.813 | 8.382 | 2.877 | 3.122 | 10.567 | 3.123 | 3.428 | 3.733 | 3.118 | 3.460 | 4.032 | 1.721 | 1.952 | 7.905 | 2.590 | 2.753 | 6.037 |
| Average | 2.446 | 2.825 | 7.663 | 2.901 | 3.207 | 9.676 | 3.150 | 3.511 | 3.790 | 3.163 | 3.558 | 3.827 | 1.852 | 2.442 | 8.176 | 2.753 | 3.275 | 5.189 |

Table 29: **Validation loss values for the Pythia 160M model on different adaptation datasets.**

| Adaptation \ Dataset | Samsum | | | German Wiki | | | Bookcorpus2 Val | | | Bookcorpus2 Train | | | Github Val | | | Enron Val | | |
|---|---|---|---|---|---|---|---|---|---|---|---|---|---|---|---|---|---|---|
| | $\varepsilon=\infty$ | $\varepsilon=8$ | $\varepsilon=0.1$ | $\varepsilon=\infty$ | $\varepsilon=8$ | $\varepsilon=0.1$ | $\varepsilon=\infty$ | $\varepsilon=8$ | $\varepsilon=0.1$ | $\varepsilon=\infty$ | $\varepsilon=8$ | $\varepsilon=0.1$ | $\varepsilon=\infty$ | $\varepsilon=8$ | $\varepsilon=0.1$ | $\varepsilon=\infty$ | $\varepsilon=8$ | $\varepsilon=0.1$ |
| Prefix | 3.011 | 3.475 | 3.436 | 3.715 | 3.742 | 4.448 | 3.608 | 3.598 | 3.808 | 3.641 | 3.641 | 3.865 | 2.571 | 2.488 | 3.138 | 3.407 | 3.389 | 3.735 |
| LoRA | 2.702 | 3.038 | 3.180 | 3.458 | 3.459 | 3.578 | 3.396 | 3.420 | 3.537 | 3.400 | 3.423 | 3.690 | 2.020 | 2.050 | 2.444 | 3.003 | 3.023 | 3.119 |
| Full fine-tune | 2.486 | 6.803 | 3.062 | 3.396 | 3.624 | 4.284 | 3.396 | 3.562 | 3.422 | 3.402 | 3.263 | 3.739 | 2.025 | 2.263 | 2.855 | 3.083 | 3.154 | 3.382 |
| Head fine-tune | 2.862 | 2.883 | 3.425 | 3.418 | 3.445 | 4.048 | 3.402 | 3.417 | 3.694 | 3.432 | 3.599 | 3.801 | 2.111 | 2.212 | 2.947 | 3.091 | 3.021 | 3.668 |
| Average | 2.765 | 4.050 | 3.276 | 3.497 | 3.567 | 4.089 | 3.450 | 3.499 | 3.615 | 3.469 | 3.563 | 3.774 | 2.182 | 2.253 | 2.846 | 3.146 | 3.147 | 3.476 |

Table 30: **Validation loss values for the Pythia 70M model on different adaptation datasets.**

| Adaptation \ Dataset | Samsum | | | German Wiki | | | Bookcorpus2 Val | | | Bookcorpus2 Train | | | Github Val | | | Enron Val | | |
|---|---|---|---|---|---|---|---|---|---|---|---|---|---|---|---|---|---|---|
| | $\varepsilon=\infty$ | $\varepsilon=8$ | $\varepsilon=0.1$ | $\varepsilon=\infty$ | $\varepsilon=8$ | $\varepsilon=0.1$ | $\varepsilon=\infty$ | $\varepsilon=8$ | $\varepsilon=0.1$ | $\varepsilon=\infty$ | $\varepsilon=8$ | $\varepsilon=0.1$ | $\varepsilon=\infty$ | $\varepsilon=8$ | $\varepsilon=0.1$ | $\varepsilon=\infty$ | $\varepsilon=8$ | $\varepsilon=0.1$ |
| Prefix | 3.451 | 3.348 | 3.956 | 4.243 | 4.167 | 4.761 | 3.970 | 3.954 | 4.144 | 4.017 | 3.986 | 4.191 | 2.902 | 2.757 | 3.064 | 3.845 | 3.787 | 4.121 |
| LoRA | 3.071 | 3.324 | 3.450 | 4.024 | 4.007 | 4.141 | 3.737 | 3.735 | 3.862 | 3.717 | 3.744 | 3.963 | 2.322 | 2.357 | 2.606 | 3.424 | 3.448 | 3.580 |
| Full fine-tune | 3.107 | 3.059 | 3.828 | 3.912 | 4.138 | 4.639 | 3.698 | 3.906 | 4.073 | 3.707 | 3.940 | 4.090 | 2.402 | 2.651 | 3.074 | 3.420 | 3.587 | 3.792 |
| Head fine-tune | 3.108 | 3.336 | 4.488 | 3.977 | 4.070 | 4.148 | 3.719 | 3.745 | 3.891 | 3.745 | 3.968 | 3.862 | 2.412 | 2.715 | 2.940 | 3.514 | 3.727 | 4.307 |
| Average | 3.184 | 3.267 | 3.930 | 4.039 | 4.095 | 4.422 | 3.781 | 3.835 | 3.993 | 3.797 | 3.909 | 4.027 | 2.509 | 2.620 | 2.921 | 3.551 | 3.637 | 3.950 |

Table 31: **Validation loss values for the GPT-Neo 1.3B model on different adaptation datasets.**

| Adaptation \ Dataset | Samsum | | | German Wiki | | | Bookcorpus2 Val | | | Bookcorpus2 Train | | | Github Val | | | Enron Val | | |
|---|---|---|---|---|---|---|---|---|---|---|---|---|---|---|---|---|---|---|
| | $\varepsilon=\infty$ | $\varepsilon=8$ | $\varepsilon=0.1$ | $\varepsilon=\infty$ | $\varepsilon=8$ | $\varepsilon=0.1$ | $\varepsilon=\infty$ | $\varepsilon=8$ | $\varepsilon=0.1$ | $\varepsilon=\infty$ | $\varepsilon=8$ | $\varepsilon=0.1$ | $\varepsilon=\infty$ | $\varepsilon=8$ | $\varepsilon=0.1$ | $\varepsilon=\infty$ | $\varepsilon=8$ | $\varepsilon=0.1$ |
| Prefix | 4.154 | 11.172 | 12.590 | 3.306 | 12.510 | 13.110 | 5.016 | 11.610 | 12.862 | 4.590 | 12.119 | 12.848 | 2.889 | 11.377 | 11.868 | 4.133 | 12.400 | 12.231 |
| LoRA | 2.723 | 2.407 | 2.724 | 2.450 | 2.409 | 2.505 | 3.062 | 3.042 | 3.062 | 3.050 | 3.033 | 3.050 | 1.247 | 2.913 | 11.451 | 2.156 | 2.153 | 2.156 |
| Full fine-tune | 2.494 | 2.630 | 3.578 | 2.568 | 3.101 | 4.375 | 3.302 | 3.509 | 4.281 | 3.311 | 3.560 | 4.324 | 2.146 | 8.471 | 2.471 | 2.344 | 2.475 | 2.760 |
| Head fine-tune | 2.713 | 2.558 | 2.999 | 2.447 | 2.617 | 2.877 | 3.060 | 6.326 | 3.568 | 3.052 | 3.312 | 3.569 | 1.325 | 1.427 | 1.546 | 2.240 | 2.292 | 2.367 |
| Average | 3.021 | 4.692 | 5.473 | 2.693 | 5.159 | 5.717 | 3.610 | 6.121 | 5.943 | 3.501 | 5.506 | 5.948 | 1.902 | 6.047 | 6.834 | 2.718 | 4.830 | 4.878 |

Table 32: **Validation loss values for the GPT-Neo 125M model on different adaptation datasets.**

| Adaptation \ Dataset | Samsum | | | German Wiki | | | Bookcorpus2 Val | | | Bookcorpus2 Train | | | Github Val | | | Enron Val | | |
|---|---|---|---|---|---|---|---|---|---|---|---|---|---|---|---|---|---|---|
| | $\varepsilon=\infty$ | $\varepsilon=8$ | $\varepsilon=0.1$ | $\varepsilon=\infty$ | $\varepsilon=8$ | $\varepsilon=0.1$ | $\varepsilon=\infty$ | $\varepsilon=8$ | $\varepsilon=0.1$ | $\varepsilon=\infty$ | $\varepsilon=8$ | $\varepsilon=0.1$ | $\varepsilon=\infty$ | $\varepsilon=8$ | $\varepsilon=0.1$ | $\varepsilon=\infty$ | $\varepsilon=8$ | $\varepsilon=0.1$ |
| Prefix | 4.891 | 14.114 | 14.174 | 5.640 | 20.577 | 20.623 | 6.251 | 14.268 | 14.337 | 7.370 | 14.299 | 14.401 | 5.117 | 13.307 | 13.368 | 6.308 | 14.242 | 14.242 |
| LoRA | 2.694 | 3.070 | 3.073 | 3.243 | 3.244 | 3.244 | 3.491 | 3.491 | 3.491 | 3.504 | 3.492 | 3.492 | 1.605 | 1.595 | 1.595 | 2.766 | 2.757 | 2.757 |
| Full fine-tune | 4.716 | 3.252 | 5.524 | 5.195 | 3.244 | 4.492 | 5.551 | 3.494 | 4.398 | 6.623 | 4.728 | 6.499 | 4.133 | 2.859 | 5.329 | 4.854 | 3.483 | 4.663 |
| Head fine-tune | 3.178 | 2.867 | 3.512 | 3.176 | 3.500 | 3.641 | 3.472 | 3.773 | 4.255 | 4.304 | 3.908 | 4.280 | 3.093 | 1.928 | 2.194 | 4.064 | 2.955 | 3.020 |
| Average | 3.870 | 5.826 | 6.571 | 4.314 | 7.641 | 8.000 | 4.691 | 6.256 | 6.620 | 5.450 | 6.607 | 7.168 | 3.487 | 4.922 | 5.622 | 4.498 | 5.859 | 6.170 |

Table 33: **Performance metrics comparison** for Pythia 1B model adapted to SAMSum with different adaptation methods.

| Adaptation \ Metric | Rouge-1 Score | | | Perplexity | | |
|---|---|---|---|---|---|---|
| | $\varepsilon=\infty$ | $\varepsilon=8$ | $\varepsilon=0.1$ | $\varepsilon=\infty$ | $\varepsilon=8$ | $\varepsilon=0.1$ |
| Prefix Tuning | 43.09 | 35.61 | 12.32 | 8.306 | 8.895 | 14.804 |
| LoRA | 44.00 | 38.44 | 17.88 | 8.279 | 8.806 | 10.890 |
| Full Fine-Tune | 44.33 | 38.81 | 21.47 | 8.070 | 8.794 | 13.882 |
| Head Fine-Tune | 31.25 | 26.51 | 15.14 | 9.584 | 10.098 | 14.708 |
| Average | 40.67 | 34.84 | 16.70 | 8.560 | 9.148 | 13.571 |

adapting the model with each dataset, and subsequently evaluating the loss of every other dataset, that is considered out-of-domain. The results can be seen in Tables 39a to 39d. The rows of the tables show the dataset used for adaptation, and the columns the dataset used for evaluation. We report the loss values here. Overall, the results demonstrate that the DP adaptations improve performance on the target (source) datasets, with only a minimal effect on utility for out-of-domain tasks. We see the highest variance in the results for Prefix-tuning. This is due to the higher perturbation introduced through the prefix in the hidden-space.

Table 34: **Performance metrics comparison** for Pythia 1B model adapted to GermanWiki with different adaptation methods.

| Metric / Adaptation | Rouge-1 Score | | | Perplexity | | |
|---|---|---|---|---|---|---|
| | $\varepsilon = \infty$ | $\varepsilon = 8$ | $\varepsilon = 0.1$ | $\varepsilon = \infty$ | $\varepsilon = 8$ | $\varepsilon = 0.1$ |
| Prefix Tuning | 14.13 | 10.95 | 9.32 | 13.105 | 15.456 | 17.082 |
| LoRA | 14.19 | 11.54 | 11.36 | 13.171 | 15.441 | 16.461 |
| Full Fine-Tune | 14.97 | 14.68 | 13.52 | 12.317 | 15.272 | 15.596 |
| Head Fine-Tune | 14.69 | 14.70 | 13.27 | 13.118 | 15.348 | 15.737 |
| Average | 14.50 | 12.97 | 11.87 | 12.928 | 15.379 | 16.219 |

Table 35: **Performance metrics comparison** for Pythia 1B model adapted to Bookcorpus2 Train with different adaptation methods.

| Metric / Adaptation | Rouge-1 Score | | | Perplexity | | |
|---|---|---|---|---|---|---|
| | $\varepsilon = \infty$ | $\varepsilon = 8$ | $\varepsilon = 0.1$ | $\varepsilon = \infty$ | $\varepsilon = 8$ | $\varepsilon = 0.1$ |
| Prefix Tuning | 22.50 | 17.59 | 15.99 | 19.453 | 19.945 | 29.577 |
| LoRA | 22.54 | 17.66 | 17.62 | 19.125 | 20.227 | 20.348 |
| Full Fine-Tune | 20.40 | 18.09 | 17.93 | 18.803 | 20.065 | 20.656 |
| Head Fine-Tune | 18.18 | 18.05 | 18.01 | 19.087 | 20.227 | 28.191 |
| Average | 20.91 | 17.85 | 17.39 | 19.117 | 20.116 | 24.693 |

Table 36: **Performance metrics comparison** for Pythia 1B model adapted to Bookcorpus2 Val with different adaptation methods.

| Metric / Adaptation | Rouge-1 Score | | | Perplexity | | |
|---|---|---|---|---|---|---|
| | $\varepsilon = \infty$ | $\varepsilon = 8$ | $\varepsilon = 0.1$ | $\varepsilon = \infty$ | $\varepsilon = 8$ | $\varepsilon = 0.1$ |
| Prefix Tuning | 21.45 | 17.43 | 14.76 | 20.025 | 19.965 | 29.666 |
| LoRA | 20.84 | 17.48 | 17.33 | 19.668 | 20.126 | 20.146 |
| Full Fine-Tune | 18.56 | 17.10 | 17.02 | 19.298 | 19.985 | 20.491 |
| Head Fine-Tune | 17.09 | 17.33 | 17.21 | 19.414 | 20.126 | 27.994 |
| Average | 19.99 | 17.34 | 16.58 | 19.601 | 20.051 | 24.074 |

Table 37: **Performance metrics comparison** for Pythia 1B model adapted to Github Val with different adaptation methods.

| Metric / Adaptation | Rouge-1 Score | | | Perplexity | | |
|---|---|---|---|---|---|---|
| | $\varepsilon = \infty$ | $\varepsilon = 8$ | $\varepsilon = 0.1$ | $\varepsilon = \infty$ | $\varepsilon = 8$ | $\varepsilon = 0.1$ |
| Prefix Tuning | 26.98 | 26.72 | 13.49 | 4.948 | 4.745 | 7.799 |
| LoRA | 24.79 | 29.93 | 29.82 | 4.749 | 4.816 | 4.749 |
| Full Fine-Tune | 28.72 | 29.53 | 28.72 | 4.943 | 4.787 | 4.840 |
| Head Fine-Tune | 30.03 | 29.07 | 28.68 | 4.840 | 4.821 | 5.755 |
| Average | 27.63 | 28.81 | 25.93 | 4.870 | 4.792 | 5.786 |

Table 38: **Performance metrics comparison** for Pythia 1B model adapted to Enron Val with different adaptation methods.

| Metric / Adaptation | Rouge-1 Score | | | Perplexity | | |
|---|---|---|---|---|---|---|
| | $\varepsilon = \infty$ | $\varepsilon = 8$ | $\varepsilon = 0.1$ | $\varepsilon = \infty$ | $\varepsilon = 8$ | $\varepsilon = 0.1$ |
| Prefix Tuning | 20.22 | 19.32 | 13.82 | 11.156 | 11.314 | 20.126 |
| LoRA | 20.07 | 20.52 | 20.17 | 10.957 | 11.045 | 11.056 |
| Full Fine-Tune | 21.24 | 20.80 | 19.84 | 10.751 | 10.990 | 11.167 |
| Head Fine-Tune | 21.05 | 19.75 | 19.22 | 11.123 | 11.056 | 12.629 |
| Average | 20.65 | 19.85 | 18.76 | 10.997 | 11.101 | 13.745 |

# F  EXPOSURE ESTIMATION

There are two common ways to estimate the exposure (Carlini et al., 2019): (1) by sampling and (2) by distribution modeling. Figure 13 shows that the two approximations are similar when using 256 non-member samples. To statistically show the correlation, we use the Pearson correlation test, where the null hypothesis is that the distributions underlying the samples are uncorrelated and normally distributed. The data yields an extremely small p-value, indicating a strong linear correlation between the two approximation methods.

Table 39: **Out-of-Domain Performance Pythia-1B adapted with $\varepsilon = 8$ and reported as the loss.** The rows of the tables show the dataset used for adaptation, and the columns the dataset used for evaluation.

(a) Prefix Tuning

| Train \ Eval | SAMSum (OOD) | GermanWiki (OOD) | Bookcorpus2 Val (IID) | Bookcorpus2 Train (Overlap) | Github Val (IID) | Enron Val (IID) |
|---|---|---|---|---|---|---|
| SAMSum (OOD) | **2.451** | 3.278 | 3.150 | 3.139 | 1.768 | 2.643 |
| GermanWiki (OOD) | 3.025 | **2.738** | 3.214 | 3.200 | 1.849 | 2.749 |
| Bookcorpus2 Val (IID) | 2.860 | 2.795 | **2.993** | 3.003 | 1.615 | 2.467 |
| Bookcorpus2 Train (Overlap) | 2.848 | 2.764 | 3.010 | **2.994** | 1.602 | 2.460 |
| Github Val (IID) | 2.862 | 2.742 | 3.041 | 3.026 | **1.557** | 2.422 |
| Enron Val (IID) | 2.864 | 2.807 | 3.050 | 3.038 | 1.598 | **2.426** |
| No Adaptation | 2.747 | 2.732 | 3.011 | 2.997 | 1.539 | 2.388 |

(b) LoRA

| Train \ Eval | SAMSum (OOD) | GermanWiki (OOD) | Bookcorpus2 Val (IID) | Bookcorpus2 Train (Overlap) | Github Val (IID) | Enron Val (IID) |
|---|---|---|---|---|---|---|
| SAMSum (OOD) | **2.462** | 2.730 | 3.019 | 3.004 | 1.544 | 2.396 |
| GermanWiki (OOD) | 2.748 | **2.729** | 3.011 | 2.997 | 1.538 | 2.388 |
| Bookcorpus2 Val (IID) | 2.747 | 2.732 | **3.007** | 2.996 | 1.538 | 2.388 |
| Bookcorpus2 Train (Overlap) | 2.748 | 2.732 | 3.010 | **3.002** | 1.538 | 2.388 |
| Github Val (IID) | 2.747 | 2.732 | 3.011 | 2.997 | **1.572** | 2.387 |
| Enron Val (IID) | 2.747 | 2.732 | 3.011 | 2.997 | 1.538 | **2.402** |
| No Adaptation | 2.747 | 2.732 | 3.011 | 2.997 | 1.539 | 2.388 |

(c) Full Fine-Tune

| Train \ Eval | SAMSum (OOD) | GermanWiki (OOD) | Bookcorpus2 Val (IID) | Bookcorpus2 Train (Overlap) | Github Val (IID) | Enron Val (IID) |
|---|---|---|---|---|---|---|
| SAMSum (OOD) | **2.457** | 2.730 | 3.015 | 3.001 | 1.541 | 2.392 |
| GermanWiki (OOD) | 2.747 | **2.726** | 3.011 | 2.997 | 1.538 | 2.388 |
| Bookcorpus2 Val (IID) | 2.743 | 2.734 | **2.999** | 2.991 | 1.537 | 2.386 |
| Bookcorpus2 Train (Overlap) | 2.743 | 2.734 | 3.005 | **2.995** | 1.537 | 2.387 |
| Github Val (IID) | 2.747 | 2.731 | 3.010 | 2.996 | **1.566** | 2.383 |
| Enron Val (IID) | 2.746 | 2.732 | 3.010 | 2.996 | 1.537 | **2.397** |
| No Adaptation | 2.747 | 2.732 | 3.011 | 2.997 | 1.539 | 2.388 |

(d) Head Fine-Tune

| Train \ Eval | SAMSum (OOD) | GermanWiki (OOD) | Bookcorpus2 Val (IID) | Bookcorpus2 Train (Overlap) | Github Val (IID) | Enron Val (IID) |
|---|---|---|---|---|---|---|
| SAMSum (OOD) | **2.454** | 2.724 | 3.068 | 3.050 | 1.573 | 2.442 |
| GermanWiki (OOD) | 2.743 | **2.712** | 3.011 | 2.997 | 1.539 | 2.388 |
| Bookcorpus2 Val (IID) | 2.747 | 2.729 | **3.007** | 2.997 | 1.538 | 2.388 |
| Bookcorpus2 Train (Overlap) | 2.747 | 2.733 | 3.011 | **3.002** | 1.538 | 2.388 |
| Github Val (IID) | 2.747 | 2.732 | 3.011 | 2.997 | **1.536** | 2.388 |
| Enron Val (IID) | 2.747 | 2.732 | 3.011 | 2.997 | 1.538 | **2.387** |
| No Adaptation | 2.747 | 2.732 | 3.011 | 2.997 | 1.539 | 2.388 |

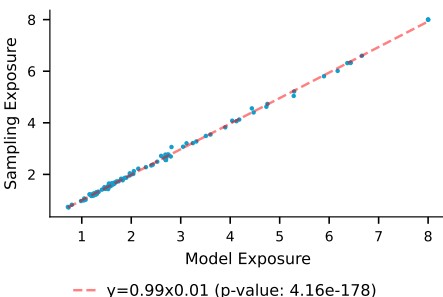

-- - y=0.99x0.01 (p-value: 4.16e-178)

Figure 13: **The two ways to approximate the exposure are similar.** The relation between the model exposure and sampling exposure. The p-value is related to the Pearson correlation test.

# G MEMORIZATION OF THE PRETRAINED MODEL

Table 40 shows the number of memorized samples in the pretrained model.

Table 40: **Set of memorized samples identified from the subsets of the Pile dataset.**

| Subset | GitHub | Bookcorpus2 | Enron | ArXiv | CC | EuroParl | FreeLaw | USPTO | Wikipedia |
|---|---|---|---|---|---|---|---|---|---|
| Memorized Samples | 192 | 3 | 18 | 2 | 8 | 0 | 7 | 4 | 2 |

## H RMIA HYPERPARAMETERS

We focus on the importance of $\gamma$, as $\alpha$ has a much more limited effect, and we set it to 0. Figure 14 shows the importance and $\gamma$ and suggests that $\gamma = 1$ is often the best choice. We omit it for simplicity, but a similar trend can be observed for the other settings.

## I BROADER IMPACT

Recognizing a potential underestimation of privacy risks in adapted LLMs due to insufficient empirical analysis of the combined effects of pretraining and adaptation, we conduct a rigorous benchmark. Our work offers impact by providing the community with clear guidance on privacy-preserving strategies, suitable adaptation techniques, thus contributing to more privacy-aware adapting LLMs.

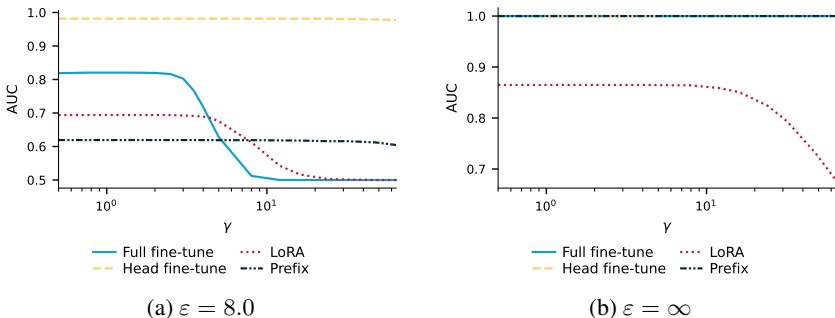

(a) $\varepsilon = 8.0$                            (b) $\varepsilon = \infty$

Figure 14: $\gamma = 1$ **is a strong baseline.** We present the AUC using RMIA with different types of values of $\gamma$ after adapting Pythia 1B on SAMSum. The evaluation was done for $\varepsilon = \{8, \infty\}$.

Furthermore, our holistic privacy auditing framework in the pretrain-adapt paradigm stands out by providing a comprehensive privacy assessment across the entire pipeline rather than isolated stages. Previous methods focus on the separated stages, thus overlooking interactions that can influence the data leakage. Our approach equips practitioners with the tools to trace privacy risks across a model's lifecycle. Let's consider a case with the LLM adapted to the medical domain with private data from one hospital. An individual finds their data can be extracted, raising the question: "Does the leakage come from the pretraining data, the hospital's fine-tuning data, or both?" Traditional auditing frameworks fail to cover cases where data appears in both stages or becomes extractable only after adaptation. However, by framing each audit as an adversarial game, our framework quantifies and localizes privacy risks, thus offering reproducible evaluations across models and datasets.

## J LIMITATIONS

This work focuses solely on auditing the private adaptations and leakage from pretraining data after adaptations. However, as we show, for holistic privacy auditing under the pretrain-adapt paradigm, we need ways to audit all process stages (jointly). We also focus only on a subset of models, particularly leaving out state-of-the-art closed models, such as GPT-4, given that they cannot easily be adapted with DP as of the current API specification.

