# OpenReview forum: "Benchmarking Empirical Privacy Protection for Adaptations of Large Language Models"
_ICLR.cc/2026/Conference — ICLR 2026 Oral_

### Official Review · Reviewer_hzjs · 2025-10-15

**Soundness:** 3
**Presentation:** 2
**Contribution:** 2
**Rating:** 4
**Confidence:** 4

**Summary:**

This paper presents a comprehensive benchmark to quantify and compare the empirical privacy protection abilities of large language models with differential privacy guarantees. The proposed benchmark leverages robust membership inference and canary data extraction to investigate the potential privacy risks. Unlike prior works, the benchmark considers data distributions of adaptation data and pretraining data. Both in-distribution cases and out-of-distribution examples are carefully studied. Moreover, the benchmark covers various LLMs, adaptation methods, privacy budgets and datasets to perform extensive evaluations. Experimental results show insights on several research questions.

**Strengths:**

1. The paper is well written with clear motivations. The empirical privacy risk assessment is an important task for DP-protected LLMs.

2. The paper systematically studies the relationship between the pretraining data and the fine-tuning data (adaptation data). Different datasets are considered for both in-domain and out-of-domain setups.

3. Comprehensive experiments are conducted with new robust MIA attacks and exposures.

**Weaknesses:**

1. My major concern is the evaluation of the utility part. According to the paper, only C.6 and E.2 discuss the privacy-utility tradeoff. Yet, utility is crucial when using various $\epsilon$, under a strict budget $\epsilon = 0.1$, if the model cannot handle the downstream tasks at all, DP tuning will be meaningless. However, the paper only evaluates perplexity and validation loss as the utility indicator. Such utility metrics are not feasible or representative of the utility performance. A more comprehensive utility evaluation should be properly studied.

2. The evaluated LLMs are rather small-scale and outdated. I would encourage OLMo 2 7B to study how the size of LLMs affects privacy performance.

3. Clarity regarding section 6. I am quite confused about the privacy auditing mentioned here. I think more background and explanations on holistic audits would be helpful to understand this section.

**Questions:**

1. How do the authors justify using perplexity and validation loss as sufficient indicators of model utility under differential privacy? Have the authors explored fine-tuning under DP constraints to assess out-of-domain task utility?

2. How do the authors expect their findings to scale with larger models such as OLMo 2 7B or Qwen2.5/3 architectures?

3. Could the authors define “holistic privacy audits” more precisely? How does it differ from existing auditing frameworks?

---

> ### Author Response · Authors · 2025-11-20
>
> We thank the Reviewer for the positive and detailed feedback. We appreciate that the Reviewer highlights the clarity of our motivation, the importance of our work and the extensive evaluations using RMIA.
>
> In summary, we present new results on the utility of adapting the model with different values of $\varepsilon$, out-of-domain task utility, the different sizes of models that we evaluate on, and provide more details in Section 6 on our holistic auditing framework.
>
> We provide detailed answers to all weaknesses and questions in-line below:
>
> >**The paper only evaluates perplexity and validation loss as the utility indicator. Such utility metrics are not feasible or representative of the utility performance. A more comprehensive utility evaluation should be properly studied.** & **How do the authors justify using perplexity and validation loss as sufficient indicators of model utility under differential privacy?**
>
> To assess the utility of our adapted models, we used perplexity and validation loss as proxies to ensure comparability across the wide range of datasets and adaptation settings studied. For datasets supporting downstream task evaluation, such as SAMSum, we additionally report Rouge-1 scores alongside perplexity in Table 31 in the initial submission (and now Table 33 in the updated version), which confirms the same privacy-utility trend across the different $\varepsilon$ values. Lower loss consistently corresponds to higher task performance, validating its use as a reliable utility indicator across settings.
>
> We also ran the same evaluation on all other datasets we considered for $\varepsilon = \\{0.1, 8.0, \infty\\}$. Using each dataset’s held out set, we provided the model with a 50 token prefix from every evaluation sequence and then measured how closely its next 50 generated tokens matched the true continuation using the Rouge-1 score. The results added in Tables 34 to 38 in Appendix E.2 of the updated paper, and also shown below, alongside the SAMSum results already reported.
>
> SAMSum
> |Adaptation|$\varepsilon=\infty$||$\varepsilon=8$||$\varepsilon=0.1$||
> |-|-|-|-|-|-|-|
> |Method|PPL|R1|PPL|R1|PPL|R1|
> |Prefix|8.306|43.09|8.895|35.61|14.804|12.32|
> |LoRA|8.279|44.00|8.806|38.44|10.890|17.88|
> |FT|8.070|44.33|8.794|38.81|13.882|21.47|
> |HT|9.584|31.25|10.098|26.51|14.708|15.14|
> GermanWiki
>
> |Adaptation|$\varepsilon=\infty$||$\varepsilon=8$||$\varepsilon=0.1$||
> |-|-|-|-|-|-|-|
> |Method|PPL|R1|PPL|R1|PPL|R1|
> |Prefix|13.105|14.13|15.456|10.95|17.082|9.32|
> |LoRA|13.171|14.19|15.441|11.54|16.461|11.36|
> |FT|12.317|14.97|15.272|14.68|15.596|13.52|
> |HT|13.118|14.69|15.348|14.702|15.737|13.27|
>
> BookCorpus2 Val
>
> |Adaptation|$\varepsilon=\infty$||$\varepsilon=8$||$\varepsilon=0.1$||
> |-|-|-|-|-|-|-|
> |Method|PPL|R1|PPL|R1|PPL|R1|
> |Prefix|19.453|22.50|19.945|17.59|29.577|15.99|
> |LoRA|19.125|22.54|20.227|17.66|20.348|17.62|
> |FT|18.803|20.40|20.065|18.09|20.656|17.93|
> |HT|19.087|18.18|20.227|18.05|28.191|18.01|
>
> BookCorpus2 Train
>
> |Adaptation|$\varepsilon=\infty$||$\varepsilon=8$||$\varepsilon=0.1$||
> |-|-|-|-|-|-|-|
> |Method|PPL|R1|PPL|R1|PPL|R1|
> |Prefix|20.025|21.45|19.965|17.43|29.666|14.76|
> |LoRA|19.668|20.84|20.126|17.48|20.146|17.33|
> |FT|19.298|18.56|19.985|17.10|20.491|17.02|
> |HT|19.414|17.09|20.126|17.33|27.994|17.21|
>
> Github Val
>
> |Adaptation|$\varepsilon=\infty$||$\varepsilon=8$||$\varepsilon=0.1$||
> |-|-|-|-|-|-|-|
> |Method|PPL|R1|PPL|R1|PPL|R1|
> |Prefix|4.948|26.98|4.745|26.72|7.799|13.49|
> |LoRA|4.749|24.79|4.816|29.93|4.749|29.82|
> |FT|4.943|28.72|4.787|29.53|4.840|28.72|
> |HT|4.840|30.03|4.821|29.07|5.755|28.68|
>
> Enron Val
>
> |Adaptation|$\varepsilon=\infty$||$\varepsilon=8$||$\varepsilon=0.1$||
> |-|-|-|-|-|-|-|
> |Method|PPL|R1|PPL|R1|PPL|R1|
> |Prefix|11.156|20.22|11.314|19.32|20.126|13.82|
> |LoRA|10.957|20.07|11.045|20.52|11.056|20.17|
> |FT|10.751|21.24|10.990|20.80|11.167|19.84|
> |HT|11.123|21.05|11.056|19.75|12.629|19.22|
>
> Across all datasets, the results show that perplexity can be reliably used as a proxy for task-level utility, as shown with Rouge-1. For every adaptation method and dataset, higher perplexity or loss is paired with a lower Rouge-1 score, indicating that likelihood-based metrics reliably reflect downstream utility even in settings where a full task benchmark is not available. The drop in utility is most pronounced at $\varepsilon=0.1$, while $\varepsilon=8$ remains close to the non-private baseline.
>
> For most of the datasets we evaluate, no established downstream metric exists, and the data are not structured as tasks such as SAMSum. In such cases, perplexity provides the only consistent and model-agnostic metric that can be applied across all domains. The additional evaluations included here confirm that, whenever a task-level metric is available, it aligns well with the perplexity-based assessment we used in our paper. Together, these results support the use of perplexity and loss as practical and informative indicators of utility under differential privacy across a set of different datasets.

---

> > ### Author Response · Authors · 2025-11-20
> >
> > >**Have the authors explored fine-tuning under DP constraints to assess out-of-domain task utility?**
> >
> > Out-of-domain performance is not such a relevant objective for our current privacy assessment, as DP adaptation is typically used to fine-tune a model on sensitive, domain-specific data rather than to improve general utility across domains. However, to address the reviewer’s comment, we do provide a comprehensive assessment and run experiments showing the out-of-domain performance on models adapted with DP on (1) all of the four adaptations (Prefix Tuning, LoRA, Full Fine-Tuning, and Head Fine-Tuning), (2) six datasets (OOD: SAMSum, GermanWiki, and IID: BookCorpus2 Val, BookCorpus2 Train, GitHub Val, and Enron Val) and (3) $\varepsilon = 8$ to create a full matrix of results.
> >
> > We have separated the results into tables for each different adaptation type. The rows indicate the source datasets used for adaptations and the columns are for the datasets on which we assess the utility with the loss. The last row within each table shows the loss on the model without any adaptation, as a baseline. We also include the results in Tables 39a-d, in Appendix E.3 in the updated version of our submission.
> >
> > |Prefix Tuning|SAMSum (OOD)|GermanWiki (OOD)|BookCorpus2 Val (IID)|BookCorpus2 Train (Overlap)|Github Val (IID)|Enron Val (IID)|
> > |-|-|-|-|-|-|-|
> > |**SAMSum (OOD)**|**2.451**|3.278|3.150|3.139|1.768|2.643|
> > |**GermanWiki (OOD)**|3.025|**2.738**|3.214|3.200|1.849|2.749|
> > |**BookCorpus2 Val (IID)**|2.860|2.795|**2.993**|3.003|1.615|2.467|
> > |**BookCorpus2 Train (Overlap)**|2.848|2.764|3.010|**2.994**|1.602|2.460|
> > |**Github Val (IID)**|2.862|2.742|3.041|3.026|**1.557**|2.422|
> > |**Enron Val (IID)**|2.864|2.807|3.050|3.038|1.598|**2.426**|
> > |**NoAdaptation**|2.747|2.732|3.011|2.997|1.539|2.388|
> >
> >
> > |LoRA|SAMSum (OOD)|GermanWiki (OOD)|BookCorpus2 Val (IID)|BookCorpus2 Train (Overlap)|Github Val (IID)|Enron Val (IID)|
> > |-|-|-|-|-|-|-|
> > |**SAMSum (OOD)**|**2.462**|2.730|3.019|3.004|1.544|2.396|
> > |**GermanWiki (OOD)**|2.748|**2.729**|3.011|2.997|1.538|2.388|
> > |**BookCorpus2 Val (IID)**|2.747|2.732|**3.007**|2.996|1.538|2.388|
> > |**BookCorpus2Train(Overlap)**|2.748|2.732|3.010|**3.002**|1.538|2.388|
> > |**GithubVal(IID)**|2.747|2.732|3.011|2.997|**1.572**|2.387|
> > |**EnronVal(IID)**|2.747|2.732|3.011|2.997|1.538|**2.402**|
> > |**NoAdaptation**|2.747|2.732|3.011|2.997|1.539|2.388|
> >
> >
> > |Full Fine-tune|SAMSum (OOD)|GermanWiki (OOD)|BookCorpus2 Val (IID)|BookCorpus2 Train (Overlap)|Github Val (IID)|Enron Val (IID)|
> > |-|-|-|-|-|-|-|
> > |**SAMSum (OOD)**|**2.457**|2.730|3.015|3.001|1.541|2.392|
> > |**GermanWiki (OOD)**|2.747|**2.726**|3.011|2.997|1.538|2.388|
> > |**BookCorpus2 Val (IID)**|2.743|2.734|**2.999**|2.991|1.537|2.386|
> > |**BookCorpus2 Train (Overlap)**|2.743|2.734|3.005|**2.995**|1.537|2.387|
> > |**Github Val (IID)**|2.747|2.731|3.010|2.996|**1.566**|2.383|
> > |**Enron Val (IID)**|2.746|2.732|3.010|2.996|1.537|**2.397**|
> > |**No Adaptation**|2.747|2.732|3.011|2.997|1.539|2.388|
> >
> >
> > |Head Fine-tune|SAMSum (OOD)|GermanWiki (OOD)|BookCorpus2 Val (IID)|BookCorpus2 Train (Overlap)|Github Val (IID)|Enron Val (IID)|
> > |-|-|-|-|-|-|-|
> > |**SAMSum (OOD)**|**2.454**|2.724|3.068|3.050|1.573|2.442|
> > |**GermanWiki (OOD)**|2.743|**2.712**|3.011|2.997|1.539|2.388|
> > |**BookCorpus2 Val (IID)**|2.747|2.729|**3.007**|2.997|1.538|2.388|
> > |**BookCorpus2 Train (Overlap)**|2.747|2.733|3.011|**3.002**|1.538|2.388|
> > |**Github Val (IID)**|2.747|2.732|3.011|2.997|**1.536**|2.388|
> > |**Enron Val (IID)**|2.747|2.732|3.011|2.997|1.538|**2.387**|
> > |**No Adaptation**|2.747|2.732|3.011|2.997|1.539|2.388|
> >
> >
> > Overall, the results demonstrate that, as expected, the DP adaptations improve performance on the target (source) datasets, with only a minimal effect on utility for out-of-domain tasks. We see the highest variance in the results for Prefix-tuning. This is due to the higher perturbation introduced through the prefix in the hidden-space.
> > Finally, note that the prefix tuning can be attached separately to each input query depending on its task. In other words, the prefix-based adaptation enables us to even process examples from multiple users/tasks in a single batch [1].
> >
> > **References:**
> >
> > [1] “Prefix-Tuning: Optimizing Continuous Prompts for Generation“ Xiang Lisa Li and Percy Liang. ACL 2021.

---

> > > ### Author Response · Authors · 2025-11-20
> > >
> > > >**The evaluated LLMs are rather small-scale and outdated. I would encourage OLMo 2 7B to study how the size of LLMs affects privacy performance.** & **How do the authors expect their findings to scale with larger models such as OLMo 2 7B or Qwen2.5/3 architectures?**
> > >
> > > We included results on OLMo 2 1B as a newer open-source baseline to the extent possible without access to IID data. We did report results for experiments on OLMo 2 1B in the initial submission, please see Tables 18 and 19. The main point is that the privacy trends observed across model sizes (160 M, 410 M, and 1 B) as well as across model versions (OLMo 1B and OLMo 2 1B) are consistent, suggesting that the main findings are expected to generalize to bigger and newer models. Our benchmark focuses on models up to 1B parameters to enable controlled, reproducible, and large-scale evaluations across adaptation methods. For other models, with undisclosed training data, we cannot identify training, IID, and OOD data, making such analysis unreliable.
> > >
> > > Following the reviewer's suggestion, we also conducted additional experiments with OLMo 2 7B using LoRA:
> > >
> > > |$\varepsilon$ | SAMSum (OOD) | GermanWiki (OOD) | DolminoWiki (IID) |
> > > |-|-|-|-|
> > > |$\varepsilon=\infty$| 0.90 | 0.98 | 1.00 |
> > > |$\varepsilon=8$| 0.52 | 0.51 | 0.64 |
> > > |$\varepsilon=0.1$| 0.51 | 0.50 | 0.51 |
> > >
> > > For non-DP settings, the 7B model shows comparable vulnerability to the 1B model, confirming that non-private adaptations remain highly vulnerable regardless of scale. We can also observe that the IID/OOD vulnerability gap persists at the 7B scale, reinforcing our main finding that data distribution significantly impacts privacy leakage. The result also demonstrates that strong DP guarantees remain effective at scale and provides evidence that our core findings generalize to larger architectures.
> > >
> > >
> > > >**Clarity regarding Section 6. I am quite confused about the privacy auditing mentioned here. I think more background and explanations on holistic audits would be helpful to understand this section.** & **Could the authors define “holistic privacy audits” more precisely? How does it differ from existing auditing frameworks?**
> > >
> > >
> > > Holistic privacy auditing extends prior work by defining privacy assessment for the entire pretrain-adapt pipeline rather than treating pretraining and fine-tuning as separate, independent stages. Existing approaches typically examine leakage from only one source at a time, assuming that risks can be decomposed cleanly by stage. Our results show that this assumption does not hold: the pretraining data and the adaptation data may interact in ways that change the model’s privacy. For example, if a medical LLM is fine-tuned on data from a single hospital and a patient discovers that their information is extractable, identifying the origin of that leakage requires determining whether it stems from the pretraining corpus S, the hospital’s dataset D, or the interaction between the two. Most existing auditing strategies cannot handle cases where the same data appears in both S and D, or where information present only in D becomes extractable only after adaptation.
> > >
> > > Our holistic framework addresses these gaps by defining four complementary audits that quantify and localize leakage across all stages of the model lifecycle. Each audit is framed as a precise adversarial game, enabling reproducible evaluation across datasets and models and providing a unified way to assess privacy risks from pretraining, adaptation, or their interaction. This formulation of the framework can serve as future guidance for the community on properly evaluating privacy risks in LLMs. Finally, to further clarify the practical use of the framework, we added a paragraph in Appendix I.
> > >
> > >
> > > We would greatly appreciate updating the rating if the above responses address the Reviewer's concerns.

---

> > > > ### Comment · Reviewer_hzjs · 2025-11-22
> > > > **Rebuttal Acknowledgement**
> > > >
> > > > Dear Authors,
> > > >
> > > > I have read the rebuttal and I would like to argue that PPL still cannot fully indicate the overall utility, as it cannot exactly reflect the downstream performance. Even if PPL only increases by no more than 1, R1 could reduce more than 6\%. When it comes to classification tasks, such as multiple-choice questions, the accuracy can be expected to be worse (even it is not covered by your experiments).
> > > >
> > > > I think I would prefer a review score of 5. However, given that there is no 5, I will raise my score to 6 and hope this paper may be accepted smoothly.
> > > >
> > > > Good luck!

---

> > > > > ### Author Response · Authors · 2025-11-23
> > > > > **Perplexity and test loss are standard metrics for evaluating language model performance**
> > > > >
> > > > > Dear Reviewer,
> > > > >
> > > > > We really appreciate your engagement in the discussion, valuable input, and the decision to raise the rating.
> > > > >
> > > > > We would like to emphasize that perplexity (PPL) and test loss are standard metrics for evaluating language model performance. This is grounded in the practices of recent and foundational works [1, 2, 3, 4], which consistently use PPL and test loss to evaluate model performance. Therefore, we also used PPL and test loss as our primary metrics. We believe this remains the best unified proxy for capturing the behavior of the wide range of models and datasets analyzed in our work.
> > > > >
> > > > > Once again, thank you for the discussion. Please, let us know if the above answer addresses the concern.
> > > > >
> > > > > --------
> > > > >
> > > > > **References:**
> > > > >
> > > > > [1] *"Privacy Auditing of Large Language Models."* Panda, Ashwinee, et al., ICLR 2025.
> > > > >
> > > > > [2] *"Scaling Laws for Differentially Private Language Models."*  McKenna, Ryan, et al. ICML 2025.
> > > > >
> > > > > [3] *"Scaling Laws for Neural Language Models"* Kaplan, Jared, et al. 2020.
> > > > >
> > > > > [4] *"An Empirical Analysis of Memorization in Fine-tuned Autoregressive Language Models"* Mireshghallah, Fatemehsadat, et al., EMNLP 2022.

---

### Official Review · Reviewer_ZdyA · 2025-10-25

**Soundness:** 3
**Presentation:** 3
**Contribution:** 2
**Rating:** 4
**Confidence:** 4

**Summary:**

This work studies privacy risks under DP adaptations in LLMs, using robust membership interference and canary data extraction. They study the effects of overlaps and interdependencies of the pretraining data with the adaptation data, using exact overlaps, in-distribution data and out-of-distribution data. They also study the vulnerability for different adaptation methods and different privacy regimes. The authors find that the data distribution strongly affects the privacy vulnerability, and that parameter-efficient fine-tuning methods, such as LoRA, achieve the highest empirical privacy protection for OOD data.

**Strengths:**

- This work studies a comprehensive range of datasets, adaptation methods, and different LLMs with a range of sizes.
- The authors run experiments on a range of IID and OOD datasets, and are able to show empirical trends on how the IID settings have higher privacy vulnerabilities than OOD settings.
- This work is highly relevant to practical applications using DP with LLMs.

**Weaknesses:**

- The experiments could be run on a wider range of privacy parameters (the authors only use $\epsilon=0.1, 8$ and the nonprivate setting).
- The authors discuss the privacy-utility tradeoffs for the different methods in the appendix. However, this is quite an important topic and I think it is worth moving this to the main paper. Similarly, it would also be useful to include some discussion on the costs of the different methods and the tradeoffs with privacy and utility.
- The authors propose a new framework for holistic privacy assessment using adversarial games, but it would help to give examples to demonstrate how this framework would help in practice.
- This work only focuses on open-source models, and it is unclear how to generalize the results to close-source models where the pretraining data distribution is not known.
- The work is empirical and does not give any theoretical explanations, for instance on why some adaptations have lower vulnerability than others.

**Questions:**

- How could this work apply to closed-source models? Are there any tools or insights from this work that could apply there?
- Are there explanations on why different adaptation methods perform better than others?

---

> ### Author Response · Authors · 2025-11-20
>
> We thank the Reviewer for the positive and detailed feedback. We appreciate that the Reviewer finds our work comprehensive, evaluating on a large range of datasets, adaptation methods, and LLMs, and highly relevant for practical applications of differential privacy with LLMs.
>
> In summary, we have added evaluations on more $\varepsilon$ values $\\{1, 3, 5\\}$, moved the privacy-utility tradeoff evaluation as RQ6 into the main paper, and showed the cost through training time of different datasets and adaptations under DP. We further explain our holistic auditing in greater detail and discuss using our framework on closed-source models.
>
> We provide detailed answers in-line below:
>
> > **The experiments could be run on a wider range of privacy parameters (the authors only use $\varepsilon = \\{0.1, 8\\}$ and the nonprivate setting).**
>
> We thank the reviewer for their suggestion. In the previous version of our paper in Appendix C1, Figure 6, we showed results for $\varepsilon \in \\{0.1, 0.5, 1, 3, 8, 50, \infty\\}$ for BookCorpus2 Val and SAMSum. The trends show that, as expected, higher $\varepsilon$ leads to higher privacy leakage.
>
> To address the reviewer’s comment further, we added more experiments and ran MIAs for *all* datasets and *all* adaptation methods under  $\varepsilon \in  \\{1, 3, 5\\}$ on Pythia 1B and added the full results into the paper as a new Figure 7.
>
>
> **SAMSum**
>
> |Adaptation|$\varepsilon = 5$|$\varepsilon = 3$|$\varepsilon = 1$|
> |-|-|-|-|
> |Prefix Tuning|0.63|0.62|0.59|
> |LoRA|0.74|0.73|0.72|
> |Full Fine-tune|0.81|0.76|0.72|
> |Head Fine-tune|0.97|0.95|0.92|
>
> **GermanWiki**
>
> |Adaptation|$\varepsilon = 5$|$\varepsilon = 3$|$\varepsilon = 1$|
> |-|-|-|-|
> |Prefix Tuning|0.54|0.54|0.57|
> |LoRA|0.62|0.60|0.55|
> |Full Fine-tune|0.69|0.66|0.62|
> |Head Fine-tune|0.75|0.73|0.72|
>
> BookCorpus2 Val
>
> |Adaptation|$\varepsilon = 5$|$\varepsilon = 3$|$\varepsilon = 1$|
> |-|-|-|-|
> |Prefix Tuning|0.87|0.85|0.77|
> |LoRA|0.69|0.68|0.63|
> |Full Fine-tune|0.88|0.87|0.84|
> |Head Fine-tune|0.80|0.78|0.74|
>
> **BookCorpus2 Train**
>
> |Adaptation|$\varepsilon = 5$|$\varepsilon = 3$|$\varepsilon = 1$|
> |-|-|-|-|
> |Prefix Tuning|0.88|0.86|0.72|
> |LoRA|0.68|0.67|0.63|
> |Full Fine-tune|0.83|0.80|0.79|
> |Head Fine-tune|0.82|0.81|0.74|
>
> **Github Val**
>
> |Adaptation|$\varepsilon = 5$|$\varepsilon = 3$|$\varepsilon = 1$|
> |-|-|-|-|
> |Prefix Tuning|0.91|0.88|0.77|
> |LoRA|0.71|0.69|0.63|
> |Full Fine-tune|0.83|0.82|0.79|
> |Head Fine-tune|0.78|0.77|0.76|
>
> **Enron Val**
> | Adaptation| $\varepsilon = 5$ | $\varepsilon = 3$ | $\varepsilon = 1$ |
> |-|-|-|-|
> | Prefix Tuning | 0.81| 0.74| 0.61|
> | LoRA| 0.70| 0.66| 0.59 |
> | Full Fine-tune | 0.90| 0.87| 0.75|
> | Head Fine-tune | 0.70| 0.64| 0.66|
>
>
> The above results support the trends we have already observed in our evaluations. We see that higher $\varepsilon$ results in higher privacy leakage across all datasets and adaptations.
>
> > **The authors discuss the privacy-utility tradeoffs for the different methods in the appendix. However, this is quite an important topic and I think it is worth moving this to the main paper.**
>
> We agree with the reviewer’s assessment of the results' importance and have added them as a separate RQ6 to the main paper, using the additional page available for the rebuttal.
>
> >**Similarly, it would also be useful to include some discussion on the costs of the different methods and the tradeoffs with privacy and utility.**
>
> Different adaptation methods exhibit varying computational costs and privacy-utility tradeoffs. For a detailed study, we refer to Hanke et al. (2024) [1] who provide a systematic cost analysis across multiple private adaptation methods, measuring both training and inference costs under different privacy budgets ($\varepsilon$ values).  We have also added a reference to this paper in Appendix C.6.
>
> To further address the reviewer’s comment, we performed additional measurements to provide general cost estimates *specific to our experimental setup*. Therefore, we have added the following table showing the training time (in minutes) for each adaptation method on Pythia 1B under differential privacy across all datasets.
>
> | Adaptation | SAMSum | German Wiki | BookCorpus2 Val | BookCorpus2 Train | Github Val | Enron Val |
> |-|-|-|-|-|-|-|
> | Prefix Tuning| 17.2 |18| 16 | 16 | 16 | 16 |
> | LoRA | 15.5 | 6.5 | 5| 5.5 | 5.2| 5.2 |
> | Full Fine-tune | 19.25| 20|18.4|18.5 | 18.2 | 18.2|
> | Head Fine-tune | 3.5| 4.7| 3.2| 3.25| 3.5| 3.5 |
>
> The results show clear efficiency differences: Head Fine-tune is the fastest (3.2-4.7 minutes), followed by LoRA (5.0-15.5 minutes), Prefix Tuning (16-18 minutes), and Full Fine-tune (18.2-20 minutes). Combining these measurements with our privacy-utility analysis in RQ6, these results suggest that LoRA provides the best overall tradeoff: it achieves strong empirical privacy protection, maintains competitive utility, and requires moderate computational cost. We have added the detailed cost analysis and this table about training time in Appendix C.6.

---

> > ### Author Response · Authors · 2025-11-20
> >
> > >**The authors propose a new framework for holistic privacy assessment using adversarial games, but it would help to give examples to demonstrate how this framework would help in practice.**
> >
> > The holistic privacy assessment defines privacy auditing for the entire pretrain-adapt pipeline, rather than independent stages. Prior work analyzed privacy risks in isolation, e.g., only pretraining data or only fine-tuning data leakage.
> >
> > However, in our work, we have shown that, in the pretrain-adapt paradigm, this is insufficient because the pretraining data and the adaptation data may interact. Therefore, our framework enables practitioners to evaluate where and how privacy leakage emerges across the model’s lifecycle. For instance, in a medical LLM trained on data from a single hospital, a patient discovers that their data can be extracted from the model. To assert their rights, this patient needs to know where the leakage comes from: pretraining data S, the hospital’s fine-tuning data D, or both.
> >
> > However, most existing auditing frameworks only attribute leakage to either S or D, and therefore fail to cover important cases: 1) when the same data appears in both S and D, 2) when the data is contained in D but only becomes extractable after adaptation. Our holistic auditing framework allows us to design meaningful privacy audits and to identify privacy risks over LLM’s multi-stage training process. By framing each audit as a well-defined adversarial game, our framework supports reproducible evaluation across models and datasets, offering a unified way for auditing privacy regardless of whether the source is pretraining or adaptation data. This makes the framework applicable to both empirical benchmarking and the development of future tools that combine threat modeling, measurement, and mitigation in a single pipeline. To explain the usability of our framework, we added a paragraph in Appendix I.
> >
> > >**This work only focuses on open-source models, and it is unclear how to generalize the results to close-source models where the pretraining data distribution is not known.**
> > >**How could this work apply to closed-source models? Are there any tools or insights from this work that could apply there?**
> >
> > We benchmark diverse open-source models to understand privacy risks from adapting LLMs, identify patterns in their leakage, and provide guidance on the choice of adaptation method and privacy regimes. These general patterns are likely to hold true for closed models, given their consistency over the diverse open-source models. However, we cannot evaluate closed models for two main reasons:
> >
> > First, the pretraining data of closed LLMs is unknown, as the reviewer states. Therefore, we cannot determine which data is IID and which is OOD. Our benchmark focuses on open-source models because known pretraining data is available to define the IID and OOD sets. As our framework is specifically built for auditors or the model owner themselves, we assume they will always be provided with the relevant information about the pretraining set for proper auditing.
> >
> > Second, most existing closed-source models do not allow adaptations (such as full fine-tuning, soft prompts, prefix tuning, or LoRA) or only to a very limited and controlled extent. Thus, our auditing is inapplicable to closed models.
> >
> > Finally, recent results show that *open LLMs are necessary for current private adaptations and outperform their closed alternatives* [1]. They also indicate that the privacy leakage from closed-source models is higher than from open-source models. Thus, closed LLMs leak at least as much as open-source models, and we focus on the *more interesting* case with higher leakage that we can analyze in depth.
> >
> > **References:**
> >
> > [1] “*Open LLMs are Necessary for Current Private Adaptations and Outperform their Closed Alternatives*” Hanke et al. NeurIPS 2024.
> >
> > >**The work is empirical and does not give any theoretical explanations, for instance on why some adaptations have lower vulnerability than others.**
> > >**Are there explanations on why different adaptation methods perform better than others?**
> >
> > We agree that theoretical insights into why certain adaptation methods offer stronger empirical privacy would be valuable. However, our focus in this paper was to establish a systematic empirical benchmark and identify practical leakage across models, data distributions, and adaptation types. We see our findings as a foundation for future theoretical work, and hope the benchmark will enable further analysis along these lines. We also include discussion points that may guide theoretical follow-ups. In particular, we highlight the connection between the number of parameters of the adaptation methods and the privacy risk.
> >
> >
> > We would greatly appreciate updating the rating if the above responses address Reviewer's concerns.

---

> > > ### Comment · Reviewer_ZdyA · 2025-11-27
> > >
> > > Thank you for the detailed responses and the additional experimental results, which makes the work more comprehensive. I have increased my score accordingly.

---

> > > > ### Author Response · Authors · 2025-11-27
> > > > **Thank you for the discussion**
> > > >
> > > > We thank the Reviewer for going through our responses and increasing their score. Should any further questions arise, we remain available to answer

---

### Official Review · Reviewer_DyRT · 2025-10-31

**Soundness:** 3
**Presentation:** 3
**Contribution:** 3
**Rating:** 6
**Confidence:** 4

**Summary:**

This paper examines how well differential privacy protects sensitive data during large language model adaptation. It studies privacy risks under various data relationships, including overlapping, in-distribution, and out-of-distribution cases, and across several adaptation methods such as full fine-tuning, LoRA, and prefix tuning. The findings show that privacy leakage increases when adaptation data closely matches pretraining data, even without direct overlap, while LoRA offers the strongest empirical protection for out-of-distribution data. The authors also propose a holistic privacy auditing framework that evaluates risks throughout the entire lifecycle of an LLM from pretraining to adaptation to ensure responsible use in sensitive domains

**Strengths:**

1. The paper provides a thorough empirical evaluation across multiple datasets, adaptation methods, and privacy regimes, offering one of the most systematic analyses of differential privacy in LLM adaptations.

2. The introduction of a structured framework for auditing privacy across pretraining and adaptation stages is a novel and forward-looking contribution, addressing a key gap in existing LLM privacy research.

3. The work directly informs real-world use of LLMs in sensitive domains, helping bridge the gap between theoretical guarantees of differential privacy and practical, empirically validated protection.

**Weaknesses:**

1. The most contradictory finding concerns Prefix Tuning: in data extraction attacks against adaptation data (RQ3), it was found to be the most vulnerable method. However, regarding the "forgetting" effect on pretraining data (RQ5), it suddenly became the most effective, showing the least leakage. The paper points out these phenomena but does not deeply explain why this trade-off exists. This makes the question of "which method should practitioners choose?" very ambiguous.
2. The paper's core finding is that "distributional closeness is the main driver of risk". However, the paper's definition of "closeness" is purely categorical, namely "Overlap, IID, and OOD. This categorization is very coarse. The paper provides no quantitative metrics (e.g., Wasserstein distance or KL divergence using word embeddings) to measure the actual distributional distance between the pretraining and adaptation data. This makes the core conclusion, while intuitive, methodologically less rigorous. In practical applications, if two datasets are not clearly IID or OOD, the paper's conclusions are difficult to apply for guidance.

**Questions:**

please see above

---

> ### Author Response · Authors · 2025-11-20
>
> We thank the Reviewer for the positive and detailed feedback. We appreciate that the Reviewer finds our empirical evaluation extensive and to be one of the most systematic, acknowledges the importance of the introduction of our auditing framework, and its use for real-world cases.
>
> In summary, we have further clarified the difference between RQ3 and RQ5, and, based on the reviewers suggestions, have shown the distributional closeness empirically using the Wasserstein distance.
>
> We provide detailed answers in-line below:
>
> > **The most contradictory finding concerns Prefix Tuning: in data extraction attacks against adaptation data (RQ3), it was found to be the most vulnerable method. However, regarding the "forgetting" effect on pretraining data (RQ5), it suddenly became the most effective, showing the least leakage. The paper points out these phenomena but does not deeply explain why this trade-off exists. This makes the question of "which method should practitioners choose?" very ambiguous.**
>
> The difference arises because RQ3 and RQ5 capture *different forms of leakage*. RQ3 measures the extractability of **canaries** from the **adaptation dataset**, whereas RQ5 evaluates the **retention of pretraining data**, specifically memorized points which are not part of the adaptation data.
>
> Our findings over the two research questions can be hence summarized as follows: Prefix Tuning introduces noise during adaptation that can reduce memorization of *pretraining data*, but does not necessarily protect the new *adaptation data* from extraction. Thus, the two results capture complementary effects rather than a contradiction.
>
> In Section 5 of our paper, we point out that a practitioner has to weigh, depending on the situation, which privacy risks they are most concerned about and choose the adaptation accordingly.
>
> > **The paper's core finding is that "distributional closeness is the main driver of risk". However, the paper's definition of "closeness" is purely categorical, namely "Overlap, IID, and OOD. This categorization is very coarse. The paper provides no quantitative metrics (e.g., Wasserstein distance or KL divergence) to measure the actual distributional distance between the pretraining and adaptation data. This makes the core conclusion, while intuitive, methodologically less rigorous. In practical applications, if two datasets are not clearly IID or OOD, the paper's conclusions are difficult to apply for guidance.**
>
> As reported in Section 3, we denote datasets as IID and OOD based on their overlap with the pretraining data:
> We refer to our IID and overlap datasets as those drawn from subsets of the pretraining corpus (the Pile for Pythia), and to our OOD datasets as those not present within the Pile.
>
> Appendix B.1 further explained: “These OOD datasets were selected because of their different degrees of variation from the original
> distribution of the Pile dataset. Although SAMSum shares the same language (English), its general dialogue format, followed by the dialogue summary, is not present in the pretraining set. GermanWiki, on the other hand, presents wide syntactic and lexical variation from the pretraining dataset.”
>
> To address the reviewer’s comment, we extended the evaluation further to also include a measure of distributional distance, using the Wasserstein distance between the pretraining data (all Pile subsets) and our chosen datasets.
>
> Based on the reviewer’s suggestion, we ran a distance evaluation using the Wasserstein distance between all Pile subsets and our chosen adaptation datasets. In the table below, we provide the mean and minimum distances across all subsets, where smaller indicates being closer to the Pile dataset.
>
> |Dataset|Mean Distance|Min Distance|
> |---|---|---|
> |BookCorpus2 Train (Overlap)|0.0171|0.0019|
> |BookCorpus2 Val (IID)|0.0193|0.0057|
> |Github Val (IID)|0.0180|0.0021|
> |Enron Val (IID)|0.0202|0.0088|
> |SAMSum (OOD)|0.0250|0.0192|
> |GermanWiki (OOD)|0.0556|0.0520|
>
> The evaluation with the Wasserstein distance supports our categorical definition of distributional closeness. As shown in the table, datasets chosen from the Pile (BookCorpus, GitHub, Enron) have very small mean and minimum distances, confirming their strong alignment with the pretraining distribution and validating their classification as IID or overlapping. In contrast, SAMSum and GermanWiki show larger distances, demonstrating clear separation from the Pile distribution. SAMSum remains close to the Pile, despite being significantly further than the IID datasets, indicating its general English-language similarity, while GermanWiki is the most distant and thus represents a strongly out-of-distribution case. We thank the reviewer for their suggestion and believe that this quantitative evaluation supports our categorical distinctions and offers a clear, data-driven measure of distributional closeness.
>
> We added this table and its corresponding discussion to the dataset description in Section 3.

---

> > ### Author Response · Authors · 2025-11-27
> >
> > We would like to follow up on our rebuttal to ensure that our responses adequately address the Reviewer's concerns. Specifically, we have:
> >
> > - clarified the behavior of Prefix Tuning on the pretraining vs. the adaptation data across the different research questions;
> >
> > - added quantitative analyses of distributional differences between our pretraining data (all Pile subsets) and our adaptation datasets by computing Wasserstein distances using sentence embeddings, as now detailed in Table 1, Section 3 of the updated paper.
> >
> >
> >
> > If the above additions address the reviewer’s concerns, we would be grateful if they could consider increasing their score accordingly.

---

> > > ### Comment · Reviewer_DyRT · 2025-11-27
> > >
> > > Thanks for the reply. I think the clarification of the conclusion makes it clearer, and the new results on the Wasserstein distance are interesting. I will adjust my score accordingly.

---

> > > > ### Author Response · Authors · 2025-11-28
> > > >
> > > > We thank the Reviewer for their time invested in reviewing our submission. Should any further questions arise, we remain available to answer.

---

### Official Review · Reviewer_61VY · 2025-10-31

**Soundness:** 4
**Presentation:** 4
**Contribution:** 3
**Rating:** 8
**Confidence:** 4

**Summary:**

This work tests how well Differential Privacy (DP) protects sensitive information when adapting Large Language Models (LLMs) to new data. The authors evaluate a variety of DP adaptation methods including Prefix Tuning, LoRA, Full Fine-Tune, and Head Fine-Tune. They evaluate if they can identify (membership inference attack) or extract (data extraction attacks) specific data used during adaptation. They test this for 4 different types of data: overlapping data, in distribution data, out of distribution data and canary data (only used for extraction). They evaluate seven pretrained models such as Pythia, GPT-Neo and OLMo that are all trained on the Pile dataset. The authors provide 5 interesting insights from these experiments such as
- Higher privacy regime is required for data closer to the pretraining distribution for similar empirical privacy.
- Efficient methods like LoRA generally offer better privacy than full fine-tuning
- Common privacy settings (like $\epsilon=8$) aren't strong enough to stop powerful attacks
- The performance of MIAs highly depends on the attacker’s knowledge of the target model and pretraining data

Finally, the authors also provide a new way to holistically audit the privacy of pretraining and private adaptation together.

**Strengths:**

- The paper evaluates a common paradigm of adapting non-privately pretrained LLMs with sensitive data using privacy preserving mechanisms.
- Very systematic evaluation of empirical privacy risks when varying different parameters like distance of private data from pretraining, type of adaptation and attacker knowledge.
- Covers a large number of models and adaptation mechanisms.

**Weaknesses:**

- Only 2 epsilons are evaluated, 0.1 which is considered very private and 8 which is considered pretty large. Would be nice to also have something in the middle.

**Questions:**

- Could the authors clarify measure the distributional closeness of the OOD vs IID datasets used?
- This might be a stretch but I think it might be interesting to compare a DP pretrained model like VaultGemma in this comparison as well. It is recently released and the dataset is not publicly known but it might be a nice addition to this analysis.

---

> ### Author Response · Authors · 2025-11-20
>
> We thank the Reviewer for the positive and detailed feedback. We appreciate that the Reviewer acknowledges the relevance of evaluating privacy-preserving adaptation of non-privately pretrained LLMs, the systematic evaluation across different data, adaptation types, and attacker knowledge, and the large number of models and methods covered in our benchmark.
>
> In summary, based on the feedback of the reviewer we show new results for $\varepsilon = \\{1, 3, 5\\}$ for all datasets and adaptation and give further insights into the distributional closeness between overlap, IID, and OOD datasets with the Wasserstein difference.
>
> We provide detailed answers in-line below:
>
> >**Only 2 epsilons are evaluated, 0.1 which is considered very private and 8 which is considered pretty large. Would be nice to also have something in the middle.**
>
> We thank the reviewer for their suggestion. In the originally submitted version of our paper in Appendix C1, Figure 6, we showed results for $\varepsilon \in \\{0.1, 0.5, 1, 3, 8, 50, \infty\\}$ for BookCorpus2 Val and SAMSum. The trends show that, as expected, higher $\varepsilon$ leads to higher privacy leakage.
>
> To address the reviewer’s comment further, we added more experiments and ran MIAs for *all* datasets and *all* adaptation methods under  $\varepsilon \in  \\{1, 3, 5\\}$ on Pythia 1B and added the full results into the paper as a new Figure 7. These results show a smooth decrease in empirical leakage as $\varepsilon$ decreases.
>
> For convenience, we also report the observed results in the table below:
>
> **SAMSum**
>
> |Adaptation|$\varepsilon = 5$|$\varepsilon = 3$|$\varepsilon = 1$|
> |---|---|---|---|
> |Prefix Tuning|0.63|0.62|0.59|
> |LoRA|0.74|0.73|0.72|
> |Full Fine-tune|0.81|0.76|0.72|
> |Head Fine-tune|0.97|0.95|0.92|
>
> **GermanWiki**
>
> |Adaptation|$\varepsilon = 5$|$\varepsilon = 3$|$\varepsilon = 1$|
> |---|---|---|---|
> |Prefix Tuning|0.54|0.54|0.57|
> |LoRA|0.62|0.60|0.55|
> |Full Fine-tune|0.69|0.66|0.62|
> |Head Fine-tune|0.75|0.73|0.72|
>
> **BookCorpus2 Val**
>
> |Adaptation|$\varepsilon = 5$|$\varepsilon = 3$|$\varepsilon = 1$|
> |---|---|---|---|
> |Prefix Tuning|0.87|0.85|0.77|
> |LoRA|0.69|0.68|0.63|
> |Full Fine-tune|0.88|0.87|0.84|
> |Head Fine-tune|0.80|0.78|0.74|
>
> **BookCorpus2 Train**
>
> |Adaptation|$\varepsilon = 5$|$\varepsilon = 3$|$\varepsilon = 1$|
> |---|---|---|---|
> |Prefix Tuning|0.88|0.86|0.72|
> |LoRA|0.68|0.67|0.63|
> |Full Fine-tune|0.83|0.80|0.79|
> |Head Fine-tune|0.82|0.81|0.74|
>
> **Github Val**
>
> |Adaptation|$\varepsilon = 5$|$\varepsilon = 3$|$\varepsilon = 1$|
> |---|---|---|---|
> |Prefix Tuning|0.91|0.88|0.77|
> |LoRA|0.71|0.69|0.63|
> |Full Fine-tune|0.83|0.82|0.79|
> |Head Fine-tune|0.78|0.77|0.76|
>
> **Enron Val**
>
> |Adaptation|$\varepsilon = 5$|$\varepsilon = 3$|$\varepsilon = 1$|
> |---|---|---|---|
> |Prefix Tuning|0.81|0.74|0.61|
> |LoRA|0.70|0.66| 0.59|
> |Full Fine-tune |0.90|0.87|0.75|
> |Head Fine-tune |0.70|0.64|0.66|
>
>
> These results again support our previous evaluations. The further we decrease the value of $\varepsilon$ the lower the privacy risk observed for the adaptation data. Furthermore, we see that LoRA again has, on average, the lowest privacy risk across all datasets, similar to what we saw for $\varepsilon = \\{0.1, 8.0\\}$ in the paper.

---

> > ### Author Response · Authors · 2025-11-20
> >
> > >**Could the authors clarify measure the distributional closeness of the OOD vs IID datasets used?**
> >
> > As reported in Section 3, we denote datasets as IID and OOD based on their overlap with the pretraining data:
> > We refer to IID and overlap datasets as those drawn from the subsets of the pretraining corpus (the Pile for Pythia) to the OOD datasets as those not present within the Pile.
> >
> > Appendix B.1 further explained: “These OOD datasets were selected because of their different degrees of variation from the original distribution of the Pile dataset. Although SAMSum shares the same language (English), its general dialogue format, followed by the dialogue summary, is not present in the pretraining set. GermanWiki, on the other hand, presents wide syntactic and lexical variation from the pretraining dataset.”
> >
> > To address the reviewer’s comment, we extended the evaluation further to also include a measure of distributional distance, using the Wasserstein distance between the pretraining data (all Pile subsets) and our chosen datasets.
> >
> > The table below reports the mean and minimum distances across all subsets, with smaller values indicating greater similarity to the Pile distribution.
> >
> > |Dataset|Mean Distance|Min Distance|
> > |---|---|---|
> > |BookCorpus2 Train (Overlap) |0.0171|0.0019|
> > |BookCorpus2 Val (IID)|0.0193|0.0057|
> > |Github Val (IID)|0.0180|0.0021|
> > |Enron Val (IID)|0.0202|0.0088|
> > |SAMSum (OOD)| 0.0250|0.0192|
> > |GermanWiki (OOD)| 0.0556|0.0520|
> >
> > These results confirm the categorizations we chose: datasets taken from the pretraining dataset Pile (i.e., BookCorpus, GitHub, Enron) show very small distances to the pretraining data, validating their classification as IID or overlapping, while OOD datasets (SAMSum and GermanWiki) show both higher minimum and mean distances. SAMSum remains close to the Pile, despite being significantly further than the IID datasets, indicating its general English-language similarity, whereas GermanWiki is clearly the most distant, making it the most OOD. We added this table and its corresponding discussion to the dataset description in Section 3.
> >
> >
> > >**This might be a stretch but I think it might be interesting to compare a DP pretrained model like VaultGemma in this comparison as well. It is recently released and the dataset is not publicly known but it might be a nice addition to this analysis.**
> >
> > As the Reviewer noted, the main obstacle is that the pretraining dataset of VaultGemma is not publicly available, which prevents reliable IID and even OOD comparison in our framework. Because our benchmark explicitly relies on known relationships between pretraining and adaptation data to categorize data as IID or OOD, we cannot fairly assess VaultGemma within our evaluation setup.

---

> > > ### Comment · Reviewer_61VY · 2025-11-25
> > >
> > > Thank you for the additional analysis. I had missed the results for the other epsilons in the appendix so I think it is good addition to the main writeup.

---

> > > > ### Author Response · Authors · 2025-11-26
> > > >
> > > > We thank the Reviewer for their time invested in reviewing our submission. Should any further questions arise, we remain available to answer.

---

### Author Response · Authors · 2025-11-20

We would like to thank all the Reviewers for their valuable feedback and insightful comments, which greatly helped us further improve our submission. Reviewers recognized that our work addresses an important and practically relevant problem: empirical privacy risk assessment for DP-protected LLMs (Reviewer hzjs), which is highly relevant to practical applications (Reviewer ZdyA) and helps bridge the gap between theoretical DP guarantees and empirically validated protection (Reviewer DyRT). All reviewers acknowledged our comprehensive and systematic evaluation across multiple models, datasets, adaptation methods, and privacy regimes (Reviewers 61VY, DyRT, ZdyA, hzjs). Additionally, Reviewer DyRT described our holistic privacy auditing framework as "a novel and forward-looking contribution", addressing a key gap in existing LLM privacy research.

To address the reviewers' concerns and strengthen our work, we conducted additional experiments and provided detailed clarifications:

1. **Extended privacy budget analysis**: In response to concerns from Reviewers 61VY and ZdyA about extending the range of epsilon values, we conducted additional experiments across $\varepsilon \in \\{1, 3, 5\\}$ in addition to our original $\varepsilon \in \\{0.1, 8, \infty \\}$ for all datasets and adaptation methods on Pythia 1B. These results demonstrate smooth trends in privacy leakage as epsilon varies, and have been added as a new figure in the main paper.
2. **Quantitative distributional distance measurement**: Addressing the suggestion by Reviewers DyRT and 61VY to *quantify* the distributional distances between our pretraining and various adaptation datasets, we computed Wasserstein distances between all Pile subsets (pretraining data) and our adaptation datasets using sentence embeddings. This quantitative analysis empirically validates our categorical IID/OOD classifications and has been added to Section 3.
3. **Privacy-utility tradeoffs**: Responding to Reviewer ZdyA's request to move the privacy-utility analysis to the main paper, we added a new research question 6 that systematically presents the privacy-utility trade-offs for different adaptation methods.
4. **Enhanced evaluation and clarifications**: We added training time measurements for cost analysis (Reviewer ZdyA), provided Rouge-1 scores and perplexity to validate utility metrics (Reviewer hzjs), clarified the Prefix Tuning behavior across different research questions (Reviewer DyRT), and provided more detailed explanations of the holistic auditing framework (Reviewer hzjs).

We believe these improvements substantially address the Reviewers' concerns and strengthen the practical applicability of our work. Once again, we thank the Reviewers for their constructive feedback. We look forward to further discussions and continued advancements in this evolving field. Please let us know if we can provide any additional information to further enhance your assessment of our work.

---

### Meta-Review · Area_Chair_wLkd · 2025-12-17

**Summary:**

The 4 reviewers had initial concerns. During the discussion, the 4 reviewers were convinced by the authors' rebuttal. This is clear when checking the text of the concerns, the rebuttal, and the reviewers' final words. 3 out of 4 reviewers declared that they have increased their score.

**Reviewer Concerns:**

Part ot the process is here.

**RC - 61VY ** Only 2 epsilons are evaluated, 0.1 which is considered very private and 8 which is considered pretty large. Would be nice to also have something in the middle. _Addressed according to the reviewer, checked by the AC_

**RC - DyRT ** Contradictory finding concerns Prefix Tuning. _Addressed according to the reviewer, checked by the AC_

**RC - DyRT ** "Distributional closeness is the main driver of risk". However, the paper's definition of "closeness" is purely categorical. _Addressed according to the reviewer, checked by the AC_

**Reviewer Scores:**

Reviwer - 61VY, according to their interaction, did not increase their score.

Reviwer - DyRT, according to their interaction, increased their score. AC Guess from 6 to 8

Reviwer - ZdyA, according to their interaction, increased their score. AC Guess from 4 to 6

Reviwer - ZdyA, according to their interaction, increased their score from 4 to 6 (as declared by the reviewer)

---

### Decision · Program_Chairs · 2026-01-26

Accept (Oral)